# ViTime: Foundation Model for Time Series Forecasting Powered by Vision Intelligence

**Luoxiao Yang**                                         *luoxiyang2-c@my.cityu.edu.hk*
*School of Automation and Information Engineering, Xi'an University of Technology, Xi'an, China*
*Electrical and Computer Engineering, Technion – Israel Institute of Technology, Israel* [*]

**Yun Wang**                                             *wangyunjeff@gmail.com*
*Electrical and Computer Engineering, Technion – Israel Institute of Technology, Israel*

**Xinqi Fan**                                            *x.fan@mmu.ac.uk*
*Department of Computing and Mathematics, Manchester Metropolitan University, UK*

**Israel Cohen**                                         *icohen@ee.technion.ac.il*
*Electrical and Computer Engineering, Technion – Israel Institute of Technology, Israel*

**Jingdong Chen**                                        *jingdongchen@ieee.org*
*Department of Information and Communication Engineering, Northwest Polytechnical University, Xi'an, China*

**Zijun Zhang** [†]                                      *zijzhang@cityu.edu.hk*
*Department of Data Science,City University of Hong Kong, Hong Kong SAR*

**Reviewed on OpenReview:** *https://openreview.net/forum?id=XInsJDBIkp*

**Code and Pretrained Models:** *https://github.com/IkeYang/ViTime*

## Abstract

Time series forecasting (TSF) possesses great practical values in various fields, including power and energy, transportation, etc. TSF methods have been studied based on knowledge from classical statistics to modern deep learning. Yet, all of them were developed based on one fundamental concept, the numerical data fitting. Thus, the models developed have long been known to be problem-specific and lacking application generalizability. Practitioners expect a TSF foundation model that serves TSF tasks in different applications. The central question is then how to develop such a TSF foundation model. This paper offers one pioneering study in the TSF foundation model development method and proposes a vision intelligence-powered framework, ViTime, for the first time. ViTime fundamentally shifts TSF from numerical fitting to operations based on a binary image-based time series metric space and naturally supports both point and probabilistic forecasting. We also provide rigorous theoretical analyses of ViTime, including quantization-induced system error bounds and principled strategies for optimal parameter selection. Furthermore, we propose RealTS, an innovative synthesis algorithm generating diverse and realistic training samples, effectively enriching the training data and significantly enhancing model generalizability. Extensive experiments demonstrate ViTime's state-of-the-art performance. In zero-shot scenarios, ViTime outperforms TimesFM by 9-15%. With just 10% fine-tuning data, ViTime surpasses both leading foundation models and fully-supervised benchmarks, a gap that widens with 100% fine-tuning. ViTime also exhibits exceptional robustness, effectively handling missing data and outperforming TimesFM by 20-30% under various data perturbations, validating the power of its visual space data operation paradigm.

---

[*]Work performed while at XAUT and Technion.
[†]Corresponding author

# 1 Introduction

Time series forecasting (TSF) is a classic but challenging topic that has been vigorously discussed in various application fields, including power and energy (Sharadga et al., 2020), environmental studies (Jacox et al., 2022), transportation studies (Lei et al., 2022), weather forecasting (Yang et al., 2021), stock market analysis (Lin et al., 2011), public healthcare (Liu et al., 2024). Although new heights of accuracies were repeatedly refreshed by new studies (Zhou et al., 2021; Wu et al., 2021; Nie et al., 2022; Zeng et al., 2023; Patro & Agneeswaran, 2024; Wu et al., 2022), most reported methods predominantly relied on a numerical fitting based modeling paradigm so that models were often dataset- or problem-specific and lack of application generalizability. The need to repeatedly train models for various TSF tasks has been the critical barrier of promoting applications of learning-based TSF methods in practice, especially ones with sophisticated mechanisms. Developing a TSF foundation model capable of serving diverse TSF tasks across different applications is thus of great practical value. The central question then becomes: *how can we develop such a TSF foundation model?*

Studying the TSF foundation model is still in its early stages, and existing efforts observed in literature are mainly devoted to exploring Large Language Model (LLM)-based and numerical fitting-based models. The LLM-based model leverages the inference capabilities of LLMs for zero-shot TSF tasks, including TimeGPT-1 (Garza & Mergenthaler-Canseco, 2023) and TIME-LLM (Jin et al., 2023). However, the prediction accuracy of LLM-based models heavily depends on the underlying capabilities of LLM, and to achieve optimal performance, the competent large language models, such as GPT-4 or Claude 3.5 (Zhou et al., 2023a), are usually employed. Meanwhile, in fine-tuning LLM-based TSF foundation models for handling various downstream tasks demanding higher precision, the computational complexity becomes prohibitively expensive, resulting in a large, redundant, less precise, and cost-ineffective paradigm for the TSF foundation model (Tan et al., 2024).

The dominant paradigm in Time Series Forecasting (TSF) is currently centered on numerical fitting-based models, such as TimesFM (Das et al., 2024) and ForecastPFN (Dooley et al., 2024). These models operate by directly learning the numerical correlations along the temporal dimension of the data. However, this purely numerical-driven approach diverges from human cognitive processes. Studies in cognitive science indicate that for tasks like trend conjecture and forecasting, humans preferentially process and remember correlations between visual representations rather than directly handling abstract numerical sequences (Pettersson, 1993; Dondis, 1974). For instance, Dondis (Dondis, 1974) noted that the visual cortex is adept at rapidly identifying patterns, shapes, and colors, making the processing of visual information more efficient than that of text or numbers.

Inspired by this cognitive paradigm, the research community has begun to explore the potential of applying vision intelligence to TSF. Early attempts involved transforming time series into images, either through direct plotting for image-to-image forecasting (Sood et al., 2021) or via multi-view visual encodings like Gramian Angular Fields (GAF), Markov Transition Fields (MTF), and Recurrence Plots (Wang & Oates, 2015; Eckmann et al., 1987). More recently, researchers have repurposed powerful vision backbones, such as Vision Transformers and Masked Autoencoders, for the TSF domain (Du et al., 2024; Chen et al., 2024). While these works have shown the promising potential, they remain largely heuristic. They often lack a rigorous theoretical framework for visual quantization and metrics, and the exploration of foundation-model paradigms within this context remains limited. These observations culminate in a fundamental question: *On the path toward a TSF foundation model, could leveraging vision intelligence as a core modeling paradigm, alongside conventional numerical methods, offer a more promising avenue?*

In addition, training data of TSF tasks typically consist of large-scale real-world datasets (Das et al., 2024), raising a critical question: *Can real-world datasets comprehensively capture the diverse range of universal time series patterns?* Specifically, what kind of foundational capabilities should a TSF foundation model possess to address a universal spectrum of time series problems?

To tackle these challenges, this paper develops a novel vision intelligence-based TSF foundation model, a Visual Time Foundation Model (ViTime), aiming to pioneer a new computational paradigm of building the TSF foundation model from the perspective of vision intelligence. Regarding the computational principle

innovation aspect, ViTime operates by transforming numerical time series into binary images, converting numerical temporal correlations into pixel spatial patterns, and solving TSF tasks, including both point and probabilistic forecasting, in binary image space. We provide detailed theoretical analyses of quantization-induced errors and establish principled guidelines for optimal parameter settings, ensuring precise control over the trade-off between computational complexity and prediction accuracy. To offer a large volume of sufficiently diverse samples for training ViTime, an innovative time-series-data generation method, Real-Time Series (RealTS), is proposed. RealTS categorizes foundational knowledge of time series analysis into "trend" and "periodicity" and synthesizes training data during the training of ViTime, ensuring it captures essential time series characteristics. Experimental results demonstrate that ViTime can achieve SOTA performance across diverse scenarios, including zero-shot generalization, fine-tuning with limited data, and robustness to data perturbations.

The main contributions of this work are listed as follows:

- **Novel Theoretical Framework for Vision Intelligence Powered TSF.** We introduce ViTime, a pioneering TSF foundation model grounded in a novel theoretical framework that shifts from conventional numerical fitting to operations within a **formally defined binary image-based time series metric space**.

- **RealTS: Advanced Data Generation and Augmentation for TSF Foundation Modeling.** To address the training-data sample diversity challenge in developing a TSF foundation model, the RealTS, a sophisticated time-series data generation method that synthesizes diverse and high-quality training data, is designed to ensure ViTime can generalize to a wide range of time series patterns.

- **Empirical Validation of Theoretical Advantages and SOTA Performance.** The efficacy of ViTime's theoretically-grounded visual intelligence paradigm is extensively validated. ViTime significantly outperforms existing foundation models and supervised benchmarks in both zero-shot point forecasting (e.g., 9-15% improvement over TimesFM) and zero-shot probabilistic forecasting, few-shot fine-tuning, and robustness against diverse data perturbations (e.g., 20-30% better than TimesFM with missing data/perturbations), confirming the practical benefits of our theoretical contributions.

## 2 Related Work

### 2.1 Problem-specific Model for TSF

The problem-specific TSF methods adopt a fully supervised learning paradigm, where specific models are trained on particular datasets. Early discussions on problem-specific TSF modeling were mainly conducted on classical statistical and machine learning models, such as autoregressive (AR) models and AR variants (Vu, 2007), Splines and their extensions (Lewis & Stevens, 1991), linear regressors (Montgomery et al., 2015), support vector regressor (Montgomery et al., 2015), neural network based regressor (Montgomery et al., 2015), etc. In comparison, the latest TSF studies have shed light on modern deep learning methods, such as recurrent neural network (RNN) and RNN variants (Hewamalage et al., 2021), transformer and various transformer-based models (Zhou et al., 2021; Wu et al., 2021; Nie et al., 2022; Liu et al., 2023), Dlinear (Zeng et al., 2023), TimeMixer (Wang et al., 2024), Mamba based method (Patro & Agneeswaran, 2024) etc.

### 2.2 Foundation Model for TSF

Inspired by recent breakthroughs of pretrained foundation models in natural language processing and computer vision, the TSF community has actively explored developing domain-general foundation models capable of forecasting across diverse datasets and scenarios. Current TSF foundation model studies in general fall into three categories: LLM-based, numerical-data-based, and the emerging vision-based approaches.

**LLM-based Models:** Several recent studies have directly adapted LLMs to forecasting tasks. Methods such as PromptCast (Xue & Salim, 2023), TIME-LLM (Jin et al., 2023), GPT4TS (Zhou et al., 2023b), TimeGPT-1 (Garza & Mergenthaler-Canseco, 2023), and LLM4TS (Chang et al., 2025) recast numerical

forecasting into text-based prompting or embedding alignment tasks. Despite their promising zero-shot forecasting capabilities, these models suffer from inherent limitations, including high computational costs, inefficiency, and domain adaptation complexity arising from fundamental discrepancies between linguistic structures and numerical temporal patterns (Tan et al., 2024).

**Numerical-data-based Models:** To address these limitations, another prevalent research direction exploits large-scale collections of real-world numerical time series to train foundation models. Representative methods include TimesFM (Das et al., 2024), Moirai (Woo et al., 2024), Chronos (Ansari et al., 2024), Moment (Goswami et al., 2024), Lag-Llama (Rasul et al., 2023), GTT (Feng et al., 2024), and TSMamba (Ma et al., 2024). Although these real-data-based models significantly enhance zero-shot generalization, their performance heavily depends on the quality, diversity, and representativeness of available real datasets. Moreover, they typically suffer substantial performance degradation when encountering data perturbations, missing values, or unseen temporal patterns. Furthermore, the reliance on extensive real-world datasets inherently risks test set leakage, as partial segments of test data may inadvertently appear during training, undermining true generalization evaluation. Recognizing these inherent limitations of real-world numerical data, recent work has explored alternative data sources. ForecastPFN (Dooley et al., 2024) trains Transformer-based models purely on synthetic numerical data generated from predefined trend and seasonality components, demonstrating limited but promising zero-shot forecasting abilities. However, due to the uncontrolled or oversimplified synthesis patterns, these synthetic-data-based methods often fail to capture the richness and complexity of real-world scenarios, thereby limiting forecasting accuracy and robustness.

**Vision-based Models:** The idea of representing time series as images to leverage powerful vision models has been explored, but its application as a primary forecasting paradigm is an area of growing interest. Early methods, such as Gramian Angular Fields (GAF) (Wang & Oates, 2015), Markov Transition Fields (MTF), and Recurrence Plots (Eckmann et al., 1987), transformed time series into images primarily for classification tasks, demonstrating the potential of image representations to reveal patterns not obvious in the 1D domain. Recently, this concept has been extended to forecasting. One intuitive approach is direct plotting, e.g., VisualAE (Sood et al., 2021) pioneered this by treating TSF as an image-to-image regression task, where a line plot is converted by a convolutional autoencoder. Other works focus on adapting vision architectures, e.g., Swin4TS (Du et al., 2024) reshaped the time series into 2D patches to apply the Swin Transformer. To enrich the input, models like LDM4TS (Ruan et al., 2025) employed a multi-view strategy, converting a time series into multiple images using techniques like segmentation and GAFs. Recently, VisionTS (Chen et al., 2024) proposed repurposing pretrained vision models (specifically, masked autoencoders trained on ImageNet) for TSF by reformulating forecasting as an image reconstruction problem. Nevertheless, directly reusing models pretrained on natural images introduces a significant domain mismatch that the visual features learned from natural images may not optimally represent temporal structures inherent in numerical time series. Furthermore, all these existing vision-based methods still fundamentally rely on numerical-space analyses and empirical mappings (e.g., standard plotting libraries or heuristic reshaping), lacking a rigorous theoretical framework explicitly tailored for visual representation and quantization of numerical sequences.

**Our Contribution in Context:** In contrast to aforementioned paradigms, our proposed ViTime framework introduces two fundamental shifts in the TSF foundation model design:

Firstly, recognizing intrinsic limitations of numerical-space-based forecasting, such as limited generalization across scales and sensitivity to data perturbations, ViTime explicitly advocates modeling time series directly in a principled visual representation space. Where prior visual methods use heuristic transformations, ViTime pioneering in rigorously defining a dedicated visual metric space for numerical time series, providing theoretical analysis of quantization-induced errors, and offering principled guidance for optimal parameter selection. We also proved that this rigorous visual modeling framework can significantly enhance the signal-to-noise ratio (SNR) of time series and improve forecasting accuracy and interpretability.

Secondly, given the inherent challenges of relying on real-world numerical datasets (limited diversity, data leakage risks), we propose, RealTS, a controlled data synthesis strategy focusing on fundamental time series components (trend, periodicity) to generate structurally sound training data. The RealTS substantially mitigates data leakage risks and enriches training data diversity, enabling ViTime to generalize robustly across diverse real-world scenarios. As demonstrated by extensive experiments, ViTime sets new SOTA

zero-shot and limited-data forecasting benchmarks, significantly outperforming existing foundation models across diverse evaluation settings.

# 3 Method

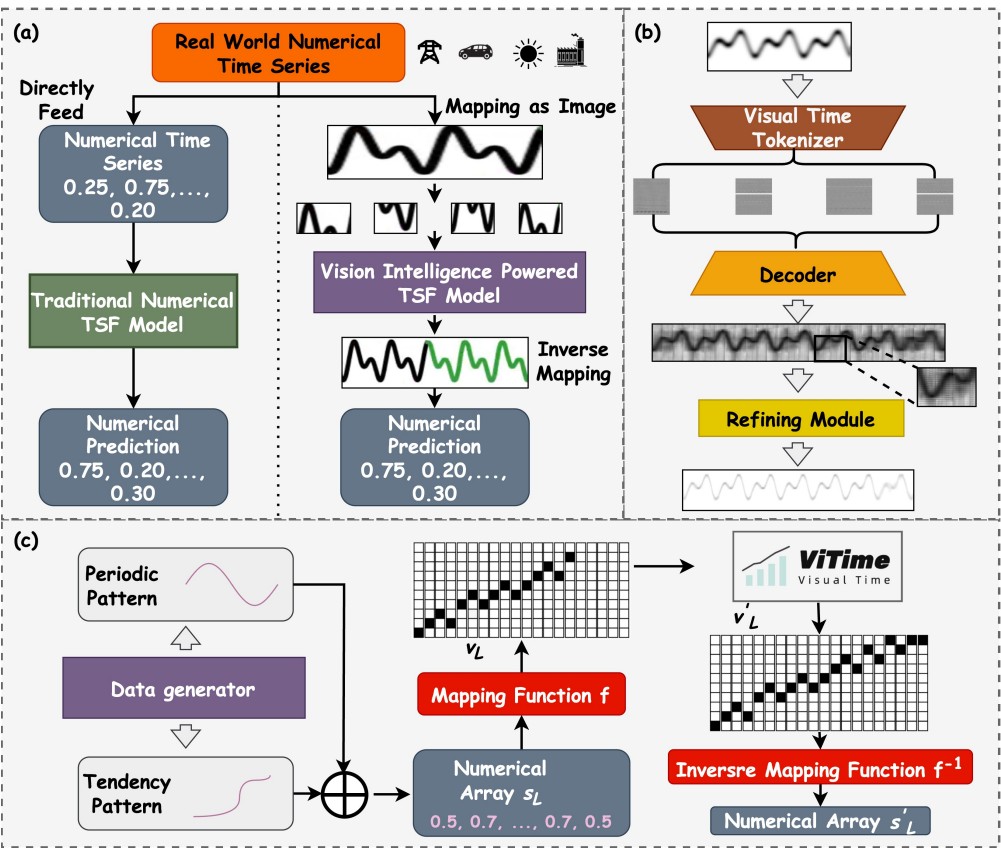

Figure 1: ViTime architecture overview. (a) Pipeline comparison between ViTime and traditional numerical TSF models, showing ViTime's paradigm shift to binary image space processing. (b) ViTime network with three modules: Visual Time Tokenizer, Decoder, and Refining Module. (c) Complete architecture: RealTS synthesis for diverse training samples, mapping function for numerical-to-binary conversion, ViTime model for visual pattern learning, and inverse mapping for prediction output, enabling zero-shot generalization across real-world time series tasks.

## 3.1 Overall Architecture

The overall framework of ViTime, schematically illustrated in Fig. 1 (c), comprises four key modules: the RealTS synthesis module, the mapping function, the proposed ViTime model, and the inverse mapping function. To address the dataset challenge of training a robust TSF foundation model, RealTS synthesizes a vast and diverse set of training samples by categorizing foundational knowledge of time series analysis into "trend" and "periodicity" patterns, which ensures ViTime captures essential time series characteristics across a wide range of scenarios. The core innovation of ViTime lies in its computational principle of mapping numerical time series into binary images. This approach allows ViTime to remember temporal pattern correlations through ordered pixel coordinates while maintaining the ability to convert results back to numerical format. The visual modeling process of ViTime learns to extract relevant features and patterns from the time series visual representation, utilizing the historical distributions of the generated binary images to predict future trends. Finally, the inverse mapping function is employed to convert the predicted image back into numerical time series data for further analysis.

In the following sections, we will introduce each component of ViTime in detail: RealTS, mapping & inverse mapping function, and ViTime Model.

## 3.2 Real-Time Series Synthesis

In this paper, we hypothesize that a robust foundation model for TSF should integrate two essential types of time series fluctuation knowledge, the periodic and trend patterns, which encompass the inherent patterns and directional changes in time series data. Real-world datasets, however, often lack representation of the full spectrum of these periodic and trend-based fluctuations, limiting the ability of the model to generalize across different scenarios and effectively learn underlying dynamics.

To address this challenge, we propose a novel time series generation algorithm, RealTS. RealTS systematically generates a large volume of synthetic time series data that exhibit diverse periodic and trend characteristics. The proposed RealTS can facilitate more comprehensive training of foundation models, exposing them to various patterns and improving their ability to generalize to unseen real-world data.

The RealTS algorithm probabilistically selects between generating periodic or trend-based time series. Given the total length $L$ of the synthesized time series, the algorithm determines the data prior hypothesis between periodic $\varphi_p$ and trend-based $\varphi_t$ patterns with probability $(\alpha)$. The distribution of generated time series $P(D)$ is defined as follows:

$$
\begin{aligned}
\mathbf{s_L} &\sim P(D) = P\left(\mathbf{s_L}|L\right) \\
&= \alpha \int P\left(\mathbf{s_L}|L, B_p\right) P\left(B_p|\varphi_p\right) P\left(\varphi_p\right) d\varphi_p + (1-\alpha) \int P\left(\mathbf{s_L}|L, B_t\right) P\left(B_t|\varphi_t\right) P\left(\varphi_t\right) d\varphi_t
\end{aligned}
\tag{1}
$$

where $\mathbf{s_L}$ is the synthesized time series with length $L$; $P(\varphi)$ represents the prior probability of hypothesis $\varphi$; $P(B|\varphi)$ is the likelihood of observing the data behavior $B$ under hypothesis $\varphi$. Data behavior $B$ is introduced to further detail the generation behavior within different data modes. RealTS employs two data behavior modes for periodic hypothesis and three for trend hypothesis as follows:

- **Periodic Hypothesis:** Inverse Fast Fourier Transform Behavior (IFFTB) and Periodic Wave Behavior (PWB).

- **Trend Hypothesis:** Random Walk Behavior (RWB), Logistic Growth Behavior (LGB) and Trend Wave Data Behavior (TWDB)

Detailed formulas for each behavior mode and illustrative examples are provided in Supplementary Section A.

## 3.3 Binary Image-based Time Series Metric Space

In ViTime, time series are fed and operated with a binary image form, leveraging a binary image-based time series metric space, as described in Definition 3.1.

**Definition 3.1** (Binary image-based time series metric space)**.** The binary image-based time series metric space is defined as a group $(V, d)$, where $V$ is a set of elements defined in Equation (2):

$$
\mathcal{V} = \left\{ v \in \mathbb{R}^{c \times h \times L} \,\middle|\, v_{i,j,k} \in \{0,1\}, i \in [c], j \in [h], k \in [L], \sum_{j=1}^{h} v_{i,j,k} = 1 \right\}
\tag{2}
$$

where $d : V \times V \to \mathbb{R}$ is a distance function based on the Earth Mover's Distance (EMD), as defined in Equation (3):

$$
d\left(v_1, v_2\right) = \int_{i=1}^{c} \int_{k=1}^{L} \inf_{\gamma \in \prod\left(\mathbf{v_1^{i,1:h,k}}, \mathbf{v_2^{i,1:h,k}}\right)} \mathbb{E}_{x,y \sim \gamma} \|x - y\|_1 \, dk \, di
\tag{3}
$$

where $c$ represents the number of variates, $L$ is the length of the time series, and $h$ is the resolution of $V$.

To enable the transition from numerical time-series values to the binary image-based metric space, we introduce mapping and inverse mapping functions as follows. Let $\mathcal{S} = \left\{ s \in R^{c \times L} \mid s_{i,k} \in R \right\}$ represent the numerical value space of time series. The Time-Series-to-Image mapping function $f : S \to \mathcal{V}$ and the Image-to-Time-Series inverse mapping function $f^{-1} : \mathcal{V} \to S$ can be defined as follows:

$$\mathbf{v_{i,1:h,k}} = \mathbf{f}\left(s_{i,k}\right) = \langle f_1\left(s_{i,k}\right), f_2\left(s_{i,k}\right), \ldots f_h\left(s_{i,k}\right) \rangle$$

$$f_j\left(s_{i,k}\right) = \begin{cases} 1, & \text{if } s_{i,k} \geq \text{MS}, j = h \\ 1, & \text{if } s_{i,k} \leq -\text{MS}, j = 1 \\ 1, & \text{if } j = \left\lfloor \frac{s_{i,k} + \text{MS}}{\frac{2\text{MS}}{h}} \right\rfloor \\ 0, & \text{otherwise.} \end{cases}, \quad j \in [h] \tag{4}$$

The Image-to-Time-Series inverse mapping function $f^{-1} : \mathcal{V} \to \mathcal{S}$ can be defined as follows:

$$s_{i,k} = \mathbf{f}^{-1}\left(\mathbf{v_{i,1:h,k}}\right) = \sum_{j=1}^{h} \left( (j - 0.5)\frac{2\text{MS}}{h} - \text{MS} \right) v_{i,j,k} \tag{5}$$

where MS > 0 denotes the maximum scale of $\mathcal{V}$. Before mapping, zero-score normalization is typically applied to the numerical time series $s_{i,k}$ to standardize the scale.

Given that the numerical data synthesized by RealTS are one-channel time series, i.e., $\mathbf{s}_L \in R^{1 \times L}$, thus the corresponding $\mathbf{v_L} \in R^{1 \times h \times L}$ is obtained via

$$\mathbf{v_L} = \mathbf{f}\left(\mathbf{s_L}\right) . \tag{6}$$

### 3.3.1 System Error Analysis

The system error (SE) emerges from the bidirectional mapping between discrete space $\mathcal{V}$ and continuous space $\mathcal{S}$, which inherently impacts prediction fidelity. A rigorous analysis of SE is essential for ensuring reliable and robust predictions in image space $\mathcal{V}$. We begin our theoretical analysis of SE with Assumption 3.2 and Theorem 3.3.

**Assumption 3.2.** After applying zero-score normalization, the continuous space follows a standard normal distribution:

$$\mathcal{S} \sim N(\mathbf{0}, \mathbf{I})$$

**Theorem 3.3** (System Error Upper Bound). *Given a tensor $\widehat{s} \in \mathcal{S} \subset \mathbb{R}^{c \times L}$, the system error defined as $\left\| f^{-1}\left(\mathbf{f}\left(\widehat{s}\right)\right) - \widehat{s} \right\|_1$ satisfies the following bound:*

$$SE := \mathbb{E}\left\| f^{-1}\left(\mathbf{f}\left(\widehat{s}\right)\right) - \widehat{s} \right\|_1 \leq g(h, MS)$$
$$= cL\left[ MS\left( \frac{1}{h}\left(\Phi(MS) - \Phi(-MS)\right) - 2 + 2\Phi(MS)\right) + \sqrt{\frac{2}{\pi}}e^{-\frac{MS^2}{2}} \right] \tag{7}$$

*where $\Phi$ denotes the cumulative distribution function of $N(\mathbf{0}, \mathbf{I})$.*

Denote $MS\left( \frac{1}{h}(\Phi(MS) - \Phi(-MS)) - 2 + 2\Phi(MS)\right) + \sqrt{\frac{2}{\pi}}e^{\frac{-MS^2}{2}}$ in Equation (7) as the upper bound of SE, whose convergence is guaranteed by Proposition 3.4.

**Proposition 3.4** (Asymptotic Convergence with $h$). *For any $\varepsilon > 0$, there exists $\delta > 0$ such that when $h \to +\infty$ and $MS \geq \delta$, the SE upper bound converges to zero:*

$$\lim_{h \to +\infty} \left| MS\left( \frac{1}{h}(\Phi(MS) - \Phi(-MS)) - 2 + 2\Phi(MS)\right) + \sqrt{\frac{2}{\pi}}e^{-\frac{MS^2}{2}} \right| = 0 \tag{8}$$

The Proposition 3.4 reveals that when we fix MS and increase the spatial resolution $h$, the upper bound $|g(h, MS)|$ of SE will reduce accordingly. On the other hand, when $h$ increases, the tensor sizes in $\mathcal{V}$ will increase exponentially, leading to higher computational costs. As such, the selection of $h$ must strike a balance between the accuracy of the estimation and the computational feasibility. Since the upper bound of SE decreases with an increase in $h$, it is generally preferable to choose the largest possible value of $h$ based on available computational resources, resulting in a fixed value of $h$ for a particular computational budget.

### 3.3.2 Theoretical Analysis of Optimal MS

*MS* determines the upper and lower limits of numerical truncation in the binary image-based time series metric space. Thus, it is necessary to conduct a detailed theoretical analysis of the selection of *MS*. Proposition 3.5 investigates how the upper bound of SE varies with a *MS* given a fixed value of $h$, which provides a theoretical guidance to choose the best MS under different computational budgets ($h$).

**Proposition 3.5** (Optimal MS Selection). *For fixed $h$, there exists a unique optimal threshold $MS^*$ minimizing the SE upper bound, characterized by:*

$$\frac{1}{h}\left(\Phi(MS^*) - \Phi(-MS^*)\right) - 2 + 2\Phi(MS^*) + \frac{MS^*}{h}\sqrt{\frac{2}{\pi}}e^{-\frac{MS^{*2}}{2}} = 0 \tag{9}$$

The fidelity of predictions in binary image space $\mathcal{V}$ heavily depends on the bidirectional mapping between discrete space $\mathcal{V}$ and continuous latent space $\mathcal{S}$. A key challenge arises from the SE, which quantifies the discrepancy between the original continuous representation and its reconstructed version after discretization. While Assumption 3.2 assumes $\mathcal{S} \sim N(0, \mathbf{I})$, real-world scenarios often exhibit larger variance in the latent space due to factors such as dataset shifts or model miscalibration. This motivates our analysis of SE under the generalized assumption $\mathcal{S} \sim N(0, k\mathbf{I})$, where $k > 1$ captures the variance scaling.

**Proposition 3.6** (Optimal Threshold under Variance Scaling). *Under the assumption $\mathcal{S} \sim N(0, k\mathbf{I})$ with $k > 1$, the optimal threshold $MS^*$ that minimizes the SE upper bound is characterized by the following condition:*

$$\frac{1}{h}\left(\Phi\left(\frac{MS^*}{\sqrt{k}}\right) - \Phi\left(-\frac{MS^*}{\sqrt{k}}\right)\right) - 2 + 2\Phi\left(\frac{MS^*}{\sqrt{k}}\right) + \frac{MS^*}{h}\sqrt{\frac{2}{\pi k}}e^{-\frac{(MS^*)^2}{2k}} = 0 \tag{10}$$

Here, $\Phi(\cdot)$ is the cumulative distribution function (CDF) of the standard normal distribution, $h$ is the spatial resolution, and $k$ is the variance scaling factor. This result generalizes Proposition 3.5 to scenarios where the latent space exhibits larger variability. In practice, it is challenging to find an analytic solution for Equation (10). Thus, the numerical method is employed to obtain solutions of Equation (10) in this work and the corresponding results are reported in Table 1.

As shown in Table 1, the numerically computed optimal *MS* increases monotonically with both the spatial resolution $h$ and the variance scaling factor $k$. This trend indicates that higher-resolution settings and larger latent variance require larger *MS* to satisfy Equation (10).

### 3.4 Theoretical Advantages of Visual Representation for Time Series Forecasting

Representing time series data visually, as explored by ViTime, is not merely an aesthetic or heuristic choice; it is fundamentally advantageous from a signal-processing standpoint. Specifically, transforming numerical signals into structured, image-like representations can significantly boost the effective signal-to-noise ratio (SNR), thereby enhancing forecasting robustness. To formally capture and quantify this advantage, we first establish conditions under which visual representation surpasses conventional numerical representation in terms of SNR. Subsequently, we explore image-based processing techniques to further amplify these benefits.

Table 1: Numerically Solved Optimal $MS^*$

|  |  | Optimal $MS^*$ |  |
| --- | --- | --- | --- |
| Resolution $h$ | $k = 1$ | $k = 1.5$ | $k = 2$ |
| 32 | 2.1 | 2.62 | 3.03 |
| 64 | 2.38 | 2.95 | 3.41 |
| 128 | 2.64 | 3.26 | 3.76 |
| 256 | 2.88 | 3.53 | 4.08 |
| 512 | 3.09 | 3.79 | 4.38 |

### 3.4.1 Visual Representation and SNR Enhancement.

Consider a noisy sinusoidal time series defined by:

$$s_k = A \sin(\omega_0 k + \phi) + \eta_k, \quad k = 0, \dots, L - 1,$$

where the signal amplitude $A > 0$, angular frequency $\omega_0 = 2\pi/P_{\text{period}}$, phase $\phi$, and Gaussian noise terms $\eta_k \sim N(0, \sigma^2)$ fully specify the system. Transforming this numerical series into a binary "stripe" image $v \in \{0,1\}^{h \times L}$ via quantization yields notable theoretical advantages. The binary representation is defined by:

$$v_{j,k} = \mathbf{1}\left(j = \left\lfloor \frac{s_k + \text{MS}}{\delta} \right\rfloor\right), \tag{11}$$

with quantization step $\delta = \Delta/h$ and total quantization range $\Delta = 2\text{MS}$. By comparing the SNR in numerical and visual domains, we obtain the following foundational result:

**Theorem 3.7** (Stripe SNR Boost). *Under mild assumptions that (i) the sinusoid amplitude spans at least one quantization bin ($\delta \leq A \leq \Delta - \delta$) and (ii) noise is small relative to quantization resolution ($\sigma < \delta/4$), the visual representation yields an SNR at the fundamental frequency $n_0 = \lfloor L/P_{\text{period}} \rfloor$ satisfying:*

$$SNR_{\text{vis}} \geq \frac{L}{4} \exp\left(\frac{\delta^2}{8\sigma^2}\right) \frac{\sigma^2}{A^2} SNR_{\text{num}}, \tag{12}$$

*where the numerical SNR is $SNR_{\text{num}} = A^2/(2\sigma^2)$.*

Theorem 3.7 provides clear quantitative conditions for visual superiority. Specifically, visual representation surpasses numerical representation ($\text{SNR}_{\text{vis}} > \text{SNR}_{\text{num}}$) whenever:

$$L > \frac{4A^2}{\sigma^2} \exp\left(-\frac{\delta^2}{8\sigma^2}\right). \tag{13}$$

Practically, this condition is typically met for moderate sequence lengths when the quantization step is comparable to or slightly larger than the noise standard deviation (e.g., $\delta \approx 2\sigma$). Under these realistic scenarios, the exponential term strongly favors visual representation, making it advantageous even at manageable $L$.

### 3.4.2 SNR Enhancement via Image Processing.

Although the theoretical advantage above is compelling, practical scenarios often involve considerable noise and subtle periodic signals. Furthermore, the binary quantization can introduce high-frequency artifacts that obscure signal patterns. To mitigate such undesirable effects and leverage the structured nature of visual representations, we propose employing image-processing operations, notably Gaussian blurring, to enhance signal fidelity further.

Applying a Gaussian blur along the image's quantization axis (the row or "value" dimension) effectively smooths quantization noise while preserving meaningful temporal structures. This simple convolutional operation yields significant amplification of the visual-domain SNR, formalized as follows:

**Theorem 3.8** (Gaussian Blur SNR Boost). *Under the conditions of Theorem 3.7, consider applying a one-dimensional Gaussian convolution kernel along the quantization dimension (rows) of the binary stripe image $v$:*

$$g_j = \frac{1}{Z} \exp\left(-\frac{j^2}{2\sigma_b^2}\right), \quad where \quad Z = \sum_j \exp\left(-\frac{j^2}{2\sigma_b^2}\right),$$

*to obtain the blurred image $w = g *_j v$. Denote the kernel's nuclear energy by $S = \sum_j g_j^2 \in (0,1)$, and define the visually blurred SNR at the fundamental frequency $n_0 = \lfloor L/P_{\text{period}} \rfloor$ as $SNR_{\text{vis}}^{blur}$. Then, the following lower bounds hold:*

$$SNR_{\text{vis}}^{blur} \geq \frac{L}{4S} \exp\left(\frac{\delta^2}{8\sigma^2}\right), \tag{14}$$

$$SNR_{\text{vis}}^{blur} \geq \frac{L\sigma^2}{2A^2 S} \exp\left(\frac{\delta^2}{8\sigma^2}\right) SNR_{\text{num}}, \tag{15}$$

*where the numerical-domain SNR is defined as $SNR_{\text{num}} = A^2/(2\sigma^2)$.*

Consequently, the blurred visual representation amplifies the numerical-domain SNR at least by a factor of:

$$\frac{\text{SNR}_{\text{vis}}^{\text{blur}}}{\text{SNR}_{\text{num}}} \geq \frac{L\sigma^2}{2A^2 S} \exp\left(\frac{\delta^2}{8\sigma^2}\right). \tag{16}$$

This result explicitly quantifies the advantage provided by Gaussian blurring in the visual representation. Notably, this amplification advantage scales linearly with the time series length $L$ and exponentially with the squared ratio of quantization step $\delta$ to noise standard deviation $\sigma$. Moreover, a smaller kernel nuclear energy $S$, corresponding to stronger blurring, yields a greater amplification of the visual-domain SNR relative to its numerical counterpart.

In practical implementations, the choice of Gaussian kernel parameters directly influences the nuclear energy $S$, and thus the SNR amplification factor. Typical examples include:

- $11 \times 11$ kernel ($\sigma_b = 2$): $S \approx 0.15$, providing substantial SNR amplification.

- $21 \times 21$ kernel ($\sigma_b = 4$): $S \approx 0.08$, approximately doubling the amplification compared to the previous case.

- $31 \times 31$ kernel ($\sigma_b = 6$): $S \approx 0.05$, further significantly enhancing the amplification factor.

In summary, even moderate Gaussian blurring substantially enhances the effective visual-domain SNR, enabling significantly improved signal discernibility and forecasting accuracy compared to traditional numerical-domain methods.

**Generalization to Complex Time Series.** While our theoretical analysis explicitly addresses a single sinusoidal component, its implications readily extend to realistic time series composed of multiple periodic components. Via linearity principles inherent in Fourier decomposition, observed visual-domain SNR advantages apply component-wise, amplifying structured periodic signals relative to unstructured and independent noise effects. Thus, real-world time series exhibiting intricate periodic behaviors benefit significantly from visual transformations and subsequent image-processing enhancements.

The rigorous theoretical results presented here establish a robust mathematical foundation for employing visual intelligence in time series analysis. Beyond aligning with human cognitive patterns, visual representations structurally amplify signal fidelity through inherent quantization and subsequent image processing techniques, such as Gaussian smoothing. Consequently, visual-domain methods provide a principled, theoretically justified route toward achieving more robust, reliable, and accurate time series forecasting, especially under challenging noise conditions.

Detailed proofs and supplementary details of the theorems presented in this section are provided in Supplementary Section C.

### 3.5 The Proposed ViTime Model

Figure 1 (b) presents the architecture of the ViTime network, which comprises three network modules: the Visual Time Tokenizer, the Decoder, and the Refining Module. The time series binary image is first fed into the Visual Time Tokenizer and outputs embedded latent representations. Next, the Decoder is developed to decode latent representations and produce initial prediction results in the image-value axis. To improve the generative quality of patch junctions, a Refining Module is designed to generate the final smooth prediction results.

**Visual Time Tokenizer.** The primary role of the Visual Time Tokenizer is to segment masked binary images into multiple patches and map these patches into the feature space. By leveraging the ViT (Dosovitskiy et al., 2020) architecture, the module captures spatial relationships between patches, thereby transforming temporal dependencies of the time series into spatial dependencies within the image space.

**Decoder.** The Decoder translates the tokenized patches back into the binary pixel metric space, providing an initial prediction where the ViT architecture is also adopted. In practice, the Decoder's prediction head applies a softmax along the height dimension $j$ to produce a probability tensor $\mathbf{p} \in \mathcal{P}$ (defined later in Section 3.6), which reduces to a one-hot vector along the height dimension $\mathbf{v} \in \mathcal{V}$ if the mass collapses to a single bin.

**Refining Module.** The transformer architecture in the Decoder can result in discontinuities at the patch junctions, which may affect the accuracy of the inverse mapping process. To address this issue, the Refining Module built with CNNs is employed. Initially, tokens decoded by Decoder are unpatched and fed into a CNN-based backbone. Next, the ASPP (Chen et al., 2015) module expands the model receptive field. Finally, the output is upsampled to the binary pixel metric space, generating the final image prediction result. The Refiner preserves the probabilistic semantics by operating on the logits (before softmax) or on probability maps to maintain consistency along the $j$-axis.

**Modeling process and masking.** The modeling process of ViTime is summarized as

$$\mathbf{v}'_{\mathbf{L}} = \text{ViTime}(\mathbf{v}_{\mathbf{L}} \odot \mathbf{M}_{\mathbf{L}}), \tag{17}$$

where $\odot$ is the element-wise product and $\mathbf{M}_{\mathbf{L}}$ is a temporal mask that zeros out the time steps to be forecast. Concretely, we use $\mathbf{M}_{\mathbf{L}} \in \{0, 1\}^{1 \times 1 \times L}$ (broadcast along $c$ and $h$) with

$$(\mathbf{M}_{\mathbf{L}})_{1,1,k} = \begin{cases} 1, & k \in \text{observed (context) time indices}, \\ 0, & k \in \text{forecast horizon (to be predicted)}. \end{cases}$$

Thus, for all $i \in [c], j \in [h], k \in [L]$,

$$(\mathbf{v}_{\mathbf{L}} \odot \mathbf{M}_{\mathbf{L}})_{i,j,k} = \begin{cases} \mathbf{v}_{i,j,k}, & \mathbf{M}_{1,1,k} = 1, \\ 0, & \mathbf{M}_{1,1,k} = 0, \end{cases}$$

i.e., the mask sets the to-be-predicted time positions to *all zeros across the $j$-axis*, removing any target information at those steps. Note that this masking operates on the input only. Although the masked columns are no longer one-hot (and hence leave $\mathcal{V}$ at those $k$), the network outputs valid distributions $\mathbf{p} \in \mathcal{P}$ over $j$ for every $(i, k)$ (see Section 3.6). During training, we sample masked spans to simulate forecasting; at inference, we set $\mathbf{M}_{\mathbf{L}}$ to zero precisely on the forecast horizon.

**Loss function.** The loss function employed in this study is defined as follows:

$$\mathcal{L} = d\left(\mathbf{v}'_{\mathbf{L}}, \mathbf{v}_{\mathbf{L}}\right) + \alpha \, \text{KLD}\left(\mathbf{v}'_{\mathbf{L}}, \mathbf{v}_{\mathbf{L}}\right), \tag{18}$$

where $d$ denotes the distance function defined in Equation (3), KLD denotes Kullback–Leibler divergence, and $\alpha$ is the hyperparameter balancing the two terms. The combined EMD and KLD loss addresses structural and probabilistic alignment along the $j$-axis. EMD minimizes spatial discrepancies in $\mathcal{V}/\mathcal{P}$, counteracting discretization-induced shift, while KLD refines distributional consistency to mitigate quantization artifacts.

This dual objective balances geometric fidelity (via EMD/Wasserstein-1 along $j$) and statistical accuracy (via KLD), which is crucial under the resolution-computation trade-off governed by $h$. In practice, to prevent information leakage and trivial identity mapping, both $d(\cdot, \cdot)$ and KLD are accumulated only over the masked time indices $\{k : \mathbf{M}_{1,1,k} = 0\}$, while the unmasked indices serve as conditioning context.

**From model outputs to forecasts.** The Decoder/Refiner produce a probability tensor $\mathbf{p} \in \mathcal{P}$ over the height $j$ at each $(i, k)$. For point forecasts, we apply the inverse mapping expectation (see Equation (5)) by replacing $v_{i,j,k}$ with $p_{i,j,k}$ to obtain $\mu_{i,k}$; this coincides with the estimator in Equation (5). For probabilistic forecasts, we retain $\mathbf{p}$ as a histogram distribution over bins and compute downstream summaries (quantiles/intervals) as detailed in Section 3.6.

### 3.6 Point and Probabilistic Forecasting in ViTime

Building on the binary image-based time series metric space in Section 3.3, ViTime treats forecasting as producing a distribution along the height (value) axis for each variate-time pair. This subsection details how ViTime yields probabilistic forecasts and how point forecasts are recovered as expectations under the same formulation.

**From one-hot images to probability tensors.** We relax the one-hot constraint $v_{i,j,k} \in \{0,1\}$ to a probability-simplex output $p_{i,j,k} \in [0, 1]$ with $\sum_{j=1}^{h} p_{i,j,k} = 1$. Define

$$\mathcal{P} \triangleq \left\{ p \in [0, 1]^{c \times h \times L} \ \middle| \ \sum_{j=1}^{h} p_{i,j,k} = 1, \ \forall i \in [c], k \in [L] \right\}, \tag{19}$$

so that $\mathcal{V} \subset \mathcal{P}$ and a one-hot tensor $\mathbf{v}$ is a degenerate case of $\mathbf{p} \in \mathcal{P}$. In practice, the prediction head of ViTime applies a softmax over the $j$-dimension to produce $\mathbf{p} \in \mathcal{P}$.

Let the bin width be $\Delta \triangleq 2\mathrm{MS}/h$, edges $b_0 = -\mathrm{MS}$, $b_j = -\mathrm{MS} + j\Delta$ for $j = 1, \ldots, h$, bins $B_j = [b_{j-1}, b_j)$, and centers $c_j = (j - 0.5)\Delta - \mathrm{MS}$.

**Probabilistic forecast (distributional output).** For each $(i, k)$, ViTime interprets the $h$-way probability vector $p_{i,1:h,k}$ as a histogram (mixture-of-uniforms) predictive distribution on $[-\mathrm{MS}, \mathrm{MS}]$ with

$$f_{i,k}(s) \triangleq \sum_{j=1}^{h} \frac{p_{i,j,k}}{\Delta} \mathbf{1}\{s \in B_j\}, \qquad F_{i,k}(s) \triangleq \sum_{m=1}^{j-1} p_{i,m,k} + p_{i,j,k} \frac{s - b_{j-1}}{\Delta}, \quad s \in B_j, \tag{20}$$

where $F_{i,k}(s) = 0$ for $s < -\mathrm{MS}$ and $F_{i,k}(s) = 1$ for $s \geq \mathrm{MS}$. This continuous relaxation preserves the geometry of the $j$-axis used by the EMD metric in Equation (3) and naturally supports uncertainty quantification.

**Point forecast as expectation.** Under Equation (20), the predictive mean recovers the inverse mapping in Equation (5) by replacing $v_{i,j,k}$ with $p_{i,j,k}$:

$$\mu_{i,k} \triangleq \mathbb{E}[S_{i,k}] = \sum_{j=1}^{h} c_j \, p_{i,j,k} = \sum_{j=1}^{h} \left( (j - 0.5) \frac{2\mathrm{MS}}{h} - \mathrm{MS} \right) p_{i,j,k}. \tag{21}$$

Thus, when only a point forecast is required, ViTime outputs $\mu_{i,k}$, which coincides with Equation (5). The predictive variance for uncertainty summaries is

$$\mathrm{Var}[S_{i,k}] = \sum_{j=1}^{h} p_{i,j,k} \left( (c_j - \mu_{i,k})^2 + \frac{\Delta^2}{12} \right). \tag{22}$$

**Quantiles and prediction intervals.** Define cumulative weights $C_{i,k}(j) \triangleq \sum_{m=1}^{j} p_{i,m,k}$ with $C_{i,k}(0) = 0$. For $\tau \in (0,1)$, let

$$J_\tau \triangleq \min \left\{ j \in [h] \mid C_{i,k}(j) \geq \tau \right\}, \qquad Q_{i,k}(\tau) = b_{J_\tau - 1} + \Delta \cdot \frac{\tau - C_{i,k}(J_\tau - 1)}{p_{i,J_\tau,k}}. \tag{23}$$

Then a central $(1 - \alpha)$ prediction interval is $[\, Q_{i,k}(\alpha/2), \ Q_{i,k}(1 - \alpha/2) \,]$.

### 3.7 Evaluation Metrics

Existing numerical fitting-based TSF foundation models, e.g., TimesFM, are typically pretrained on comprehensive real-world datasets. While the specific nomenclature of the testing set may not be explicitly listed in the training data, there is a possibility that the real-world dataset encompasses similar data sources, potentially leading to issues of test set leakage. To address this concern and ensure a more rigorous and equitable experimental comparison, we propose novel metrics for **zero-shot evaluation**. For point forecasts, we introduce the Rescale-Mean Absolute Error (ReMAE) and Rescale-Mean Squared Error (ReMSE). To evaluate probabilistic forecasts, we extend this approach with the Rescale-Continuous Ranked Probability Score (ReCRPS).

The fundamental principle underlying these metrics involves rescaling the test dataset across various time resolutions, as illustrated in Equation (24). The time series interpolation (TSI) method is employed to rescale the original test time series of length T to $\beta T$:

$$S_{\beta T} = TSI\left(S_T, \text{rescaling factor} = \beta\right). \tag{24}$$

For point forecasts, the formulas for ReMAE and ReMSE are based on the standard Mean Absolute Error (MAE) and Mean Squared Error (MSE):

$$ReMSE = \frac{\sum_{\beta \in \mathbf{U}} MSE\left(S'_{\beta T}, S_{\beta T}\right)}{len(\mathbf{U})} \tag{25}$$

$$ReMAE = \frac{\sum_{\beta \in \mathbf{U}} MAE\left(S'_{\beta T}, S_{\beta T}\right)}{len(\mathbf{U})} \tag{26}$$

For probabilistic forecasts, the Continuous Ranked Probability Score (CRPS) is considered, which generalizes the MAE by comparing the entire predictive distribution with the ground truth. The CRPS is defined as:

$$\text{CRPS}(F, y) = \int_{-\infty}^{\infty} \left(F(x) - \mathbb{1}_{\{x \geq y\}}\right)^2 dx \tag{27}$$

where $F$ is the predicted cumulative distribution function (CDF) and $y$ is the observed value, with $\mathbb{1}$ being the Heaviside step function. Following the same rescaling principle, we define ReCRPS as:

$$ReCRPS = \frac{\sum_{\beta \in \mathbf{U}} CRPS\left(F_{\beta T}, S_{\beta T}\right)}{len(\mathbf{U})} \tag{28}$$

In Equation (24)-Equation (28), $S'_{\beta T}$ represents the point prediction, $F_{\beta T}$ is the predicted distribution for the rescaled series, $S_{\beta T}$ is the rescaled ground truth, and $\mathbf{U}$ is the set of scaling factors:

$$\mathbf{U} = [0.5, \ 0.66, 1, 1.5, 2]. \tag{29}$$

The proposed ReMSE, ReMAE, and ReCRPS metrics address a critical challenge in evaluating time series foundation models: mitigating test set leakage caused by overlapping data distributions between training and

testing phases. By rescaling the test set across multiple resolutions ($\beta \in \mathbf{U}$) via time series interpolation (TSI, Equation (24)), these metrics introduce synthetic scale variations that disrupt exact temporal patterns, thereby reducing the risk of evaluating models on memorized or overfitted data. This approach ensures a leakage-resistant evaluation framework, as models must generalize to unseen scales rather than relying on spurious correlations learned from the training set.

A key implication of this work is the necessity of scale-agnostic evaluation in time series forecasting. Traditional single-scale metrics like MSE, MAE, and CRPS risk conflating memorization with true generalization, particularly when training data encompasses diverse real-world sources. By averaging errors across $\beta$, our rescaled metrics incentivize models to capture invariant temporal structures–such as periodicity, trends, and noise resilience–that persist across resolutions. This applies to both the accuracy of point forecasts and the calibration of predicted uncertainty. This aligns with recent theoretical insights in self-supervised learning, where augmentation-induced invariance improves out-of-distribution robustness (Yao et al., 2022). It is worth noting that in the fine-tuning study, i.e., Section 4.4, in order to ensure the consistency of the distribution between the test data and the fine-tuning data, we still adopt the traditional MSE/MAE evaluation metrics.

## 4 Computational Experiments

### 4.1 Experimental Configuration

**Datasets**

Seven popular publicly accessible datasets: Electricity, Traffic, Weather, ETTh1, ETTh2, ETTm1, and ETTm2 (Wu et al., 2021) are employed in computational experiments to validate the effectiveness of the proposed ViTime.

**Model setup**

The ViTime model is developed using data sequences synthesized by RealTS. During each training epoch, 20,000 sequences are randomly generated. After training, zero-shot testing and fine-tuning are implemented accordingly. For multivariate time series, a channel-independent strategy (Nie et al., 2022) is applied, predicting each variable separately before combining them to form the final multivariate forecast.

The default parameters for the ViTime model are set as follows: $h = 128$, $MS = 3.5$, maximum lookback window $T = 512$, and maximum prediction length $l = 720$. For a fair comparison, all considered models employ a lookback length of 512 to forecast future sequences of lengths $96, 192, 336, 720$. Additionally, we adopt the Adam optimizer (Kingma, 2014) with a learning rate of 2x10-4 during the training process. More details on training are available in Supplementary Section B.

To further enhance temporal resolution and information density practically, input sequences are initially interpolated to twice their original length (2L) and the prediction results are interpolated back to the original length. This interpolation increases temporal granularity, facilitating more precise pattern extraction. Furthermore, Gaussian blurring with kernel size of 31 applied to the binary images before processing by ViTime significantly reduces sparsity and increases local information density, thereby reinforcing the theoretical advantages outlined in Section 3.4.1.

### 4.2 Comparison of ViTime to SOTA TSF Benchmarks Under Zero-shot Point Forecasting Setting

For zero-shot performance comparison, we consider four variants: (1) ViTime - our proposed TSF foundation model, trained on generative data from RealTS and adopting a zero-shot paradigm; (2) ViTime-TFM - a variant of ViTime, which is trained on the same public available dataset as TimesFM. (See Supplementary Section B.3 for more information.) (3) PatchTST-ZS - trained on the same RealTS-generated data as ViTime, using a numerical fitting paradigm to create a zero-shot version of PatchTST. (4) Moirai (Woo et al., 2024), Moment (Goswami et al., 2024), VisionTS (Chen et al., 2024) and TimesFM (Das et al., 2024) - powerful TSF foundation model pre-trained on extensive real-world datasets. All models employ a lookback length of 512 to ensure a fair comparison. Details of benchmark model configurations are reported in Supplementary Section B.4.

(a) Experimental Results With Metrics of MSE and MAE

| Model | ETTh1 | | ETTh2 | | ETTm1 | | ETTm2 | | Electricity | | Traffic | | Weather | |
|---|---|---|---|---|---|---|---|---|---|---|---|---|---|---|
| | MSE | MAE | MSE | MAE | MSE | MAE | MSE | MAE | MSE | MAE | MSE | MAE | MSE | MAE |
| *Numerical Models* | | | | | | | | | | | | | | |
| Moirai | 0.434 | 0.439 | 0.346 | 0.382 | 0.382 | 0.388 | 0.272 | 0.321 | 0.188 | 0.274 | 1.779 | 0.766 | 0.238 | 0.261 |
| Moment | 0.691 | 0.585 | 0.341 | 0.350 | 0.845 | 0.580 | **0.257** | 0.317 | 0.837 | 0.763 | 1.375 | 0.788 | 0.348 | 0.429 |
| VisionTS | **0.390** | 0.414 | 0.333 | 0.375 | **0.374** | **0.372** | 0.282 | 0.321 | 0.207 | 0.294 | 0.443 | 0.284 | 0.269 | 0.292 |
| TimesFM | 0.442 | 0.430 | 0.356 | 0.389 | 0.424 | 0.419 | 0.328 | 0.347 | 0.151 | 0.245 | 0.369 | 0.245 | 0.229 | 0.255 |
| PatchTST-ZS | 1.237 | 0.831 | 0.903 | 0.710 | 1.356 | 0.825 | 0.839 | 0.622 | 1.311 | 0.885 | 1.873 | 0.945 | 0.907 | 0.588 |
| *Vision-Assisted Models* | | | | | | | | | | | | | | |
| ViTime-TFM | 0.398 | **0.387** | 0.321 | 0.350 | 0.382 | 0.377 | 0.295 | **0.312** | **0.136** | **0.221** | **0.332** | **0.221** | **0.206** | **0.229** |
| ViTime | 0.545 | 0.449 | **0.284** | **0.344** | 0.409 | 0.398 | 0.302 | 0.341 | 0.196 | 0.280 | 0.730 | 0.386 | 0.286 | 0.289 |

(b) Experimental Results With Metrics of ReMSE and ReMAE

| Model | ETTh1 | | ETTh2 | | ETTm1 | | ETTm2 | | Electricity | | Traffic | | Weather | |
|---|---|---|---|---|---|---|---|---|---|---|---|---|---|---|
| | ReMSE | ReMAE | ReMSE | ReMAE | ReMSE | ReMAE | ReMSE | ReMAE | ReMSE | ReMAE | ReMSE | ReMAE | ReMSE | ReMAE |
| *Numerical Models* | | | | | | | | | | | | | | |
| Moirai | 1.144 | 0.722 | 0.754 | 0.467 | 1.448 | 0.849 | 0.455 | 0.397 | 0.859 | 0.676 | 1.416 | 0.894 | 0.706 | 0.414 |
| Moment | 1.089 | 1.240 | 0.498 | **0.321** | 0.894 | 0.618 | 0.542 | 0.582 | 0.907 | 0.743 | 1.138 | 0.69 | 0.545 | 0.349 |
| VisionTS | 0.988 | 1.016 | 0.524 | 0.350 | 0.873 | 0.559 | 0.773 | 0.516 | 0.851 | 0.669 | 1.173 | 0.669 | 0.519 | 0.327 |
| TimesFM | 0.490 | 0.467 | 0.374 | 0.396 | 0.671 | 0.503 | 0.355 | 0.359 | 0.367 | 0.404 | 0.744 | 0.519 | 0.284 | 0.306 |
| PatchTST-ZS | 1.477 | 0.903 | 1.097 | 0.775 | 1.295 | 0.798 | 0.805 | 0.613 | 1.414 | 0.921 | 2.054 | 1.002 | 0.911 | 0.584 |
| *Vision-Assisted Models* | | | | | | | | | | | | | | |
| ViTime-TFM | 0.481 | 0.451 | 0.314 | 0.354 | 0.519 | 0.455 | 0.276 | 0.325 | 0.301 | 0.350 | **0.718** | 0.460 | 0.237 | 0.261 |
| ViTime | **0.457** | **0.431** | **0.29** | 0.346 | **0.473** | **0.420** | **0.237** | **0.301** | **0.225** | **0.308** | 0.730 | **0.400** | **0.203** | **0.228** |

Table 2: Overall Experimental Results Comparison

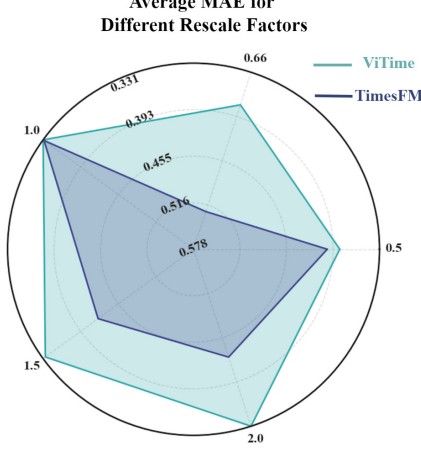

Figure 2: Radar plots comparing the average MAE of ViTime and TimesFM across different rescale factors. The radial axis represents MAE, with lower values (larger radius) indicating better performance. Each axis corresponds to a specific rescale factor.

Table 2 summarizes the zero-shot performance of all models using traditional metrics (MSE, MAE) and our proposed scale-invariant metrics (ReMSE, ReMAE). As shown in Table 2a, our vision-assisted models demonstrate highly competitive performance. ViTime achieves the best results on the ETTh2, ETTm2, and Weather datasets, while its variant, ViTime-TFM, secures the top performance on Electricity and Traffic datasets. Notably, ViTime-TFM, which shares the same training data as TimesFM, consistently outperforms it on most datasets, underscoring the inherent advantages of our vision-based modeling approach.

The superiority of ViTime becomes even more pronounced when evaluated with scale-invariant metrics, as shown in Table 2b. ViTime demonstrates remarkable dominance by achieving the best ReMSE or ReMAE on 11 out of 14 evaluation settings. This highlights its robust generalization ability across different temporal resolutions in zero-shot scenarios. Furthermore, ViTime significantly outperforms PatchTST-ZS across all datasets and metrics, confirming the effectiveness of visual intelligence strategies over numerical fitting for

zero-shot forecasting. The strong performance of ViTime on ReMSE and ReMAE, compared to the still-strong but less consistent performance of ViTime-TFM, suggests that the synthetic training data from RealTS is crucial for enhancing zero-shot generalization across varying temporal scales.

To further assess robustness, Figure 2 presents the performance across different rescaling factors. TimesFM exhibits optimal accuracy only at the original scale ($\beta = 1$), suffering significant degradation when evaluated at other scales. Such behavior indicates sensitivity to scale-specific patterns and suggests potential data leakage from the original resolution. In contrast, ViTime maintains consistently robust forecasting performance across all rescaling factors, as evidenced by stable ReMSE and ReMAE metrics. This illustrates ViTime's ability to learn intrinsic temporal relationships independent of specific time resolutions, further reinforcing the robustness and generalization benefits of vision-based modeling trained on RealTS data.

**Large-scale benchmark note.** A large-scale zero-shot evaluation on the community GIFT-EVAL benchmark (Aksu et al., 2024) is provided in Supplementary Section D.2.

### 4.3 Comparison of ViTime to SOTA TSF Benchmarks Under Zero-shot Probabilistic Forecasting Settings

Table 3: Comparison of probabilistic forecasting performance. Within each scenario, the best results (lower is better) for each dataset are **bolded**.

| Method | ETTh1 | ETTh2 | ETTm1 | ETTm2 | Electricity | Traffic | Weather |
|---|---|---|---|---|---|---|---|
| *CRPS* | | | | | | | |
| Moirai | 0.506 | **0.274** | 0.538 | 0.309 | 0.522 | 0.626 | 0.506 |
| Lag-Llama | 0.441 | 0.401 | 0.435 | 0.432 | 0.470 | 0.548 | 0.394 |
| ViTime | **0.356** | 0.319 | **0.344** | **0.286** | **0.267** | **0.327** | **0.244** |
| *ReCRPS* | | | | | | | |
| Moirai | 0.537 | 0.382 | 0.631 | 0.329 | 0.609 | 0.684 | 0.620 |
| Lag-Llama | 0.478 | 0.442 | 0.463 | 0.420 | 0.514 | 0.629 | 0.377 |
| ViTime | **0.358** | **0.318** | **0.346** | **0.283** | **0.266** | **0.324** | **0.241** |

For zero-shot probabilistic forecasting, we evaluate three variants: (1) ViTime - our vision-assisted TSF foundation model trained on generative data from RealTS and deployed in a zero-shot paradigm; (2) Moirai (Woo et al., 2024) - a strong TSF foundation model pretrained on large-scale real data; and (3) Lag-Llama (Rasul et al., 2023) - an LLM-based probabilistic forecaster. All models adopt a lookback length of 512 to ensure a fair comparison. We report both CRPS and our rescaling-invariant metric, ReCRPS.

As summarized in Table 3, ViTime delivers state-of-the-art zero-shot probabilistic performance. Under the standard CRPS metric, ViTime achieves the best results on 6 out of 7 datasets, with particularly substantial gains on large multivariate benchmarks. For instance, compared to the strongest baseline, ViTime achieves relative CRPS reductions of approximately 43% on Electricity, 40% on Traffic, and 38% on Weather. On the ETTh2 dataset, while Moirai attains a slightly lower CRPS, this narrow advantage is reversed when scale effects are controlled for. Using the ReCRPS metric, ViTime achieves the best performance on all 7 datasets, demonstrating superior distributional calibration across scales. These results indicate that ViTime not only produces sharper forecasts but also maintains this accuracy and calibration robustness across diverse temporal resolutions, establishing it as a highly reliable zero-shot probabilistic forecaster.

Overall, ViTime emerges as a robust, accurate, and reliable zero-shot time series forecasting model. The effectiveness of ViTime stems from two key innovations: vision-assisted modeling and synthetic training data generated by RealTS. Together, these features enable ViTime to generalize effectively across heterogeneous datasets and varying temporal scales. The strengths of ViTime are evident in both point and probabilistic forecasting settings. For point forecasts, ViTime delivers strong performance across diverse applications. For probabilistic forecasts, ViTime demonstrates exceptional qualities—it maintains scale-robust calibration and consistently achieves lower CRPS and ReCRPS scores across different datasets and resolutions. These comprehensive results establish ViTime as a dependable zero-shot forecaster that excels in both point estimation and distributional prediction tasks.

## 4.4 Comparison of ViTime to SOTA TSF Benchmarks Under Fine-tuning Settings

Table 4: Comparisons of Fine-tuning forecasting results with MAE. FT is short for fine-tuning. The best MAE results are **bolded**, and the second best are underlined. Standard deviations shown in the second row for ViTime.

| Method | Data proportion | ETTh1 | ETTh2 | ETTm1 | ETTm2 | Electricity | Traffic | Weather |
|---|---|---|---|---|---|---|---|---|
| TimesFM (FT) | 10% | 0.426 | 0.410 | 0.388 | 0.334 | - | - | - |
| GPT4TS (FT) | 10% | 0.542 | 0.431 | 0.466 | 0.343 | Not Reported | | |
| TIME-LLM (FT) | 10% | 0.522 | 0.394 | 0.426 | 0.323 | - | - | - |
| **ViTime (FT)** | 10% | 0.422 (±0.034) | 0.370 (±0.007) | 0.376 (±0.003) | 0.312 (±0.008) | 0.250 (±0.008) | 0.251 (±0.008) | 0.252 (±0.005) |
| PatchTST | 10% | 0.542 | 0.431 | 0.466 | 0.343 | 0.268 | 0.286 | 0.283 |
| PatchTST | 100% | 0.434 | 0.381 | 0.382 | 0.317 | 0.253 | 0.264 | 0.264 |
| SiMBA | 100% | 0.433 | 0.392 | 0.396 | 0.328 | 0.274 | 0.291 | 0.281 |
| TIMESNET | 100% | 0.450 | 0.427 | 0.406 | 0.333 | 0.295 | 0.336 | 0.286 |
| iTransformer | 100% | 0.448 | 0.407 | 0.410 | 0.332 | 0.270 | 0.282 | 0.278 |
| TimeMixer | 100% | 0.423 | 0.384 | 0.376 | 0.316 | 0.246 | 0.263 | 0.262 |
| **ViTime (FT)** | 100% | **0.406** (±0.039) | **0.344** (±0.004) | **0.366** (±0.003) | **0.297** (±0.017) | **0.245** (±0.004) | **0.248** (±0.005) | **0.249** (±0.004) |

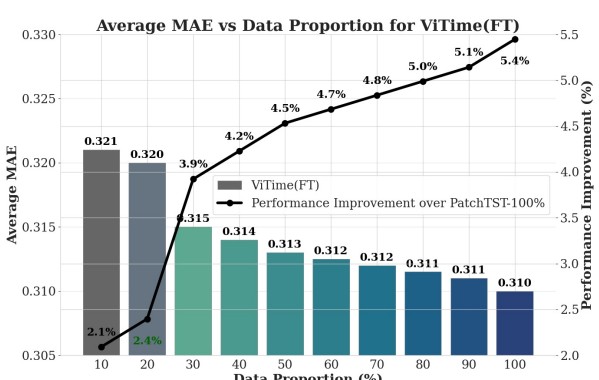

Figure 3: Performance with different fine-tuning data proportion.

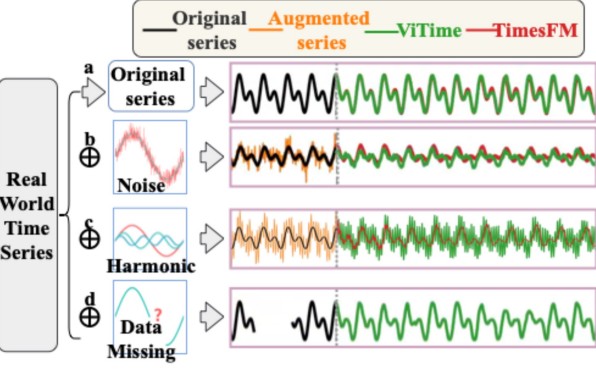

Figure 4: Performance comparison of ViTime versus TimesFM on TSF tasks under various data perturbations: a. Original time series. b. Time series with noises injected. c. Time series with harmonic added. d. Time series with missing data.

While zero-shot results demonstrate the predictive capability of ViTime on unseen data, some high-precision TSF tasks might require further fine-tuning studies to enhance prediction accuracy. Thus, this section focuses on fine-tuning studies across various specialized datasets.

To comprehensively evaluate the fine-tuning performance of ViTime, we compare ViTime with other foundation models and SOTA supervised TSF models. Foundational models including TimesFM (Das et al., 2024), GPT4TS (Zhou et al., 2023a), and TIME-LLM (Jin et al., 2023) are fine-tuned using 10% of the training data. Recent SOTA-supervised TSF models such as SiMBA (Patro & Agneeswaran, 2024), TIMESNET (Wu et al., 2022), iTransformer (Liu et al., 2023), TimeMixer (Wang et al., 2024) and PatchTST (Nie et al., 2022) use 100% of the training data, as reported in their respective papers. We also fine-tune ViTime using between 10% to 100% of the training data to provide a comprehensive comparison.

Results of the fine-tuning study are provided in Table 4. ViTime fine-tuned with 10% of the training data can outperform other foundational models and the latest supervised models updated on 100% of the training data. Furthermore, as shown in Figure 3, when the fine-tuning data proportion approaches 100%, the prediction accuracy of ViTime gradually increases and significantly surpasses all existing models, which suggests that ViTime excels in both low-data-availability environments (10% fine-tuning) and full-data-availability scenarios

(100% fine-tuning), consistently outperforming both other foundation models and specialized supervised models.

## 4.5 Robust Inference and Generalizability Analysis

Table 5: Comparison of average ReMAE forecasting results.

| Method | ETTh1 | ETTh2 | ETTm1 | ETTm2 | Electricity | Traffic | Weather |
|---|---|---|---|---|---|---|---|
| *GN standard deviations = 0.1* | | | | | | | |
| TimesFM | 0.471±0.002 | 0.393±0.001 | 0.495±0.012 | 0.352±0.005 | 0.403±0.004 | 0.512±0.016 | 0.280±0.005 |
| ViTime | **0.449±0.002** | **0.360±0.001** | **0.429±0.001** | **0.329±0.002** | **0.340±0.006** | **0.402±0.014** | **0.279±0.007** |
| *GN standard deviations = 0.3* | | | | | | | |
| TimesFM | 0.473±0.007 | 0.390±0.002 | 0.485±0.010 | 0.344±0.004 | 0.414±0.006 | 0.518±0.011 | **0.288±0.005** |
| ViTime | **0.455±0.005** | **0.363±0.001** | **0.434±0.002** | **0.333±0.003** | **0.355±0.007** | **0.426±0.016** | 0.288±0.009 |
| *GN standard deviations = 0.5* | | | | | | | |
| TimesFM | 0.479±0.006 | 0.392±0.003 | 0.488±0.009 | 0.345±0.005 | 0.433±0.005 | 0.529±0.012 | 0.295±0.005 |
| ViTime | **0.466±0.003** | **0.370±0.002** | **0.445±0.003** | **0.337±0.003** | **0.371±0.007** | **0.461±0.013** | **0.294±0.013** |
| *GN standard deviations = 0.7* | | | | | | | |
| TimesFM | **0.483±0.009** | 0.394±0.003 | 0.492±0.006 | 0.349±0.005 | 0.450±0.005 | 0.543±0.015 | 0.302±0.005 |
| ViTime | 0.484±0.004 | **0.394±0.003** | **0.443±0.003** | **0.346±0.005** | **0.377±0.006** | **0.510±0.021** | **0.301±0.014** |
| *GN standard deviations = 1.0* | | | | | | | |
| TimesFM | **0.487±0.013** | **0.399±0.003** | 0.500±0.009 | 0.359±0.006 | 0.475±0.006 | 0.567±0.007 | 0.312±0.007 |
| ViTime | 0.487±0.006 | 0.408±0.005 | **0.471±0.004** | **0.358±0.005** | **0.415±0.010** | **0.546±0.022** | **0.305±0.010** |
| *DM P = 0.3* | | | | | | | |
| ViTime | **0.453±0.002** | **0.378±0.001** | **0.432±0.001** | **0.337±0.003** | **0.343±0.006** | **0.417±0.014** | **0.281±0.013** |

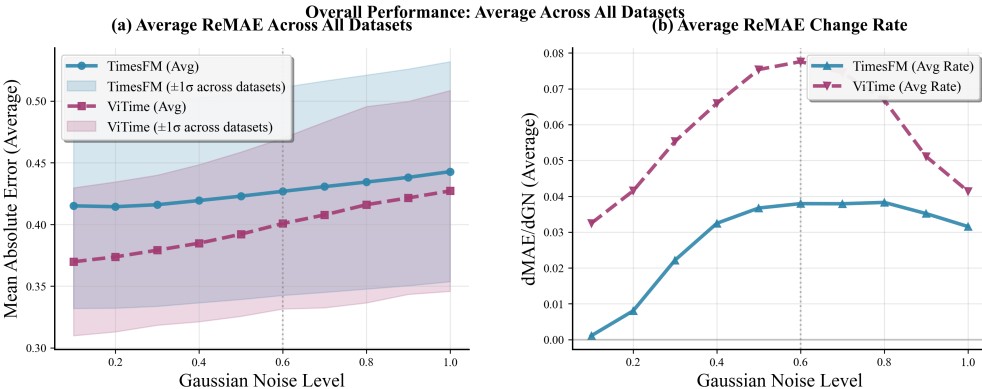

Figure 5: Robustness analysis under increasing Gaussian noise levels.

To rigorously assess the robustness and generalizability of ViTime, we conducted comprehensive zero-shot experiments comparing its performance against TimesFM under various data perturbation scenarios, including original time series, Gaussian noise (GN), harmonic augmentation, and missing data (DM). These scenarios represent challenges often encountered in practical forecasting tasks, evaluating the capabilities of models in maintaining predictive accuracy amidst compromised data quality.

In our analysis of robustness against noise, we consider two complementary dimensions:

1. **Absolute Robustness**, defined as the ability of a model to maintain a superior absolute performance (i.e., lower prediction error) across varying levels of noise.

2. **Relative Robustness**, which refers to the rate of performance degradation as the noise intensity increases. A model with higher relative robustness exhibits a smaller increase in error for a given increase in noise.

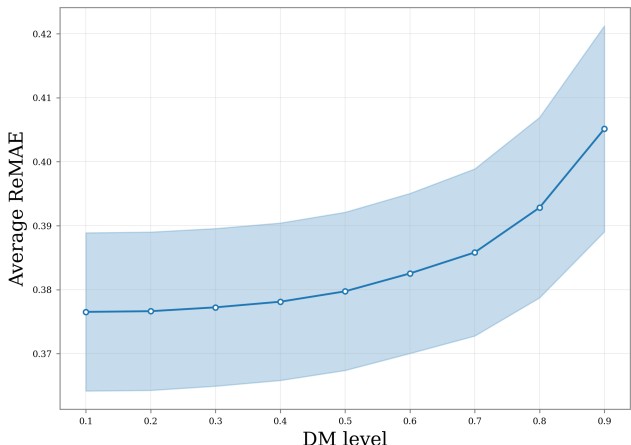

Figure 6: Average ReMAE of ViTime across different DM rates.

Our extended experimental results as depicted in Figure 5 and detailed in Table 5, provide a nuanced view of the behavior of models. In terms of **absolute robustness**, Figure 5 (a) clearly shows that the ReMAE of ViTime is consistently and significantly lower than that of TimesFM across all tested Gaussian noise levels, from a standard deviation of 0.1 to 1.0. This demonstrates that ViTime reliably delivers more accurate predictions in both low- and high-noise environments. We argue that this sustained performance advantage is a critical aspect of robustness for real-world applications, where the primary goal is to achieve the highest possible accuracy under given conditions.

Regarding **relative robustness**, the analysis of the performance degradation rate as shown in Figure 5 (b), confirms the observation that the performance of ViTime is more sensitive to increasing noise. The slope of ViTime's ReMAE curve is generally steeper than that of TimesFM, indicating a lower relative robustness. We posit that this is an expected phenomenon; as noise approaches an infinite magnitude, the predictive power of any model should diminish, and their error rates will converge towards a high value determined by the inherent scale of data.

In summary, while ViTime is more sensitive to increases in noise (lower relative robustness), its foundational performance is so strong that its absolute prediction accuracy remains superior to TimesFM across the entire spectrum of tested noise levels. For instance, as shown in Table 5, even at a high noise level of 0.7, ViTime outperforms or matches TimesFM on all datasets. In practice, an end-user is more concerned with which model provides a more reliable result (lower ReMAE) in a given noisy environment, rather than which model's performance curve is flatter. Therefore, ViTime's exceptional **absolute robustness** makes it a more dependable and effective choice for forecasting in the presence of noise.

The most distinctive performance disparity emerges under the scenario of data missing (DM). As shown for the DM P=0.3 case in Table 5, ViTime decisively outperforms the baseline across all tested datasets. TimesFM, being reliant on numerical fitting, would necessitate explicit imputation strategies to handle such data gaps. Conversely, ViTime robustly accommodates missing values by interpreting them as zero-valued pixels within its visual representations. Consequently, ViTime leverages spatial dependencies among available data points effectively, maintaining high prediction accuracy even amidst substantial data sparsity.

To further validate the robustness of ViTime to varying degrees of missing data, we systematically evaluate its forecasting accuracy across data missing ratios ranging from 10% to 90% (Figure 6). Results reveal that ViTime sustains remarkable forecasting performance with minimal degradation until data missingness surpasses 50%, underscoring its exceptional resilience to incomplete data scenarios.

Collectively, these extensive evaluations substantiate the superior robustness of ViTime and generalizability compared to traditional numerical fitting-based methods. Its inherent capability to effectively mitigate perturbations through visual representation learning positions it as a highly promising approach for real-world forecasting applications, where consistent data quality cannot always be guaranteed.

Table 6: Empirical Forecasting Performance under Different MS Values

| | MS | | | | | | |
|---|---|---|---|---|---|---|---|
| | 2.38 | 2.64 | 2.88 | 3.09 | 3.50 | 5.00 | 6.00 |
| ReMSE | 0.4423 | 0.4404 | 0.4400 | 0.4348 | **0.4178** | 0.4780 | 0.4724 |
| ReMAE | 0.3818 | 0.3812 | 0.3811 | 0.3788 | **0.3759** | 0.3990 | 0.3959 |

## 4.6 Ablation Study

### 4.6.1 Ablation of MS

Proposition 3.6 establishes the theoretical relationship between the optimal MS threshold and the variance scaling factor $k$ in the latent space. For stationary data ($\mathcal{S} \sim N(0, \mathbf{I})$, i.e., $k = 1$), Proposition 3.6 reveals that with $h = 128$, the optimal MS should be 2.64. However, real-world time series often exhibit non-stationary characteristics. Our pre-analysis of the target variable's variance after input-based standardization (see Supplementary Section B.2) demonstrates that the effective $k$ value for the prediction horizon falls within $[1.5, 2]$ across all benchmark datasets.

Table 1 provides numerically solved optimal $MS^*$ values under different $k$ and $h$ configurations. For $h = 128$ (our experimental setting) and $k \in [1.5, 2]$, the theoretical optimal MS ranges between 3.26-3.76. This motivates our selection of $MS = 3.5$ as a balanced configuration within this interval.

To validate this choice, Table 6 presents the average relative ReMSE and ReMAE across six benchmark datasets under zero-shot setting. The results demonstrate that $MS = 3.5$ achieves the minimum forecasting error, reducing ReMSE by 4.1% and ReMAE by 1.8% compared to the stationary optimal $MS = 2.64$. This strong alignment between theoretical predictions (Table 1) and empirical performance (Table 6) confirms that our MS selection strategy effectively minimizes system error while accommodating real-world data characteristics.

### 4.6.2 Ablation of Loss Function

Table 7: Ablation study of loss function components on prediction performance.

| Metric | Loss Configuration | | |
|---|---|---|---|
| | EMD Only | JSD+EMD (Ours) | JSD Only |
| Average ReMAE | 0.3941 | **0.3759** | 0.3956 |
| Average ReMSE | 0.4586 | **0.4178** | 0.4637 |

In this section, we conducted ablation studies on the loss function components of ViTime under zero-shot setting. Table 7 compares model performance under three configurations: (1) EMD alone, (2) our proposed loss function in Equation (18),where $\alpha = 0.2$ to balance the quantity level, and (3) JSD alone. The results demonstrate that our dual-objective loss achieves optimal performance on both ReMSE and ReMAE.

### 4.6.3 Ablation of Other Configuration

In this section, we perform several ablation studies to gain deeper insights into ViTime model configuration. The results are reported in Figure 7. Figure 7 a depicts the influence of varying spatial resolutions ($h$) on model accuracy. Although increasing $h$ slightly improves the prediction results, the associated computational cost increases exponentially. Thus, setting $h$ to 128 is more economical and efficient. Figure 7 b illustrates the effect of different lookback window lengths ($T$) on prediction accuracy. It is evident that a longer lookback window length significantly enhances the model's prediction accuracy. Figure 7 c reports the prediction accuracy across different model sizes. The data shows that models with more parameters tend to perform better. Moreover, the proposed ViTime achieves superior performance with only **93M** parameters compared

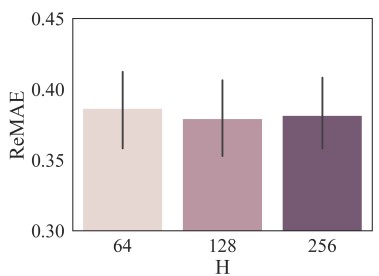
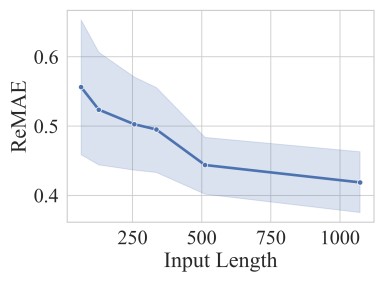
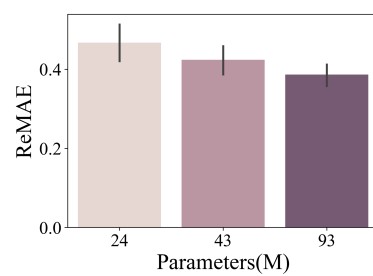

(a) Forecasting performance varying with h

(b) Forecasting performance varying with the length of lookback window

(c) Forecasting performance across different model sizes

Figure 7: Ablation studies with zero-shot forecasting.

Note: The model size of ViTime used in computational experiments is 93M parameters version.

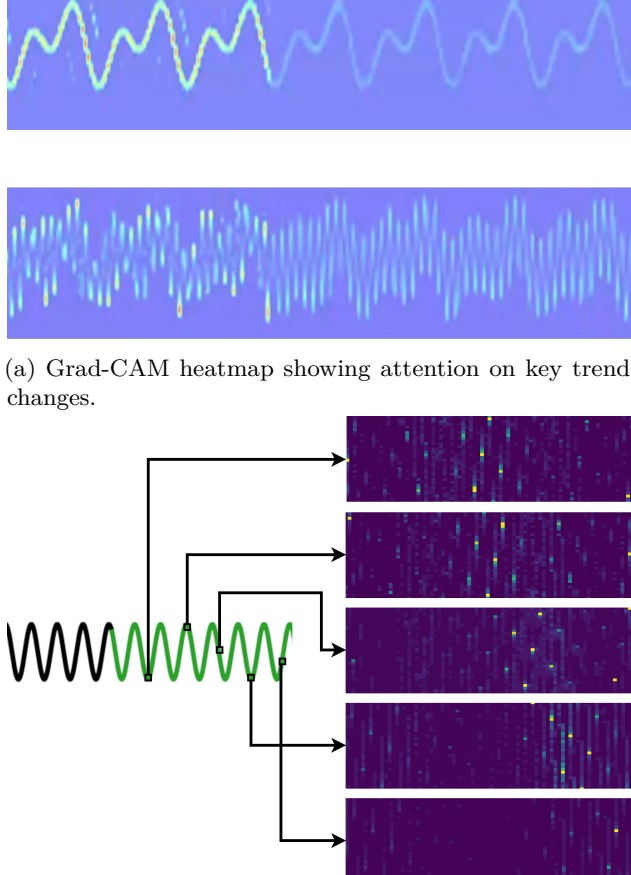

(a) Grad-CAM heatmap showing attention on key trend changes.

(b) Attention maps at different prediction positions demonstrating temporal dependencies.

Figure 8: Visualization of ViTime's attention mechanism. Despite not using an autoregressive paradigm, ViTime exhibits sequential processing patterns through its multi-layer self-attention modules.

with TimesFM, which is over 200M parameters, further demonstrating the efficiency and effectiveness of ViTime.

## 4.7 Interpretation of ViTime

Figure 8 illustrates the attention mechanism of ViTime through Grad-CAM (Selvaraju et al., 2017) heatmaps and position-specific attention maps. The Grad-CAM results demonstrate that ViTime focuses strongly on periods of fundamental trend changes. Further analysis through attention maps at different prediction positions reveals an interesting pattern: despite not adopting an autoregressive paradigm, ViTime's multi-layer self-attention modules process information in a temporal sequence. The input data and the predicted results from previous time steps determine the spatiotemporal distribution of predictions at each time step. This aligns with human cognitive patterns, where information is processed from the recent to the distant past while maintaining awareness of known information.

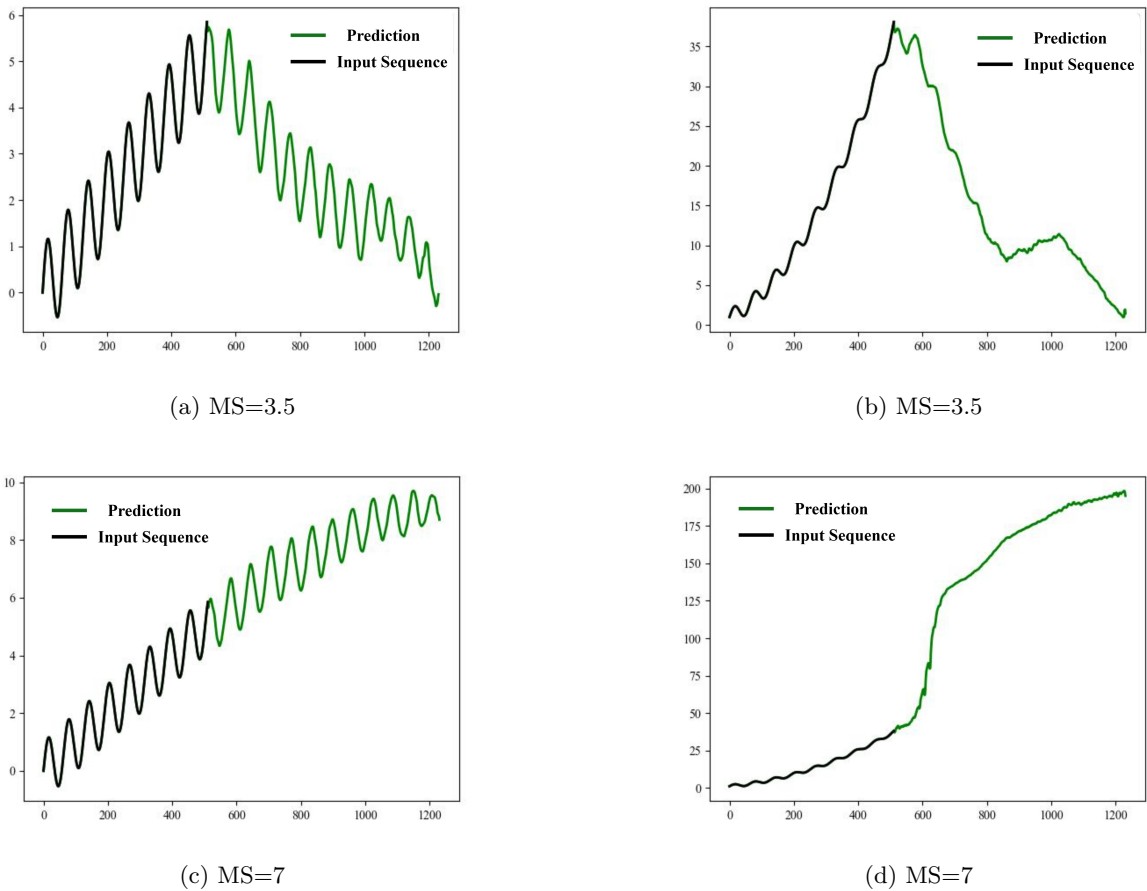

(a) MS=3.5          (b) MS=3.5

(c) MS=7          (d) MS=7

Figure 9: Resolution analysis for explosive growth patterns: (a-b) With MS=3.5, ViTime incorrectly predicts peak decline due to spatial constraints. (c-d) Doubling MS to 7 enables accurate growth trend capture.

# 5 Discussion

While ViTime demonstrates state-of-the-art performance in accuracy and robustness, two key challenges warrant further investigation.

## 5.1 Resolution Constraints & Adaptive Enhancement.

The mapping function's truncation imposes resolution limits, particularly evident in explosive growth patterns (Figure 9 a-b). A key limitation of ViTime arises from its assumption of $\mathcal{S} \sim N(0, \mathbf{I})$, which fails to capture

the high-variance nature of explosive growth data that typically follows $\mathcal{S} \sim N(0, k\mathbf{I})$ with $k \gg 1$. As shown in Proposition 3.6, the optimal threshold $MS^*$ scales as $\sqrt{k}$, implying that fixed thresholds (e.g., $MS = 3.5$ for $k = 1.5$) become suboptimal for high-variance scenarios, introducing significant system errors and degrading prediction accuracy.

Our empirical analysis reveals that doubling the MS parameter from 3.5 to 7 significantly improves prediction fidelity for explosive growth patterns (Figure 9c-d). However, excessively large MS values increase system error, as demonstrated in Theorem 3.3, leading to computational inefficiency. This trade-off suggests two complementary research directions:

- **Elastic Resolution Enhancement**: Techniques to dynamically adjust spatial resolution $h$ based on data variance, ensuring sufficient granularity for high-variance regions without unnecessary computational overhead.

- **Adaptive MS Estimation**: Algorithms to estimate the variance scaling factor $k$ and compute the optimal $MS^*$ in real-time, balancing prediction fidelity with spectral efficiency.

These enhancements would enable ViTime to handle explosive growth patterns more effectively while maintaining computational tractability.

### 5.2 Future Directions for Synthetic Data Generation in ViTime

RealTS, our synthetic data generation algorithm, plays a crucial role in performance of ViTime by creating diverse and realistic training samples that significantly enhance model generalizability. The algorithm enriches the training data through sophisticated pattern synthesis, enabling ViTime to achieve superior zero-shot and few-shot learning capabilities as demonstrated in our experiments.

While RealTS has proven effective in the current framework, several enhancements could further improve ViTime's predictive quality: 1) More advanced pattern injection mechanisms to capture complex real-world dynamics such as non-stationary processes and regime-switching scenarios. 2) Development of quantitative metrics for assessing simulation fidelity across different temporal regimes. 3) Extension to multivariate time series generation. Although theoretically feasible by incorporating additional channels in RealTS, this extension presents practical challenges in computational demand and in generating synthetic data that preserves realistic inter-variable correlations. These represent important directions for strengthening ViTime's data generation capabilities.

## 6 Conclusions

This work developed a vision intelligence-powered computational paradigm, ViTime, for developing the TSF foundation model, as compared with the numerical data fitting principles prevalently considered in literature. ViTime was inspired by human visual cognitive processes in understanding and analyzing time series. By introducing a paradigm of operating numerical data in image space and a unique deep network based computing pipeline, ViTime is capable of handling both point and probabilistic forecasting, elevating the SOTA performance on zero-shot/fine-tuning TSF without relying on prior data samples. This demonstrates the great potential for reshaping the computational mechanism in TSF foundation model development. Moreover, as data often suffer from diverse contamination and variability in reality, ViTime's visual approach enables robust performance under various real-world data perturbations and alterations, showcasing its superior resilience.

### Acknowledgments

This work was supported in part by the National Natural Science Foundation of China under grant 62402384, in part by the Hong Kong RGC General Research Fund Project under grant 11213124, in part by the Hong Kong RGC Collaborative Research Fund Project under grant C1049-24GF, in part by the Shenzhen-Hong

Kong-Macau Science and Technology Category C Project under grant SGDX20220530111205037, in part by the Hong Kong ITC Innovation and Technology Fund Project under grant ITS/034/22MS, and in part by InnoHK initiative, The Government of the HKSAR, and Laboratory for AI-Powered Financial Technologies.

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

# A   Details of RealTS

We present RealTS, a versatile framework for synthesizing realistic time series data. RealTS employs multiple data behavior modes under two main hypotheses: periodic ($\varphi_p$) and trend ($\varphi_t$). This section details the various behavior modes, their configurations, and provides visual examples.

## A.1   Periodic Hypothesis Behaviors

Under the periodic hypothesis $\varphi_p$, we employ two distinct data behavior modes:

### A.1.1   Inverse Fast Fourier Transform Behavior (IFFTB)

To ensure the synthesized data adequately reflects the variation paradigms of real-world time series, we utilize IFFT as expressed in Equation (30) to simulate the underlying behavior of real-world periodic time series:

$$P\left(\mathbf{s_L}|L, B_p\right)|_{B_p=\mathrm{IFFT}} = \iint_{-\infty}^{\infty} \mathbf{N}\left(\mathbf{A_m}; \mu_{\mathbf{A_m}}, \sigma_{\mathbf{A_m}}^2\right) \cdot \mathbf{N}\left(\phi; \mu_{\mathbf{P}}, \sigma_{\mathbf{P}}^2\right) \times \delta\left(\mathbf{s_L} - \mathrm{IFFT}\left(\mathbf{A_m}, \phi, L\right)\right) d\phi d\mathbf{A_m} \quad (30)$$

where two empirical distributions of Fourier transform amplitudes and phases, $N(A_m; \mu_{A_m}, \sigma_{A_m}^2)$ and $N(\phi; \mu_P, \sigma_P^2)$, are maintained, and $\delta$ denotes the Dirac delta function. By sampling from empirical distributions, we can obtain the amplitude and Phase vector, which is then inversely transformed back to the time domain via IFFT.

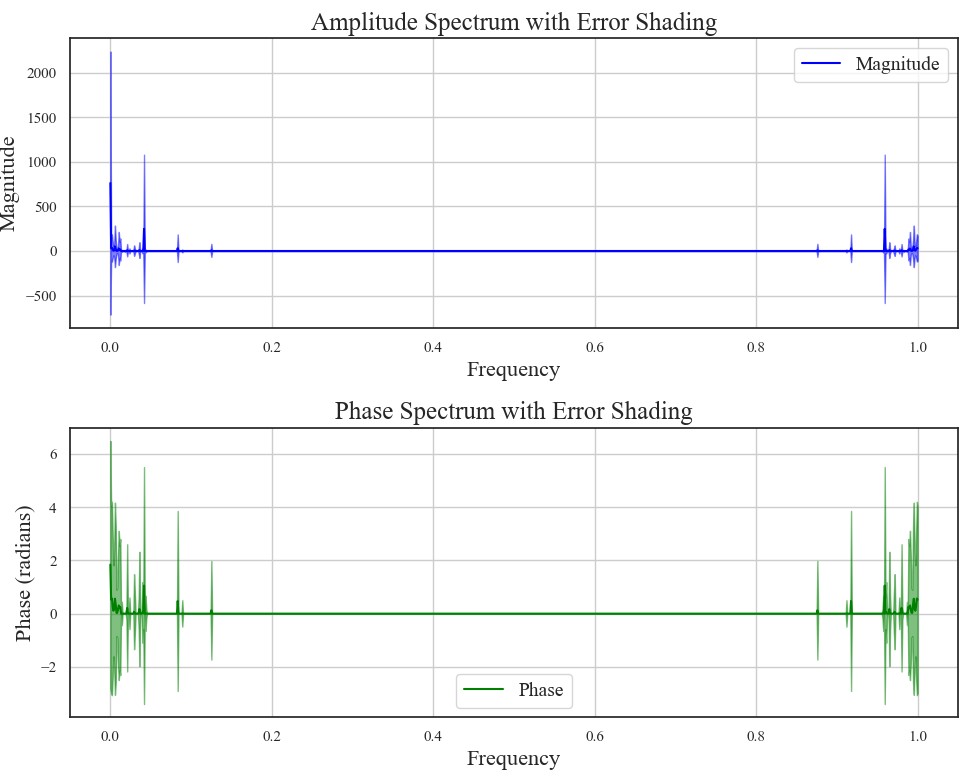

Figure 10: Empirical distribution I employed in IFFTB.

The empirical distributions utilized in $\mathbf{A_m}$ and $\phi$ are illustrated in Figure 10-Figure 11. During experiments, we randomly select one of two empirical distributions for generating $\mathbf{A_m}$ and $\phi$. Figure 12 shows examples of time series generated using IFFTB.

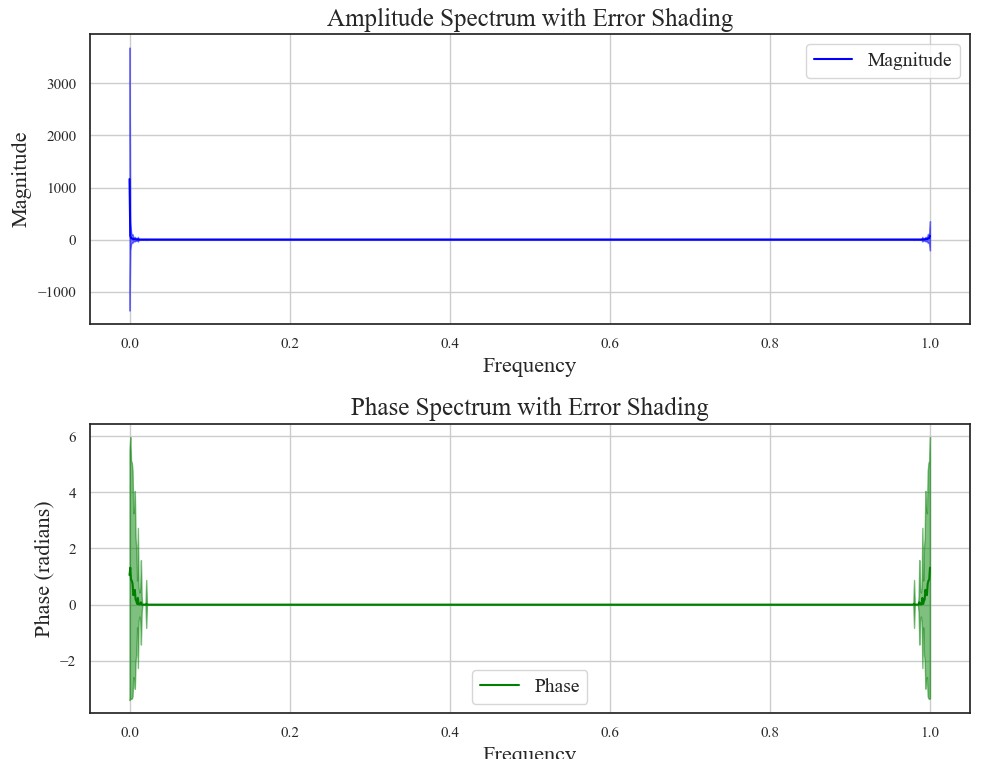

Figure 11: Empirical distribution II employed in IFFTB.

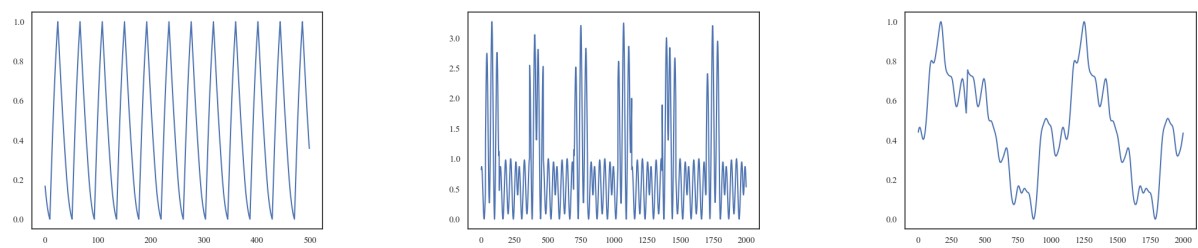

Figure 12: Examples of time series generated using IFFTB.

### A.1.2 Periodic Wave Behavior (PWB)

This behavior generates data by superimposing multiple periodic waves, which is modeled as a sum of sin, cos, and other periodic functions, $f_{\text{periodic}}$, with different frequencies and amplitudes:

$$P\left(\mathbf{s_L}|L, B_p\right)|_{B_p=\text{PWB}} = \iint_{-\infty}^{\infty} \mathbf{N}\left(\mathbf{s_L}; \sum_{i=1}^{k_{\text{PWB}}} A_i f_{\text{periodic}}\left(\omega_i t\right), \sigma_\epsilon^2\right) \times \mathbf{P}\left(\mathbf{A}\right) \mathbf{P}\left(\omega\right) d\omega d\mathbf{A} \tag{31}$$

where $\mathbf{P(A)}$ and $\mathbf{P}(\boldsymbol{\omega})$ denote predefined prior distributions of amplitudes and frequency; $k_{PWB}$ denotes the number of mixed periodic functions.

For PWB, we define the prior distributions for amplitude and frequency as:

$$\mathbf{A} \sim \mathbf{U}(\mathbf{0.5}, \mathbf{5}) \tag{32}$$

$$\ln(\omega) \sim \mathbf{U}(\ln(\mathbf{11}), \ln(\mathbf{2L})) \tag{33}$$

The parameter $k_{PWB}$ is modeled as:

$$P(k_{PWB} = k) = \frac{1}{8}, \text{ for } k = 1, 2, \ldots, 8 \tag{34}$$

Figure 13 shows examples of time series generated using PWB.

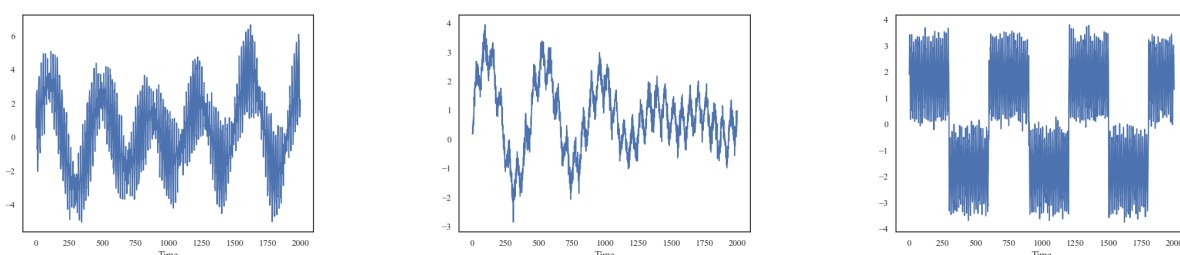

Figure 13: Examples of time series generated using PWB.

## A.2 Trend Data Hypothesis Behaviors

Under the trend data hypothesis $\varphi_t$, we employ three distinct data behavior modes:

### A.2.1 Random Walk Behavior (RWB)

The RWB models data as a stochastic process where each value is the previous value plus a random step:

$$P\left(s_i|s_{i-1}, L, B_p\right)|_{B_p=\text{RWB}} = \mathbf{N}\left(0, \sigma^2\right) \tag{35}$$

Figure 14 shows examples of time series generated using RWB.

### A.2.2 Logistic Growth Behavior (LGB)

The LGB models data with a logistic growth function, capturing the S-shaped growth pattern:

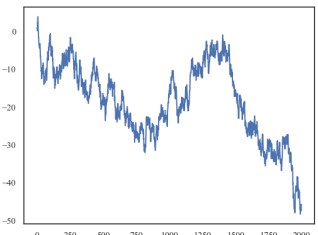 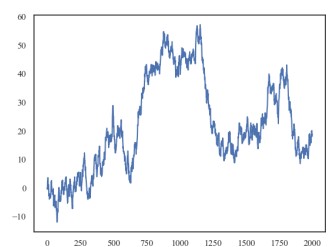 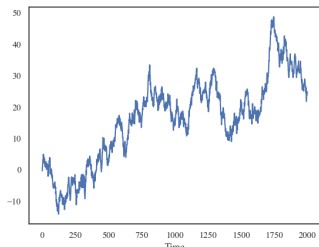

Figure 14: Examples of time series generated using RWB.

$$
\begin{aligned}
&P\left(\mathbf{s_L}|L, B_p\right)|_{B_p=\mathrm{LGB}} \\
&= \iint_{-\infty}^{\infty} \mathbf{N}\left(\mathbf{s_L}; \frac{K}{1+e^{-r(\mathbf{L}-L_0)}}, \sigma_\epsilon^2\right) P(K)P(r)dKdr
\end{aligned}
\tag{36}
$$

where $P(K)$ and $P(r)$ denote predefined prior distributions of S-shaped function hyperparameters.

For LGB, we define the probability densities for Carrying Capacity $K$ and Growth Rate $r$ as:

$$
\ln(K) \sim U(\ln(1), \ln(10))
\tag{37}
$$

$$
\ln(r) \sim U(\ln(0.001), \ln(0.1))
\tag{38}
$$

Figure 15 shows examples of time series generated using LGB.

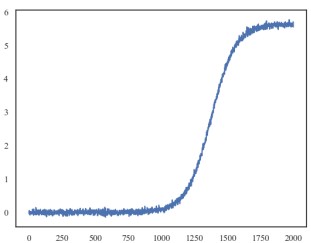 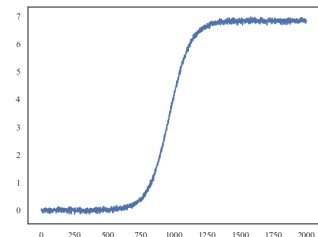 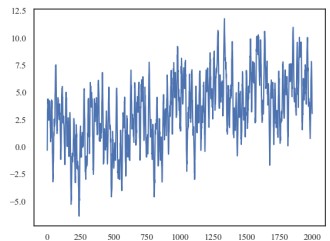

Figure 15: Examples of time series generated using LGB.

### A.2.3 Trend Wave Data Behavior (TWDB)

TWDB combines linear trends with periodic fluctuations:

$$
P\left(\mathbf{s_L}|L, B_p\right)|_{B_p=\mathrm{TWDB}} = \iint_{-\infty}^{\infty} \mathbf{N}\left(\mathbf{s_L}; a\mathbf{L} + b + \sum_{i=1}^{k_{\mathrm{TWDB}}} A_i f_{\mathrm{periodic}}\left(\omega_i t\right), \sigma_\epsilon^2\right) \times P(a)P(b)\mathbf{P}\left(\mathbf{A}\right)\mathbf{P}\left(\omega\right) da\,db\,d\mathbf{A}\,d\omega
\tag{39}
$$

where $P(a)$, $P(b)$, $P\left(\mathbf{A}\right)$ and $\mathbf{P}\left(\omega\right)$ are predefined prior distributions of hyperparameters.

In the TWDB, we define the probability densities for linear function random variables $P(a)$ and $P(b)$, as well as for the superimposed periodic wave components $P(\mathbf{A})$ and $P(\omega)$. The settings for $P(\mathbf{A})$, $P(\omega)$, and

$k_{TWDB}$ are consistent with those used in the PWB module. The probability densities for $P(a)$ and $P(b)$ are detailed below:

$$a \sim U(-1, 1) \tag{40}$$

$$b \sim U(-10, 10) \tag{41}$$

Figure 16 shows examples of time series generated using TWDB.

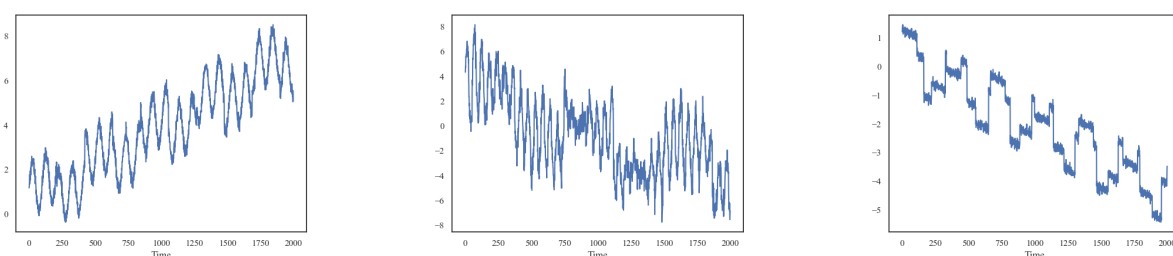

Figure 16: Examples of time series generated using TWDB.

### A.3   Data Augmentation Techniques

We sample IFFTBI, IFFTBII, PWB, RWB, LGB, and TWDB included in RealTS according to the following probability distribution: $P(\text{IFFTBI}) = 0.3$, $P(\text{IFFTBII}) = 0.3$, $P(\text{PWB}) = 0.16$, $P(\text{RWB}) = 0.08$, $P(\text{LGB}) = 0.08$, $P(\text{TWDB}) = 0.08$. To enhance the diversity and robustness of synthetic data, we employ various data augmentation techniques, such as:

- Multiple period replication - repeating the generated periodic data over multiple cycles to capture long-term periodic patterns.

- Data flipping - reversing the time series to create new patterns while preserving underlying characteristics.

- Convolution smoothing and detrending - removing underlying trends from the data to isolate periodic components, making it easier for the model to learn these patterns.

- Data perturbation - introducing sudden changes or anomalies into the data, simulating real-world disturbances and improving the ability of the model to handle unexpected variations.

More details of RealTS are offered in the code part of the Supplementary Material.

## B   Training Configuration

### B.1   ViTime Model Structure

The detailed network configuration of the proposed ViTime is reported in Table 8.

### B.2   Data Normalization

To ensure ViTime can effectively capture patterns involving sudden changes, an in-sequence data normalization based on L2 normalization is implemented. By normalizing each sequence within the data sequence, the model can pay more attention to abrupt variations. The normalization process is defined as follows:

Table 8: Details of model architecture

| Module | Embed_dim | Depth | Patch size | Num_heads |
|---|---|---|---|---|
| Visual Time Tokenizer | 768 | 9 | (4,32) | 12 |
| Decoder | 384 | 4 | \ | 12 |

| Refining Module | | |
|---|---|---|
| Component | Configuration | Details |
| Backbone | MobileNetV2 | Downsample factor: 16 |
| ASPP | 5 branches | Dilation rates: 1,6,12,18,GAP |
| Low-level features | Conv1x1 | 24→48 channels |
| Feature fusion | Conv3x3×2 | 256 channels |
| Upsampling | DeconvASPP | 6 branches with SE attention |

$$\mathbf{S_L} = \frac{\mathbf{S_L} - \mathrm{mean}\left(\|\mathbf{S_{1:T}}\|_2\right)}{\mathrm{std}\left(\mathbf{S_{1:T}}\right)} \tag{42}$$

### B.3 Training Set of ViTime-TFM

The complete training datasets of ViTime-TFM are detailed in Table 9.

Table 9: Training dataset composition for ViTime-TFM

| Dataset | Granularity | Time series | Time points |
|---|---|---|---|
| Synthetic | Hourly | 3,000,000 | 6,144,000,000 |
| Electricity | Hourly | 321 | 8,443,584 |
| Traffic | Hourly | 862 | 15,122,928 |
| Weather | 10 Min | 42 | 2,213,232 |
| Favorita Sales | Daily | 111,840 | 139,179,538 |
| LibCity | 15 Min | 6,159 | 34,253,622 |
| M4 hourly | Hourly | 414 | 353,500 |
| M4 daily | Daily | 4,227 | 9,964,658 |
| M4 monthly | Monthly | 48,000 | 10,382,411 |
| M4 quarterly | Quarterly | 24,000 | 2,214,108 |
| M4 yearly | Yearly | 22,739 | 840,644 |

### B.4 Benchmark Model Configurations

To ensure full transparency and reproducibility, this section details the configurations for the baseline models used in our comparisons.

**1. Zero-Shot Forecasting (Table 2):** For the zero-shot forecasting results, we used the following publicly available pre-trained models:

- **TimesFM:** `timesfm-1.0-200m`

- **Moirai:** `moirai-1.1-R-large`

- **Moment:** `MOMENT-1-large`

- **VisionTS:** `mae_base`

- **PatchTST:** `PatchTST/64`

**2. Fine-Tuning (Table 4):** The performance scores reported for the fine-tuning experiments are the official results cited directly from the original publications of the respective baseline models.

**3. Probabilistic Forecasting (Table 3):** For the probabilistic forecasting experiments, we configured the baseline models as follows:

- **Moirai:** We used the pre-trained model `moirai-1.1-R-large`.

- **Lag-Llama:** We utilized the publicly available implementation from the original Lag-Llama paper and its provided checkpoint, `lag-llama.ckpt`.

## C Proofs

This section provides the detailed proofs for the theorems and propositions presented in the main text.

### C.1 Proof of Theorem 3.3 (System Error Upper Bound)

**Theorem C.1** (Theorem 3.3 restated). *Given a tensor $\widehat{s} \in \mathcal{S} \subset \mathbb{R}^{c \times L}$, where $\mathcal{S}$ follows $N(\mathbf{0}, \mathbf{I})$ as per Assumption 3.2, the system error (SE) from mapping to $\mathcal{V}$ and back, defined as $\left\| f^{-1}\left(\mathbf{f}\left(\widehat{s}\right)\right) - \widehat{s} \right\|_1$, satisfies the following expectation bound:*

$$\text{SE} := \mathbb{E} \left\| f^{-1}\left(\mathbf{f}\left(\widehat{s}\right)\right) - \widehat{s} \right\|_1 \le cL \left[ MS \left( \frac{1}{h} \left( \Phi(MS) - \Phi(-MS) \right) - 2 + 2\Phi(MS) \right) + \sqrt{\frac{2}{\pi}} e^{-\frac{MS^2}{2}} \right], \quad (43)$$

*where $\Phi$ is the cumulative distribution function (CDF) of $N(0,1)$, $c$ is the number of variates, $L$ is the time series length, $h$ is the image height (resolution), and $MS$ is the maximum scale.*

*Proof.* The proof considers the error for a single element $s$ of $\widehat{s}$ and then scales by $cL$. Let $P(s)$ be the PDF of $N(0,1)$. The expected absolute error for a single element is $\mathbb{E}|f^{-1}(f(s)) - s|$. This error can be decomposed into two parts: quantization error for $|s| \le MS$ and truncation error for $|s| > MS$.

**1. Quantization Error ($|s| \le MS$):**

When $|s| \le MS$, the value $s$ is mapped to a bin $j = \lfloor (s + MS)/(2MS/h) \rfloor$. The inverse mapping $f^{-1}(f(s))$ reconstructs this as the midpoint of the bin, $(j - 0.5)(2MS/h) - MS$. The maximum error in this case is half the bin width, $\delta/2 = (2MS/h)/2 = MS/h$.

The expected quantization error is:

$$E_Q = \int_{-MS}^{MS} |f^{-1}(f(s)) - s| P(s) ds \quad (44)$$

$$\le \int_{-MS}^{MS} \frac{MS}{h} P(s) ds = \frac{MS}{h} \int_{-MS}^{MS} P(s) ds \quad (45)$$

$$= \frac{MS}{h} \left[ \Phi(MS) - \Phi(-MS) \right]. \quad (46)$$

**2. Truncation Error ($|s| > MS$):**

If $s > MS$, $f(s)$ maps to the highest bin $h$, and $f^{-1}(f(s)) = MS - (MS/h)$. The error is $s - (MS - MS/h)$.

If $s < -MS$, $f(s)$ maps to the lowest bin 1, and $f^{-1}(f(s)) = -MS + (MS/h)$. The error is $(-MS + MS/h) - s$.

For simplicity in bounding, we consider the error magnitude as $|s| - MS$ when $|s| > MS$. The expected truncation error is:

$$E_T = \int_{MS}^{\infty} (s - MS)P(s)ds + \int_{-\infty}^{-MS} (-MS - s)P(s)ds \tag{47}$$

$$= 2\int_{MS}^{\infty} (s - MS)P(s)ds \quad \text{(by symmetry of } P(s)) \tag{48}$$

$$= 2\left[\int_{MS}^{\infty} sP(s)ds - MS\int_{MS}^{\infty} P(s)ds\right]. \tag{49}$$

We know $\int_{MS}^{\infty} sP(s)ds = \int_{MS}^{\infty} s\frac{1}{\sqrt{2\pi}}e^{-s^2/2}ds = \frac{1}{\sqrt{2\pi}}e^{-MS^2/2}$.

And $\int_{MS}^{\infty} P(s)ds = 1 - \Phi(MS)$.

So,

$$E_T = 2\left[\frac{1}{\sqrt{2\pi}}e^{-MS^2/2} - MS(1 - \Phi(MS))\right] \tag{50}$$

$$= \sqrt{\frac{2}{\pi}}e^{-MS^2/2} - 2MS(1 - \Phi(MS)). \tag{51}$$

### 3. Total Expected Error per Element:

The total expected absolute error for one element is $E_Q + E_T$:

$$\mathbb{E}|f^{-1}(f(s)) - s| \leq \frac{MS}{h}[\Phi(MS) - \Phi(-MS)] + \sqrt{\frac{2}{\pi}}e^{-MS^2/2} - 2MS(1 - \Phi(MS)) \tag{52}$$

$$= MS\left(\frac{1}{h}(\Phi(MS) - \Phi(-MS)) - 2(1 - \Phi(MS))\right) + \sqrt{\frac{2}{\pi}}e^{-\frac{MS^2}{2}} \tag{53}$$

$$= MS\left(\frac{1}{h}(\Phi(MS) - \Phi(-MS)) - 2 + 2\Phi(MS)\right) + \sqrt{\frac{2}{\pi}}e^{-\frac{MS^2}{2}}. \tag{54}$$

Multiplying by $cL$ (number of elements) gives the bound for $\mathbb{E}\|f^{-1}(\mathbf{f}(\widehat{s})) - \widehat{s}\|_1$. $\qquad\square$

### C.2 Proof of Proposition 3.4 (Asymptotic Convergence with $h$)

**Proposition C.2** (Proposition 3.4 restated). *For any $\varepsilon > 0$, there exists $\delta_0 > 0$ such that when $h \to +\infty$ and $MS \geq \delta_0$, the per-element SE upper bound*

$$g_1(h, MS) = MS\left(\frac{1}{h}(\Phi(MS) - \Phi(-MS)) - 2 + 2\Phi(MS)\right) + \sqrt{\frac{2}{\pi}}e^{-\frac{MS^2}{2}} \tag{55}$$

*converges to zero.*

*Proof.* Let $g_1(h, MS)$ be the per-element upper bound from Theorem 3.3:

$$g_1(h, MS) = \frac{MS}{h}(\Phi(MS) - \Phi(-MS)) - 2MS(1 - \Phi(MS)) + \sqrt{\frac{2}{\pi}}e^{-\frac{MS^2}{2}}. \tag{56}$$

As $h \to +\infty$, the term $\frac{MS}{h}(\Phi(MS) - \Phi(-MS)) \to 0$ since $\Phi(MS) - \Phi(-MS) \leq 1$.

The remaining terms are $R(MS) = -2MS(1 - \Phi(MS)) + \sqrt{\frac{2}{\pi}}e^{-\frac{MS^2}{2}}$.

We use Mill's ratio for the tail probability of a standard normal distribution: for $MS > 0$,

$$1 - \Phi(MS) \sim \frac{\phi(MS)}{MS} = \frac{1}{MS\sqrt{2\pi}}e^{-MS^2/2} \quad \text{as } MS \to \infty. \tag{57}$$

So,

$$-2MS(1 - \Phi(MS)) \sim -2MS\left(\frac{1}{MS\sqrt{2\pi}}e^{-MS^2/2}\right) \tag{58}$$

$$= -\sqrt{\frac{2}{\pi}}e^{-MS^2/2}. \tag{59}$$

Thus, as $MS \to \infty$,

$$R(MS) \sim -\sqrt{\frac{2}{\pi}}e^{-MS^2/2} + \sqrt{\frac{2}{\pi}}e^{-MS^2/2} \tag{60}$$

$$= 0. \tag{61}$$

Therefore, for any $\varepsilon > 0$, we can find a $\delta_0$ such that for $MS \geq \delta_0$, $|R(MS)| < \varepsilon/2$.

And for any $MS \geq \delta_0$, we can find an $H_0$ such that for $h \geq H_0$,

$$\left|\frac{MS}{h}(\Phi(MS) - \Phi(-MS))\right| < \varepsilon/2. \tag{62}$$

This implies that $\lim_{h \to +\infty, MS \to \infty} g_1(h, MS) = 0$. More precisely, for a fixed large enough $MS$, as $h \to \infty$, the limit is $R(MS)$, which can be made arbitrarily small by choosing $MS$ large.

The statement asks for convergence as $h \to \infty$ for $MS \geq \delta_0$.

Let $MS \geq \delta_0$. Then

$$\lim_{h \to +\infty} g_1(h, MS) = -2MS(1 - \Phi(MS)) + \sqrt{\frac{2}{\pi}}e^{-MS^2/2}. \tag{63}$$

This limit itself tends to 0 as $MS \to \infty$. The proposition asks for the expression to be small when $h \to \infty$ AND $MS \geq \delta_0$.

Taking the limit as $h \to \infty$ first, we get:

$$\lim_{h \to +\infty}\left|MS\left(\frac{1}{h}(\Phi(MS) - \Phi(-MS)) - 2 + 2\Phi(MS)\right) + \sqrt{\frac{2}{\pi}}e^{-\frac{MS^2}{2}}\right| = \left|-2MS(1 - \Phi(MS)) + \sqrt{\frac{2}{\pi}}e^{-\frac{MS^2}{2}}\right|. \tag{64}$$

This term goes to 0 as $MS \to \infty$. So, for any $\varepsilon > 0$, there exists $\delta_0$ such that for $MS \geq \delta_0$, the term is less than $\varepsilon$. $\square$

### C.3 Proof of Proposition 3.5 (Optimal MS Selection)

**Proposition C.3** (Proposition 3.5 restated). *For a fixed $h$, there exists a unique optimal threshold $MS^* > 0$ that minimizes the per-element SE upper bound $g_1(h, MS)$. This $MS^*$ is the solution to:*

$$\frac{1}{h}(\Phi(MS^*) - \Phi(-MS^*)) - 2 + 2\Phi(MS^*) + \frac{MS^*}{h} \cdot 2\phi(MS^*) = 0, \tag{65}$$

*where $\phi(x) = \frac{1}{\sqrt{2\pi}}e^{-x^2/2}$ is the PDF of $N(0,1)$. (Note: The original equation had $\sqrt{2/\pi}e^{-MS^{*2}/2}$, which is $2\phi(MS^*)$).*

*Proof.* Let $g_1(MS) = MS \left( \frac{1}{h}(\Phi(MS) - \Phi(-MS)) - 2 + 2\Phi(MS) \right) + \sqrt{\frac{2}{\pi}} e^{-\frac{MS^2}{2}}$.

We want to find $MS^*$ such that $g_1'(MS^*) = 0$.

Using $\Phi(-x) = 1 - \Phi(x)$ and $\phi(-x) = \phi(x)$, we have $\Phi(MS) - \Phi(-MS) = 2\Phi(MS) - 1$.

So, $g_1(MS) = MS \left( \frac{1}{h}(2\Phi(MS) - 1) - 2 + 2\Phi(MS) \right) + 2\phi(MS)$.

Derivative with respect to $MS$:

$$\frac{dg_1}{dMS} = \left( \frac{1}{h}(2\Phi(MS) - 1) - 2 + 2\Phi(MS) \right) + MS \left( \frac{2\phi(MS)}{h} + 2\phi(MS) \right) + 2\phi'(MS) \tag{66}$$

$$= \frac{2\Phi(MS) - 1}{h} - 2 + 2\Phi(MS) + \frac{2MS\phi(MS)}{h} + 2MS\phi(MS) - 2MS\phi(MS) \quad \text{(since } \phi'(MS) = -MS\phi(MS)) \tag{67}$$

$$= \frac{2\Phi(MS) - 1}{h} - 2 + 2\Phi(MS) + \frac{2MS\phi(MS)}{h}. \tag{68}$$

Setting $dg_1/dMS = 0$:

$$\frac{2\Phi(MS^*) - 1}{h} - 2 + 2\Phi(MS^*) + \frac{2MS^*\phi(MS^*)}{h} = 0. \tag{69}$$

This matches the condition in the proposition since $\Phi(MS^*) - \Phi(-MS^*) = 2\Phi(MS^*) - 1$.

To show uniqueness and minimality, we examine the second derivative or the behavior of the first derivative.

Let $f(MS) = dg_1/dMS$.

$f(0) = (0 - 1)/h - 2 + 2(0.5) + 0 = -1/h - 2 + 1 = -1 - 1/h < 0$.

As $MS \to \infty$, $\Phi(MS) \to 1$ and $\phi(MS) \to 0$.

So $\lim_{MS \to \infty} f(MS) = 1/h - 2 + 2 + 0 = 1/h > 0$ (assuming $h > 0$).

Since $f(MS)$ is continuous and goes from negative to positive, there must be at least one root $MS^* > 0$.

The second derivative:

$$\frac{d^2g_1}{dMS^2} = \frac{2\phi(MS)}{h} + 2\phi(MS) + \frac{2\phi(MS) + 2MS\phi'(MS)}{h} \tag{70}$$

$$= 2\phi(MS) \left( \frac{1}{h} + 1 \right) + \frac{2\phi(MS) - 2MS^2\phi(MS)}{h} \tag{71}$$

$$= 2\phi(MS) \left( 1 + \frac{2}{h} - \frac{MS^2}{h} \right). \tag{72}$$

For small $MS$, $d^2g_1/dMS^2 > 0$, indicating convexity. If $1 + 2/h - MS^2/h > 0$, i.e., $MS^2 < h + 2$.

If $MS^* < \sqrt{h + 2}$, then $g_1(MS)$ is convex at $MS^*$, ensuring a local minimum.

The function $f(MS)$ starts negative, becomes positive, and its derivative $d^2g_1/dMS^2$ is positive for $MS < \sqrt{h + 2}$ and can become negative for $MS > \sqrt{h + 2}$. This structure ensures a unique minimum for $MS > 0$. $\square$

### C.4 Proof of Proposition 3.6 (Optimal Threshold under Variance Scaling)

**Proposition C.4** (Proposition 3.6 restated). *Under the assumption $\mathcal{S} \sim N(0, k\mathbf{I})$ with $k > 1$, the optimal threshold $MS^*$ that minimizes the per-element SE upper bound is characterized by:*

$$\frac{1}{h} \left( \Phi\left( \frac{MS^*}{\sqrt{k}} \right) - \Phi\left( -\frac{MS^*}{\sqrt{k}} \right) \right) - 2 + 2\Phi\left( \frac{MS^*}{\sqrt{k}} \right) + \frac{MS^*}{h} \sqrt{\frac{2}{\pi k}} e^{-\frac{(MS^*)^2}{2k}} = 0. \tag{73}$$

*Proof.* Let $s \sim N(0, k)$. Then $s' = s/\sqrt{k} \sim N(0, 1)$.

The original values $s$ are scaled by $\sqrt{k}$. The mapping function $f$ operates on $s$. The bins are from $-MS$ to $MS$. Bin width $\delta_s = 2MS/h$.

The SE upper bound for a single element is: $g_k(MS) = \mathbb{E}_{s \sim N(0,k)} |f^{-1}(f(s)) - s|$.

This is equivalent to scaling the original problem. Let $s = \sqrt{k}z$ where $z \sim N(0, 1)$. The function operates on $s$. The effective range for $z$ is $-MS/\sqrt{k}$ to $MS/\sqrt{k}$.

**The quantization error part:**

$s$ is in $[-MS, MS]$. The error is bounded by $MS/h$.

$$\int_{-MS}^{MS} \frac{MS}{h} P_k(s) ds = \frac{MS}{h} \int_{-MS}^{MS} \frac{1}{\sqrt{2\pi k}} e^{-s^2/(2k)} ds \tag{74}$$

Let $u = s/\sqrt{k}$. Then $ds = \sqrt{k} du$. Limits become $-MS/\sqrt{k}$ to $MS/\sqrt{k}$.

$$= \frac{MS}{h} \int_{-MS/\sqrt{k}}^{MS/\sqrt{k}} \frac{1}{\sqrt{2\pi}} e^{-u^2/2} du \tag{75}$$

$$= \frac{MS}{h} \left[ \Phi\left(\frac{MS}{\sqrt{k}}\right) - \Phi\left(-\frac{MS}{\sqrt{k}}\right) \right] \tag{76}$$

**The truncation error part:**

$2 \int_{MS}^{\infty} (s - MS) P_k(s) ds$.

$$= 2 \left[ \int_{MS}^{\infty} s \frac{1}{\sqrt{2\pi k}} e^{-s^2/(2k)} ds - MS \int_{MS}^{\infty} \frac{1}{\sqrt{2\pi k}} e^{-s^2/(2k)} ds \right] \tag{77}$$

The first integral:

$$\int_{MS}^{\infty} s \frac{1}{\sqrt{2\pi k}} e^{-s^2/(2k)} ds = \sqrt{k} \int_{MS/\sqrt{k}}^{\infty} u \frac{1}{\sqrt{2\pi}} e^{-u^2/2} du \tag{78}$$

$$= \sqrt{k} \frac{1}{\sqrt{2\pi}} e^{-(MS/\sqrt{k})^2/2} \tag{79}$$

$$= \sqrt{\frac{k}{2\pi}} e^{-MS^2/(2k)} \tag{80}$$

The second integral: $MS \left(1 - \Phi\left(\frac{MS}{\sqrt{k}}\right)\right)$.

So,

$$E_{T,k} = 2 \left[ \sqrt{\frac{k}{2\pi}} e^{-MS^2/(2k)} - MS \left(1 - \Phi\left(\frac{MS}{\sqrt{k}}\right)\right) \right] \tag{81}$$

The per-element SE bound $g_{1,k}(MS)$ is:

$$g_{1,k}(MS) = \frac{MS}{h} \left[ \Phi_k(MS) - \Phi_k(-MS) \right] + \sqrt{\frac{2k}{\pi}} e^{-MS^2/(2k)} - 2MS(1 - \Phi_k(MS)) \tag{82}$$

where $\Phi_k(x) = \Phi(x/\sqrt{k})$.

This can be written as:

$$g_{1,k}(MS) = MS\left(\frac{1}{h}(\Phi_k(MS) - \Phi_k(-MS)) - 2 + 2\Phi_k(MS)\right) + \sqrt{\frac{2k}{\pi}}e^{-\frac{MS^2}{2k}} \tag{83}$$

To find the optimal $MS^*$, we differentiate $g_{1,k}(MS)$ with respect to $MS$ and set to zero.

Let $\phi_k(x) = \frac{1}{\sqrt{k}}\phi(x/\sqrt{k})$ be the PDF of $N(0,k)$ in terms of $\phi$.

$\frac{d}{dMS}\Phi_k(MS) = \frac{d}{dMS}\Phi(MS/\sqrt{k}) = \phi(MS/\sqrt{k}) \cdot \frac{1}{\sqrt{k}} = \phi_k(MS).$

$\frac{d}{dMS}\left(\sqrt{\frac{2k}{\pi}}e^{-\frac{MS^2}{2k}}\right) = \sqrt{\frac{2k}{\pi}}e^{-\frac{MS^2}{2k}}\left(-\frac{2MS}{2k}\right) = -\frac{MS}{\sqrt{k}}\sqrt{\frac{2}{\pi}}e^{-\frac{MS^2}{2k}} = -2MS\phi_k(MS).$

The derivative $\frac{dg_{1,k}}{dMS}$ is:

$$= \left(\frac{1}{h}(\Phi_k(MS) - \Phi_k(-MS)) - 2 + 2\Phi_k(MS)\right) + MS\left(\frac{1}{h}(\phi_k(MS) - (-\phi_k(MS))) + 2\phi_k(MS)\right) - 2MS\phi_k(MS) \tag{84}$$

$$= \frac{\Phi_k(MS) - \Phi_k(-MS)}{h} - 2 + 2\Phi_k(MS) + \frac{2MS\phi_k(MS)}{h} \tag{85}$$

Setting this to zero gives:

$$\frac{1}{h}\left(\Phi\left(\frac{MS^*}{\sqrt{k}}\right) - \Phi\left(-\frac{MS^*}{\sqrt{k}}\right)\right) - 2 + 2\Phi\left(\frac{MS^*}{\sqrt{k}}\right) + \frac{2MS^*}{h}\frac{1}{\sqrt{k}}\phi\left(\frac{MS^*}{\sqrt{k}}\right) = 0 \tag{86}$$

Substituting $2\phi(x/\sqrt{k})/\sqrt{k} = \sqrt{2/(\pi k)}e^{-(MS^*)^2/(2k)}$, we get the stated condition.

The uniqueness follows a similar argument to Proposition 3.5. $\qquad\square$

### C.5 Proof of Theorem 3.7 (Stripe SNR Boost)

**Theorem C.5** (Theorem 3.7 restated). *Let the length–$L$ time series $s_k = A\sin(\omega_0 k + \phi) + \eta_k$, $k = 0, \ldots, L-1$, with amplitude $A > 0$, angular frequency $\omega_0 = 2\pi/P_{\text{period}}$ ($P_{\text{period}} \in \mathbb{N}^+$) and i.i.d. Gaussian noise $\eta_k \sim N(0, \sigma^2)$ be visualised as the binary stripe image $v \in \{0,1\}^{h \times L}$ defined through $v_{j,k} = \mathbf{1}(j = \lfloor (s_k + \text{MS})/\delta \rfloor)$, where $\delta = \Delta/h$, $\Delta = 2\text{MS}$.*

*Denote $SNR_{\text{num}} = A^2/(2\sigma^2)$ and $SNR_{\text{vis}} = \mathbb{E}[|\mathcal{F}_{2D}(v_{\text{clean}})[0, n_0]|^2]/\mathbb{E}[|\mathcal{F}_{2D}(v_{\text{noise}})[0, n_0]|^2]$, with $n_0 = \lfloor L/P_{\text{period}} \rfloor$.*

*Assume (i) $\delta \leq A \leq \Delta - \delta$ and (ii) $\sigma < \delta/4$.*

*Then, for every $L \geq P_{\text{period}}$:*

$$SNR_{\text{vis}} \geq \frac{L}{4}\exp\left(\frac{\delta^2}{8\sigma^2}\right)\frac{\sigma^2}{A^2}SNR_{\text{num}} \tag{87}$$

$$SNR_{\text{vis}} \geq \frac{L}{4}\exp\left(\frac{\delta^2}{8\sigma^2}\right) \tag{88}$$

*Proof.* Let $v_{\text{clean}}$ be the image from $A\sin(\omega_0 k + \phi)$ and $v = v_{\text{clean}} + v_{\text{noise}}$ where $v_{\text{noise}}$ is the change due to $\eta_k$.

**1. Deterministic Signal Power in Visual Domain.**

The 2D Discrete Fourier Transform (DFT) is

$$\mathcal{F}_{2D}(v)[m, n] = \sum_{k=0}^{L-1}\sum_{j=0}^{h-1} v_{j,k}e^{-i2\pi(mk/L + nj/h)} \tag{89}$$

We are interested in the coefficient at $(m, n_p) = (0, n_0)$, where $n_0 = L/P_{\text{period}}$ (assuming $L$ is a multiple of $P_{\text{period}}$ for simplicity, or $\lfloor L/P_{\text{period}} \rfloor$ otherwise).

The transform of the clean signal component at $(0, n_0)$ is

$$\mathcal{F}_{2D}(v_{\text{clean}})[0, n_0] = \sum_{k=0}^{L-1} \sum_{j=0}^{h-1} (v_{\text{clean}})_{j,k} e^{-i2\pi(n_0 j/h)} \tag{90}$$

Following the provided analysis, $\mathbb{E}[|\mathcal{F}_{2D}(v_{\text{clean}})[0, n_0]|^2] = L^2$.

### 2. Probability of Quantization Flip.

A flip means $v_{j,k}$ changes due to noise $\eta_k$. This occurs if $s_k = s_{k,\text{clean}} + \eta_k$ crosses a quantization boundary $\theta_j = j\delta - MS$.

The clean value $s_{k,\text{clean}}$ falls into bin $j_0$. A flip occurs if $s_k$ falls into $j_0 \pm 1, j_0 \pm 2, \ldots$.

The closest boundaries are $j_0\delta - MS$ and $(j_0 + 1)\delta - MS$.

$s_{k,\text{clean}}$ is at least $\epsilon$ from any boundary. A flip to an adjacent bin occurs if $|\eta_k| > \epsilon$.

The condition $\sigma < \delta/4$ implies noise is small. A flip occurs if $\eta_k$ moves $s_k$ to another bin. This primarily happens if $s_k$ crosses $s_{k,\text{clean}} \pm \delta/2$ (approximately).

So, $p_{\text{flip}} = \Pr(|\eta_k| > \delta/2)$. Using Gaussian tail bound $\Pr(|X| > t) \leq 2e^{-t^2/(2\sigma^2)}$:

$$p_{\text{flip}} \leq 2\exp\left(-\frac{(\delta/2)^2}{2\sigma^2}\right) = 2\exp\left(-\frac{\delta^2}{8\sigma^2}\right) \tag{91}$$

### 3. Energy of the Noise Image $v_{\text{noise}}$.

$v_{\text{noise}}$ has entries $1, -1, 0$. If $s_k$ flips from bin $j_0$ to $j_1$: $(v_{\text{noise}})_{j_0,k} = -1$, $(v_{\text{noise}})_{j_1,k} = 1$.

$\|v_{\text{noise}}\|_F^2 = \sum_{k,j}(v_{\text{noise}})_{j,k}^2$. Each flip changes two pixels, so contributes $1^2 + (-1)^2 = 2$ to this sum.

$\mathbb{E}[\|v_{\text{noise}}\|_F^2] = \sum_k \mathbb{E}[\text{contribution at } k] = L \cdot (2 \cdot p_{\text{flip}})$.

### 4. Bound on a Single DFT Coefficient of Noise.

By Parseval's identity for 2D DFT: $\sum_{m,n} |\mathcal{F}_{2D}(v_{\text{noise}})[m, n]|^2 = \|v_{\text{noise}}\|_F^2$ (with appropriate normalization).

Thus, for any specific $(m, n)$, $|\mathcal{F}_{2D}(v_{\text{noise}})[m, n]|^2 \leq \|v_{\text{noise}}\|_F^2$.

Therefore, $\mathbb{E}[|\mathcal{F}_{2D}(v_{\text{noise}})[0, n_0]|^2] \leq \mathbb{E}[\|v_{\text{noise}}\|_F^2] = 2Lp_{\text{flip}}$.

### 5. Visual SNR Bound.

$$\text{SNR}_{\text{vis}} = \frac{L^2}{\mathbb{E}[|\mathcal{F}_{2D}(v_{\text{noise}})[0, n_0]|^2]} \tag{92}$$

$$\geq \frac{L^2}{2Lp_{\text{flip}}} \tag{93}$$

$$= \frac{L}{2p_{\text{flip}}} \tag{94}$$

Using $p_{\text{flip}} \leq 2\exp(-\delta^2/(8\sigma^2))$:

$$\text{SNR}_{\text{vis}} \geq \frac{L}{2 \cdot 2\exp(-\delta^2/(8\sigma^2))} \tag{95}$$

$$= \frac{L}{4}\exp\left(\frac{\delta^2}{8\sigma^2}\right) \tag{96}$$

This is the second part of the result.

### 6. Relation to Numerical SNR.

$\text{SNR}_{\text{num}} = A^2/(2\sigma^2)$.

$$\frac{\text{SNR}_{\text{vis}}}{\text{SNR}_{\text{num}}} = \text{SNR}_{\text{vis}}\frac{2\sigma^2}{A^2} \tag{97}$$

$$\geq \frac{L}{4}\exp\left(\frac{\delta^2}{8\sigma^2}\right)\frac{2\sigma^2}{A^2} \tag{98}$$

This yields

$$\text{SNR}_{\text{vis}} \geq \frac{L}{4}\exp\left(\frac{\delta^2}{8\sigma^2}\right)\frac{\sigma^2}{A^2}\text{SNR}_{\text{num}} \tag{99}$$

Thus, both inequalities in the theorem statement are proven. $\qquad\square$

### C.6 Proof of Theorem 3.8 (Gaussian-Blur SNR Boost)

**Theorem C.6** (Theorem 3.8 restated)**.** *Under the assumptions of Theorem 3.7, apply a 1D normalized Gaussian convolution $g_j = (1/Z)\exp(-j^2/(2\sigma_b^2))$ with $\sum_j g_j = 1$ along the row direction of $v$ to get $w = g *_j v$. Let $S = \sum_j g_j^2 \in (0,1)$ be the filter's nuclear energy.*

*Define $SNR_{\text{vis}}^{blur} = \mathbb{E}[|\mathcal{F}_{2D}(w_{\text{clean}})[0,n_0]|^2]/\mathbb{E}[|\mathcal{F}_{2D}(w_{\text{noise}})[0,n_0]|^2]$. Then,*

$$SNR_{\text{vis}}^{blur} \geq \frac{L}{4S}\exp\left(\frac{\delta^2}{8\sigma^2}\right) \tag{100}$$

$$SNR_{\text{vis}}^{blur} \geq \frac{L\sigma^2}{2A^2 S}\exp\left(\frac{\delta^2}{8\sigma^2}\right)SNR_{\text{num}} \tag{101}$$

*The visual SNR is amplified by at least $1/S > 1$ compared to the unblurred case.*

*Proof.* Let $G(m_v)$ be the DFT of the 1D filter $g_j$ with respect to the value-axis frequency $m_v$.

$\mathcal{F}_{2D}(w)[m_t, m_v] = G(m_v)\mathcal{F}_{2D}(v)[m_t, m_v]$.

We are interested in the frequency $(0, n_0)$, where $n_0$ is the time-axis frequency index.

### 1. Signal Power after Blurring.

Using the frequency index $(n_t, n_j)$ for (time, value/row), we examine the coefficient $\mathcal{F}_{2D}(w)[n_t, n_j]$.

The specific coefficient in focus is $\mathcal{F}_{2D}(w_{\text{clean}})[0, n_0]$, where $n_0$ is the time index.

Signal power:

$$\mathbb{E}[|\mathcal{F}_{2D}(w_{\text{clean}})[n_0, 0]|^2] = |G(0)|^2\mathbb{E}[|\mathcal{F}_{2D}(v_{\text{clean}})[n_0, 0]|^2] \tag{102}$$

Since $\sum_j g_j = 1$, we have $G(0) = 1$. So signal power remains $L^2$.

**2. Noise Power after Blurring.**

The noise image is $w_{\text{noise}} = g *_j v_{\text{noise}}$.

The total energy of $w_{\text{noise}}$:

$$\|w_{\text{noise}}\|_F^2 = \sum_k \|g * (v_{\text{noise}})_{:,k}\|_2^2 \tag{103}$$

For each column $k$, $(v_{\text{noise}})_{:,k}$ is a vector. Convolution is along $j$.

$$\|g * (v_{\text{noise}})_{:,k}\|_2^2 = S\|(v_{\text{noise}})_{:,k}\|_2^2 \tag{104}$$

where $S = \|g\|_2^2 = \sum_j g_j^2$.

So,

$$\|w_{\text{noise}}\|_F^2 = S\|v_{\text{noise}}\|_F^2 \tag{105}$$
$$\mathbb{E}[\|w_{\text{noise}}\|_F^2] = S \cdot \mathbb{E}[\|v_{\text{noise}}\|_F^2] \tag{106}$$
$$= S \cdot (2Lp_{\text{flip}}) \tag{107}$$

**3. Bound on Single DFT Coefficient of Blurred Noise.**

$$\mathbb{E}[|\mathcal{F}_{2D}(w_{\text{noise}})[n_0, 0]|^2] \leq \mathbb{E}[\|w_{\text{noise}}\|_F^2] \tag{108}$$
$$= 2LSp_{\text{flip}} \tag{109}$$

**4. SNR after Blurring.**

$$\text{SNR}_{\text{vis}}^{\text{blur}} = \frac{L^2}{\mathbb{E}[|\mathcal{F}_{2D}(w_{\text{noise}})[n_0, 0]|^2]} \tag{110}$$

$$\geq \frac{L^2}{2LSp_{\text{flip}}} \tag{111}$$

$$= \frac{L}{2Sp_{\text{flip}}} \tag{112}$$

Using $p_{\text{flip}} \leq 2\exp(-\delta^2/(8\sigma^2))$:

$$\text{SNR}_{\text{vis}}^{\text{blur}} \geq \frac{L}{2S \cdot 2\exp(-\delta^2/(8\sigma^2))} \tag{113}$$

$$= \frac{L}{4S}\exp\left(\frac{\delta^2}{8\sigma^2}\right) \tag{114}$$

This means $\text{SNR}_{\text{vis}}^{\text{blur}} \geq (1/S) \cdot \text{SNR}_{\text{vis}}$ (unblurred).

For the relation to numerical SNR:

$$\text{SNR}_{\text{vis}}^{\text{blur}} \geq \frac{L}{4S}\exp\left(\frac{\delta^2}{8\sigma^2}\right) \tag{115}$$

$$\geq \frac{L}{4S}\exp\left(\frac{\delta^2}{8\sigma^2}\right)\frac{2\sigma^2}{A^2} \cdot \frac{A^2}{2\sigma^2} \tag{116}$$

$$= \frac{L\sigma^2}{2A^2S}\exp\left(\frac{\delta^2}{8\sigma^2}\right)\text{SNR}_{\text{num}} \tag{117}$$

Since $S < 1$, the factor $1/S > 1$ provides an amplification of the SNR compared to the unblurred case. $\quad\square$

# D    Additional Results of Computational Experiments

## D.1    Zero-shot Study

The full results of the zero-shot study are reported in Table 10 - Table 14. We also illustrate zero-shot TSF examples with prediction length equals 720 of the proposed ViTime versus TimesFM in Figure 19 - Figure 24. It is observable that ViTime consistently demonstrates superior zero-shot prediction performance compared to TimesFM across a range of rescale factors.

## D.2    Additional Zero-Shot Results on the GIFT-EVAL Benchmark

To rigorously assess cross-domain generalization under a truly zero-shot setup, we evaluate ViTime on the community-adopted GIFT-EVAL benchmark (Aksu et al., 2024), which provides a standardized protocol and strong baselines, including TimesFM (Das et al., 2024). We report Mean Absolute Scaled Error (MASE; lower is better) across short-, medium-, and long-horizon settings. TimesFM results are taken from the official GIFT-EVAL repository[1].

As shown in Table 15, ViTime achieves substantial gains on medium- and long-horizon forecasting, improving MASE by 38.3% and 28.9%, respectively. The short-horizon result is comparable but slightly weaker than TimesFM, which motivates a more careful analysis of potential training–test overlaps in baseline data.

**Leakage-controlled analysis.**   We hypothesize that short-horizon performance of TimesFM may benefit from latent overlap with the M4 dataset family, since its large-scale pre-training includes extensive Google Trends data that is plausibly correlated with M4 categories. Because M4 contributes substantially to short-horizon tasks within GIFT-EVAL, such overlap could compromise a strictly zero-shot comparison. To mitigate this risk, we conduct a leakage-controlled evaluation by excluding all M4 datasets from GIFT-EVAL and recomputing metrics for both models.

The controlled results in Table 16 are consistent: after excluding M4, ViTime outperforms TimesFM across short, medium, and long horizons. This substantiates ViTime's robustness under strict zero-shot conditions and clarifies the discrepancy on short-horizon tasks.

**Reproducibility.**   To facilitate transparency and reproducibility, we release complete evaluation logs and full GIFT-EVAL results in our repository[2].

## D.3    Probabilistic Forecasting Study

The complete results of the study on zero-shot probabilistic forecasting are detailed in Table 17 and Table 18. Furthermore, we present sample zero-shot TSF generated by our proposed ViTime model for a prediction horizon of 720 in Figure 25–Figure 36. These visualizations include both 90% and 50% prediction intervals. As shown in Figure 25–Figure 36, ViTime consistently demonstrates superior performance across a range of scaling factors.

## D.4    Fine-tuning study

Complete results of the fine-tuning study are reported in Table 19.

### D.4.1    Analysis of Fine-Tuning on Limited and Noisy Data

An intriguing observation from our fine-tuning experiments is that, under certain conditions, zero-shot (ZS) performance can surpass that of models fine-tuned on small data subsets. For instance, on the ETTh2 dataset, the ZS MAE for ViTime is 0.344, which is superior to the 0.370 MAE achieved after fine-tuning on 10% of the data. This counterintuitive phenomenon warrants a deeper investigation.

---

[1] https://github.com/SalesforceAIResearch/gift-eval/tree/main/results/timesfm
[2] https://github.com/IkeYang/ViTime

Table 10: Computational results for Scale = 0.5 (Landscape). The best results are **bolded**, the second best are underlined.

| Dataset | H | ViTime | | ViTime-TFM | | TimesFM | | Moment | | Moirai | | VisionTS | | PatchTST-ZS | |
|---|---|---|---|---|---|---|---|---|---|---|---|---|---|---|---|
| | | MSE | MAE | MSE | MAE | MSE | MAE | MSE | MAE | MSE | MAE | MSE | MAE | MSE | MAE |
| ETTh1 | 96 | 0.454 | 0.448 | 0.335 | 0.366 | **0.308** | **0.361** | 2.334 | 4.666 | 0.340 | 0.402 | 1.673 | 2.944 | 1.425 | 0.919 |
| | 192 | 0.437 | 0.449 | 0.368 | 0.384 | **0.335** | **0.383** | 2.349 | 4.781 | 0.450 | 0.475 | 1.675 | 3.050 | 1.519 | 0.953 |
| | 336 | 0.454 | 0.459 | 0.439 | 0.424 | **0.412** | **0.383** | 2.438 | 5.044 | 0.601 | 0.552 | 1.810 | 3.450 | 1.513 | 0.951 |
| | 720 | 0.555 | 0.510 | 0.639 | 0.534 | 0.617 | 0.560 | 2.601 | 5.506 | 1.575 | 0.880 | 2.055 | 4.109 | 1.521 | 0.952 |
| ETTh2 | 96 | **0.266** | **0.328** | 0.164 | 0.236 | 0.305 | 0.344 | 1.560 | 3.027 | 0.375 | 0.402 | 1.650 | 2.875 | 1.359 | 0.918 |
| | 192 | **0.288** | **0.348** | 0.173 | 0.243 | 0.333 | 0.377 | 1.615 | 3.171 | 0.338 | 0.400 | 1.642 | 2.987 | 1.407 | 0.930 |
| | 336 | **0.298** | **0.362** | 0.197 | 0.266 | 0.406 | 0.434 | 1.833 | 3.635 | 0.518 | 0.513 | 1.772 | 3.353 | 1.408 | 0.928 |
| | 720 | 0.399 | 0.435 | 0.242 | 0.312 | 0.585 | 0.548 | 2.196 | 4.503 | 0.825 | 0.647 | 2.027 | 4.023 | 1.409 | 0.929 |
| ETTm1 | 96 | 0.563 | 0.442 | 0.362 | 0.362 | 0.411 | 0.403 | 0.884 | 0.606 | 0.407 | 0.407 | 0.668 | **0.398** | 1.251 | 0.805 |
| | 192 | 0.571 | 0.450 | 0.399 | 0.382 | 0.447 | 0.427 | 0.869 | 0.597 | 0.370 | 0.393 | 0.622 | **0.382** | 1.311 | 0.823 |
| | 336 | 0.583 | 0.460 | 0.442 | 0.401 | 0.485 | 0.454 | 0.839 | 0.582 | 1.642 | 0.871 | 0.609 | **0.393** | 1.355 | 0.838 |
| | 720 | 0.595 | 0.475 | 0.462 | 0.417 | 0.537 | 0.493 | 0.786 | 0.565 | 1.182 | 0.771 | 0.622 | **0.433** | 1.335 | 0.831 |
| ETTm2 | 96 | 0.222 | 0.297 | 0.188 | 0.259 | 0.269 | 0.317 | 0.505 | 0.563 | 0.080 | 0.180 | 0.547 | 0.355 | 0.719 | 0.606 |
| | 192 | 0.280 | 0.336 | 0.234 | 0.291 | 0.342 | 0.357 | 0.531 | 0.598 | 0.093 | 0.197 | 0.531 | 0.345 | 0.818 | 0.639 |
| | 336 | 0.321 | 0.366 | 0.277 | 0.320 | 0.433 | 0.415 | 0.645 | 0.726 | 0.407 | 0.373 | 0.536 | **0.349** | 0.846 | 0.650 |
| | 720 | 0.395 | 0.411 | 0.335 | 0.362 | 0.522 | 0.483 | 0.738 | 0.824 | 0.303 | 0.342 | 0.564 | 0.367 | 0.840 | 0.647 |
| Traffic | 96 | 0.839 | 0.431 | 0.374 | 0.283 | **0.515** | 0.383 | 1.149 | 0.798 | 0.757 | 0.423 | 0.687 | **0.355** | 1.575 | 0.854 |
| | 192 | 0.724 | 0.405 | 0.362 | 0.274 | **0.539** | 0.402 | 1.195 | 0.805 | 0.732 | 0.451 | 0.666 | **0.303** | 1.544 | 0.870 |
| | 336 | 0.723 | 0.403 | 0.372 | 0.281 | **0.612** | 0.429 | 1.164 | 0.799 | 0.737 | 0.490 | 0.664 | **0.330** | 1.595 | 0.869 |
| | 720 | **0.693** | 0.409 | 0.423 | 0.306 | 0.795 | 0.505 | 1.163 | 0.804 | 2.459 | 1.204 | 0.691 | **0.342** | 1.529 | 0.867 |
| Weather | 96 | **0.172** | **0.212** | 0.197 | 0.242 | 0.209 | 0.253 | 0.538 | 0.332 | 0.367 | 0.360 | 0.537 | 0.301 | 0.871 | 0.597 |
| | 192 | **0.208** | **0.252** | 0.245 | 0.280 | 0.289 | 0.312 | 0.552 | 0.344 | 0.527 | 0.454 | 0.558 | 0.329 | 0.955 | 0.623 |
| | 336 | **0.272** | **0.297** | 0.295 | 0.312 | 0.365 | 0.360 | 0.567 | 0.350 | 1.065 | 0.619 | 0.589 | 0.367 | 0.954 | 0.620 |
| | 720 | 0.338 | 0.339 | 0.361 | 0.355 | 0.471 | 0.432 | 0.702 | 0.461 | 1.246 | 0.726 | 0.695 | 0.459 | 0.988 | 0.629 |
| Electricity | 96 | 0.267 | 0.339 | 0.193 | 0.277 | **0.184** | **0.279** | 0.951 | 0.795 | 0.323 | 0.372 | 0.533 | 0.390 | 1.131 | 0.822 |
| | 192 | 0.264 | 0.343 | 0.194 | 0.281 | **0.220** | **0.308** | 0.963 | 0.805 | 0.395 | 0.422 | 0.498 | 0.331 | 1.195 | 0.846 |
| | 336 | 0.289 | 0.363 | 0.211 | 0.296 | **0.277** | **0.347** | 0.980 | 0.819 | 0.575 | 0.505 | 0.549 | 0.374 | 1.201 | 0.848 |
| | 720 | **0.320** | **0.385** | 0.276 | 0.342 | 0.406 | 0.432 | 0.982 | 0.820 | 2.383 | 1.228 | 0.538 | 0.362 | 1.229 | 0.851 |

Table 11: Computational results for Scale = 0.66 (Landscape). The best results are **bolded**, the second best are underlined.

| Dataset | H | ViTime | | ViTime-TFM | | TimesFM | | Moment | | Moirai | | VisionTS | | PatchTST-ZS | |
|---|---|---|---|---|---|---|---|---|---|---|---|---|---|---|---|
| | | MSE | MAE | MSE | MAE | MSE | MAE | MSE | MAE | MSE | MAE | MSE | MAE | MSE | MAE |
| ETTh1 | 96 | 0.492 | 0.453 | **0.387** | **0.389** | 0.587 | 0.516 | 0.778 | 0.558 | 0.647 | 0.529 | 0.827 | 0.539 | 1.827 | 1.011 |
| | 192 | 0.485 | 0.458 | **0.408** | **0.405** | 0.655 | 0.555 | 0.799 | 0.576 | 0.747 | 0.593 | 0.820 | 0.540 | 1.953 | 1.044 |
| | 336 | 0.506 | 0.471 | **0.433** | **0.423** | 0.711 | 0.588 | 0.825 | 0.601 | 1.282 | 0.770 | 0.788 | 0.542 | 1.979 | 1.051 |
| | 720 | 0.553 | 0.503 | **0.552** | **0.492** | 0.798 | 0.645 | 0.878 | 0.656 | 1.106 | 0.742 | 0.840 | 0.605 | 1.978 | 1.051 |
| ETTh2 | 96 | 0.230 | 0.299 | **0.170** | **0.247** | 0.277 | 0.342 | 0.508 | 0.332 | 0.340 | 0.362 | 0.572 | 0.354 | 1.273 | 0.843 |
| | 192 | 0.254 | 0.319 | **0.194** | **0.268** | 0.344 | 0.394 | 0.527 | 0.353 | 0.292 | 0.357 | 0.583 | 0.373 | 1.330 | 0.863 |
| | 336 | 0.303 | 0.357 | **0.229** | **0.295** | 0.392 | 0.431 | 0.549 | 0.375 | 0.423 | 0.449 | 0.575 | 0.381 | 1.328 | 0.863 |
| | 720 | 0.372 | 0.414 | **0.266** | **0.328** | 0.508 | 0.509 | 0.672 | 0.472 | 0.554 | 0.516 | 0.656 | 0.450 | 1.328 | 0.863 |
| ETTm1 | 96 | 0.475 | 0.411 | **0.318** | **0.352** | 0.490 | 0.441 | 0.936 | 0.643 | 1.164 | 0.697 | 1.210 | 0.794 | 1.344 | 0.825 |
| | 192 | 0.522 | 0.433 | **0.357** | **0.374** | 0.502 | 0.461 | 0.924 | 0.636 | 1.239 | 0.700 | 0.978 | 0.659 | 1.463 | 0.860 |
| | 336 | 0.547 | 0.447 | **0.412** | **0.398** | 0.562 | 0.495 | 0.919 | 0.631 | 1.690 | 0.873 | 0.978 | 0.660 | 1.477 | 0.861 |
| | 720 | 0.596 | 0.475 | **0.461** | **0.425** | 0.669 | 0.551 | 0.871 | 0.602 | 1.248 | 0.753 | 0.959 | 0.642 | 1.449 | 0.853 |
| ETTm2 | 96 | 0.199 | 0.278 | 0.194 | 0.264 | 0.258 | 0.303 | 0.442 | 0.488 | **0.122** | **0.222** | 0.989 | 0.643 | 0.787 | 0.622 |
| | 192 | 0.263 | 0.320 | 0.249 | 0.301 | 0.384 | 0.362 | 0.513 | 0.567 | **0.139** | **0.238** | 0.835 | 0.543 | 0.848 | 0.642 |
| | 336 | 0.313 | 0.356 | **0.288** | **0.328** | 0.456 | 0.409 | 0.608 | 0.673 | 0.472 | 0.395 | 0.862 | 0.560 | 0.872 | 0.650 |
| | 720 | 0.376 | 0.397 | **0.345** | **0.369** | 0.531 | 0.474 | 0.708 | 0.784 | 0.373 | **0.347** | 0.871 | 0.566 | 0.866 | 0.651 |
| Traffic | 96 | 0.742 | 0.433 | **0.467** | **0.317** | 1.133 | 0.672 | 1.195 | 0.802 | 1.579 | 0.796 | 1.495 | 1.091 | 2.437 | 1.126 |
| | 192 | 0.837 | 0.426 | **0.466** | **0.317** | 1.279 | 0.737 | 1.180 | 0.802 | 1.618 | 0.834 | 1.591 | 1.172 | 2.570 | 1.148 |
| | 336 | 0.705 | 0.422 | **0.496** | **0.328** | 1.432 | 0.807 | 1.177 | 0.804 | 1.966 | 0.943 | 1.474 | 1.045 | 2.516 | 1.150 |
| | 720 | 0.743 | 0.438 | **0.523** | **0.341** | 1.542 | 0.865 | 1.214 | 0.810 | 1.748 | 0.863 | 1.273 | 0.852 | 2.593 | 1.153 |
| Weather | 96 | **0.164** | **0.199** | 0.175 | 0.221 | 0.182 | 0.233 | 0.537 | 0.374 | 0.315 | 0.363 | 0.460 | 0.291 | 0.800 | 0.562 |
| | 192 | **0.200** | **0.238** | 0.221 | 0.260 | 0.258 | 0.296 | 0.544 | 0.374 | 0.472 | 0.452 | 0.505 | 0.324 | 0.864 | 0.582 |
| | 336 | **0.251** | **0.281** | 0.272 | 0.295 | 0.317 | 0.341 | 0.560 | 0.369 | 0.952 | 0.643 | 0.543 | 0.345 | 0.892 | 0.590 |
| | 720 | **0.323** | **0.328** | 0.359 | 0.348 | 0.454 | 0.426 | 0.564 | 0.351 | 1.424 | 0.680 | 0.577 | 0.366 | 0.886 | 0.589 |
| Electricity | 96 | 0.247 | 0.334 | **0.203** | **0.294** | 0.602 | 0.599 | 0.928 | 0.774 | 1.026 | 0.802 | 1.195 | 0.985 | 1.858 | 1.067 |
| | 192 | 0.241 | 0.333 | **0.207** | **0.299** | 0.746 | 0.674 | 0.934 | 0.780 | 1.118 | 0.843 | 1.273 | 1.046 | 1.915 | 1.081 |
| | 336 | 0.257 | 0.348 | **0.221** | **0.310** | 0.903 | 0.752 | 0.952 | 0.797 | 2.213 | 1.036 | 1.168 | 0.951 | 1.967 | 1.093 |
| | 720 | 0.293 | 0.372 | **0.274** | **0.347** | 1.072 | 0.835 | 0.994 | 0.829 | 3.476 | 1.180 | 1.065 | 0.864 | 1.938 | 1.088 |

Table 12: Computational results for Scale = 1.0 (Landscape). The best results are **bolded**, the second best are underlined.

| Dataset | H | ViTime | | ViTime-TFM | | TimesFM | | Moment | | Moirai | | VisionTS | | PatchTST-ZS | |
|---|---|---|---|---|---|---|---|---|---|---|---|---|---|---|---|
| | | MSE | MAE | MSE | MAE | MSE | MAE | MSE | MAE | MSE | MAE | MSE | MAE | MSE | MAE |
| ETTh1 | 96 | 0.512 | 0.424 | 0.377 | **0.368** | 0.379 | 0.390 | 0.387 | 0.410 | **0.373** | 0.394 | 0.594 | 0.383 | 1.191 | 0.814 |
| | 192 | 0.536 | 0.438 | **0.392** | **0.378** | 0.429 | 0.416 | 0.397 | 0.420 | 0.454 | 0.443 | 0.626 | 0.410 | 1.254 | 0.837 |
| | 336 | 0.529 | 0.445 | **0.400** | **0.396** | 0.453 | 0.434 | 0.413 | 0.434 | 1.420 | 0.807 | 0.638 | 0.423 | 1.249 | 0.836 |
| | 720 | 0.603 | 0.489 | **0.424** | **0.406** | 0.506 | 0.479 | 0.454 | 0.472 | 1.299 | 0.828 | 0.637 | 0.441 | 1.254 | 0.837 |
| ETTh2 | 96 | 0.232 | 0.300 | 0.288 | 0.335 | 0.285 | 0.330 | 0.288 | 0.345 | **0.120** | **0.222** | 0.521 | 0.328 | 0.870 | 0.697 |
| | 192 | 0.264 | 0.327 | 0.308 | 0.343 | 0.329 | 0.366 | 0.306 | 0.359 | **0.132** | **0.243** | 0.573 | 0.367 | 0.916 | 0.716 |
| | 336 | **0.281** | **0.346** | 0.324 | 0.358 | 0.374 | 0.403 | 0.332 | 0.381 | 0.306 | 0.364 | 0.587 | 0.381 | 0.917 | 0.715 |
| | 720 | 0.359 | 0.402 | 0.364 | **0.365** | 0.438 | 0.456 | 0.403 | 0.439 | **0.332** | 0.399 | 0.623 | 0.422 | 0.909 | 0.711 |
| ETTm1 | 96 | 0.340 | 0.362 | 0.322 | **0.346** | 0.345 | 0.369 | **0.293** | 0.349 | 0.380 | 0.399 | 0.584 | 0.347 | 1.299 | 0.804 |
| | 192 | 0.375 | 0.381 | 0.351 | 0.362 | 0.405 | 0.406 | **0.310** | 0.359 | 0.564 | 0.471 | 0.600 | **0.360** | 1.375 | 0.831 |
| | 336 | 0.421 | 0.404 | 0.404 | 0.392 | 0.447 | 0.431 | **0.336** | 0.375 | 1.887 | 0.898 | 0.614 | **0.374** | 1.385 | 0.836 |
| | 720 | 0.499 | 0.445 | 0.451 | 0.408 | 0.501 | 0.470 | **0.405** | 0.416 | 1.300 | 0.745 | 0.645 | **0.405** | 1.365 | 0.830 |
| ETTm2 | 96 | 0.207 | 0.280 | 0.234 | 0.296 | 0.197 | 0.270 | 0.170 | 0.260 | **0.113** | **0.217** | 0.477 | 0.282 | 0.775 | 0.596 |
| | 192 | 0.273 | 0.322 | 0.270 | 0.304 | 0.300 | 0.327 | 0.200 | 0.280 | **0.162** | **0.256** | 0.512 | 0.305 | 0.867 | 0.633 |
| | 336 | 0.329 | 0.356 | 0.316 | 0.320 | 0.345 | 0.358 | **0.244** | **0.309** | 0.518 | 0.450 | 0.541 | 0.328 | 0.865 | 0.632 |
| | 720 | 0.421 | 0.409 | **0.360** | **0.327** | 0.470 | 0.433 | 0.363 | 0.387 | 0.480 | 0.401 | 0.586 | 0.370 | 0.849 | 0.626 |
| Traffic | 96 | 0.703 | 0.375 | **0.306** | **0.206** | 0.324 | 0.225 | 0.391 | 0.282 | 0.975 | 0.470 | 0.630 | 0.298 | 1.786 | 0.922 |
| | 192 | 0.708 | 0.375 | **0.320** | **0.213** | 0.338 | 0.233 | 0.400 | 0.286 | 0.598 | 0.329 | 0.641 | 0.256 | 1.911 | 0.955 |
| | 336 | 0.719 | 0.379 | **0.345** | **0.229** | 0.386 | 0.246 | 0.414 | 0.293 | 2.496 | 1.071 | 0.678 | 0.284 | 1.863 | 0.950 |
| | 720 | 0.790 | 0.416 | **0.357** | **0.236** | 0.428 | 0.276 | 0.450 | 0.310 | 2.055 | 0.949 | 0.711 | 0.299 | 1.933 | 0.953 |
| Weather | 96 | 0.196 | 0.217 | 0.181 | 0.221 | **0.121** | **0.166** | 0.154 | 0.209 | 0.351 | 0.402 | 0.469 | 0.257 | 0.840 | 0.566 |
| | 192 | 0.216 | 0.265 | 0.191 | 0.225 | **0.162** | **0.207** | 0.179 | 0.229 | 0.557 | 0.517 | 0.494 | 0.275 | 0.926 | 0.596 |
| | 336 | 0.314 | 0.311 | **0.215** | **0.233** | 0.247 | 0.275 | 0.216 | 0.258 | 0.830 | 0.610 | 0.529 | 0.299 | 0.914 | 0.588 |
| | 720 | 0.391 | 0.363 | **0.237** | **0.237** | 0.386 | 0.371 | 0.315 | 0.336 | 1.287 | 0.724 | 0.574 | 0.337 | 0.949 | 0.600 |
| Electricity | 96 | 0.164 | 0.257 | 0.118 | 0.209 | **0.109** | **0.209** | 0.138 | 0.242 | 0.336 | 0.366 | 0.421 | 0.266 | 1.223 | 0.858 |
| | 192 | 0.175 | 0.265 | **0.126** | 0.215 | 0.128 | 0.228 | 0.149 | 0.252 | 0.299 | 0.353 | 0.434 | 0.277 | 1.337 | 0.894 |
| | 336 | 0.194 | 0.279 | **0.139** | 0.227 | 0.157 | 0.252 | 0.166 | 0.266 | 1.599 | 0.920 | 0.455 | 0.296 | 1.334 | 0.892 |
| | 720 | 0.250 | 0.318 | **0.162** | **0.233** | 0.209 | 0.292 | 0.211 | 0.305 | 1.601 | 0.994 | 0.506 | 0.337 | 1.349 | 0.896 |

Table 13: Computational results for Scale = 1.5 (Landscape). The best results are **bolded**, the second best are underlined.

| Dataset | H | ViTime MSE | ViTime MAE | ViTime-TFM MSE | ViTime-TFM MAE | TimesFM MSE | TimesFM MAE | Moment MSE | Moment MAE | Moirai MSE | Moirai MAE | VisionTS MSE | VisionTS MAE | PatchTST-ZS MSE | PatchTST-ZS MAE |
|---|---|---|---|---|---|---|---|---|---|---|---|---|---|---|---|
| ETTh1 | 96 | 0.450 | **0.413** | 0.377 | 0.387 | **0.412** | 0.416 | 0.886 | 0.592 | 0.926 | 0.609 | 0.929 | 0.598 | 1.270 | 0.828 |
| | 192 | 0.494 | **0.435** | 0.419 | 0.407 | **0.451** | 0.446 | 0.909 | 0.607 | 1.083 | 0.683 | 0.959 | 0.632 | 1.362 | 0.864 |
| | 336 | 0.503 | **0.444** | 0.443 | 0.421 | 0.521 | 0.485 | 0.913 | 0.609 | 2.037 | 0.909 | 0.896 | 0.608 | 1.374 | 0.866 |
| | 720 | 0.535 | **0.463** | 0.445 | 0.428 | 0.614 | 0.534 | 0.867 | 0.584 | 1.089 | 0.721 | 0.863 | 0.574 | 1.387 | 0.871 |
| ETTh2 | 96 | 0.220 | 0.297 | 0.177 | 0.256 | 0.238 | 0.310 | 0.392 | **0.266** | 0.231 | 0.310 | 0.442 | 0.288 | 0.960 | 0.701 |
| | 192 | 0.269 | 0.334 | 0.206 | 0.275 | 0.291 | 0.339 | 0.366 | **0.252** | 0.236 | 0.331 | 0.424 | 0.285 | 1.137 | 0.770 |
| | 336 | 0.290 | 0.350 | 0.232 | 0.294 | 0.335 | 0.374 | 0.345 | **0.240** | 0.943 | 0.590 | 0.389 | 0.266 | 1.138 | 0.770 |
| | 720 | 0.329 | 0.379 | 0.273 | 0.325 | 0.413 | 0.434 | **0.336** | **0.235** | 0.462 | 0.456 | 0.361 | 0.243 | 1.177 | 0.785 |
| ETTm1 | 96 | 0.351 | **0.360** | 0.300 | 0.329 | 0.604 | 0.487 | 0.944 | 0.648 | 1.319 | 0.727 | 1.342 | 0.902 | 1.146 | 0.742 |
| | 192 | 0.430 | **0.398** | 0.329 | 0.350 | 0.622 | 0.496 | 0.930 | 0.638 | 1.545 | 0.788 | 1.069 | 0.726 | 1.257 | 0.775 |
| | 336 | 0.454 | **0.418** | 0.360 | 0.368 | 0.790 | 0.552 | 0.941 | 0.642 | 2.478 | 1.001 | 1.034 | 0.714 | 1.256 | 0.780 |
| | 720 | 0.524 | **0.451** | 0.405 | 0.394 | 0.927 | 0.604 | 0.917 | 0.630 | 1.502 | 0.805 | 1.030 | 0.702 | 1.270 | 0.782 |
| ETTm2 | 96 | **0.139** | **0.225** | 0.144 | 0.229 | 0.168 | 0.260 | 0.396 | 0.435 | 0.153 | 0.263 | 1.097 | 0.713 | 0.781 | 0.587 |
| | 192 | **0.180** | **0.266** | 0.187 | 0.258 | 0.223 | 0.299 | 0.493 | 0.543 | 0.266 | 0.339 | 0.912 | 0.593 | 0.808 | 0.598 |
| | 336 | **0.229** | **0.298** | 0.238 | 0.291 | 0.319 | 0.359 | 0.620 | 0.684 | 0.634 | 0.526 | 0.912 | 0.593 | 0.777 | 0.588 |
| | 720 | **0.301** | **0.346** | 0.290 | 0.328 | 0.392 | 0.409 | 0.652 | 0.721 | 0.631 | 0.475 | 0.935 | 0.608 | 0.808 | 0.600 |
| Traffic | 96 | **0.695** | **0.384** | 0.410 | 0.285 | 0.941 | 0.523 | 1.111 | 0.768 | 1.447 | 0.793 | 1.356 | 1.013 | 2.056 | 0.998 |
| | 192 | **0.630** | **0.371** | 0.416 | 0.281 | 0.996 | 0.557 | 1.060 | 0.754 | 1.543 | 0.843 | 1.476 | 1.081 | 2.186 | 1.033 |
| | 336 | **0.693** | **0.371** | 0.417 | 0.284 | 0.996 | 0.580 | 1.103 | 0.764 | 2.880 | 1.163 | 1.444 | 1.027 | 2.146 | 1.041 |
| | 720 | **0.746** | **0.418** | 0.438 | 0.294 | 1.249 | 0.678 | 1.114 | 0.771 | 2.095 | 1.014 | 1.206 | 0.859 | 2.217 | 1.042 |
| Weather | 96 | **0.120** | **0.141** | 0.157 | 0.192 | 0.176 | 0.216 | 0.494 | 0.341 | 0.381 | 0.408 | 0.420 | 0.249 | 0.837 | 0.549 |
| | 192 | **0.156** | **0.189** | 0.199 | 0.237 | 0.237 | 0.273 | 0.517 | 0.364 | 0.391 | 0.419 | 0.472 | 0.306 | 0.918 | 0.564 |
| | 336 | **0.200** | **0.231** | 0.252 | 0.283 | 0.297 | 0.328 | 0.568 | 0.406 | 0.628 | 0.489 | 0.526 | 0.359 | 0.941 | 0.574 |
| | 720 | **0.267** | **0.286** | 0.339 | 0.344 | 0.408 | 0.406 | 0.668 | 0.494 | 0.701 | 0.570 | 0.637 | 0.460 | 0.931 | 0.573 |
| Electricity | 96 | **0.171** | **0.262** | 0.148 | 0.246 | 0.227 | 0.306 | 0.796 | 0.662 | 0.767 | 0.681 | 1.071 | 0.876 | 1.294 | 0.889 |
| | 192 | **0.172** | **0.266** | 0.156 | 0.251 | 0.254 | 0.333 | 0.799 | 0.667 | 0.871 | 0.729 | 1.148 | 0.937 | 1.412 | 0.924 |
| | 336 | **0.186** | **0.275** | 0.168 | 0.260 | 0.283 | 0.358 | 0.816 | 0.682 | 1.575 | 0.915 | 1.114 | 0.916 | 1.396 | 0.918 |
| | 720 | **0.223** | **0.302** | 0.200 | 0.285 | 0.393 | 0.438 | 0.883 | 0.733 | 1.320 | 0.891 | 0.991 | 0.813 | 1.425 | 0.928 |

Table 14: Computational results for Scale = 2.0 (Landscape). The best results are **bolded**, the second best are underlined.

| Dataset | H | ViTime | | ViTime-TFM | | TimesFM | | Moment | | Moirai | | VisionTS | | PatchTST-ZS | |
|---|---|---|---|---|---|---|---|---|---|---|---|---|---|---|---|
| | | MSE | MAE | MSE | MAE | MSE | MAE | MSE | MAE | MSE | MAE | MSE | MAE | MSE | MAE |
| ETTh1 | 96 | 0.521 | 0.429 | **0.352** | **0.362** | 0.358 | 0.384 | 0.791 | 0.543 | 0.674 | 0.444 | 0.834 | 0.537 | 1.279 | 0.818 |
| | 192 | 0.541 | 0.446 | **0.388** | **0.381** | 0.406 | 0.415 | 0.820 | 0.557 | 0.855 | 0.564 | 0.843 | 0.557 | 1.366 | 0.844 |
| | 336 | 0.537 | 0.449 | **0.431** | **0.400** | 0.454 | 0.444 | 0.855 | 0.576 | 1.427 | 0.928 | 0.823 | 0.558 | 1.420 | 0.856 |
| | 720 | 0.579 | 0.474 | **0.458** | **0.418** | 0.581 | 0.508 | 0.909 | 0.605 | 1.562 | 1.052 | 0.891 | 0.575 | 1.419 | 0.860 |
| ETTh2 | 96 | 0.210 | 0.293 | **0.187** | **0.258** | 0.219 | 0.295 | 0.450 | 0.307 | 0.602 | 0.387 | 0.500 | 0.329 | 0.701 | 0.601 |
| | 192 | 0.258 | 0.326 | **0.227** | **0.285** | 0.283 | 0.336 | 0.447 | 0.307 | 0.629 | 0.426 | 0.506 | 0.347 | 0.786 | 0.632 |
| | 336 | 0.300 | 0.356 | **0.253** | 0.304 | 0.316 | 0.363 | 0.433 | **0.298** | 0.855 | 0.561 | 0.471 | 0.327 | 0.789 | 0.632 |
| | 720 | 0.367 | 0.401 | **0.298** | 0.335 | 0.415 | 0.432 | 0.385 | **0.269** | 0.983 | 0.693 | 0.411 | 0.283 | 0.789 | 0.629 |
| ETTm1 | 96 | 0.308 | 0.342 | **0.286** | **0.324** | 1.029 | 0.569 | 0.955 | 0.662 | 1.311 | 0.822 | 1.286 | 0.837 | 1.037 | 0.690 |
| | 192 | 0.408 | 0.395 | **0.306** | **0.342** | 1.068 | 0.605 | 0.938 | 0.650 | 1.250 | 0.800 | 0.987 | 0.647 | 1.139 | 0.719 |
| | 336 | 0.420 | 0.407 | **0.342** | **0.366** | 1.245 | 0.655 | 0.946 | 0.650 | 1.539 | 0.981 | 1.012 | 0.660 | 1.173 | 0.729 |
| | 720 | 0.500 | 0.448 | **0.395** | **0.395** | 1.329 | 0.700 | 0.947 | 0.641 | 1.651 | 1.086 | 1.007 | 0.634 | 1.219 | 0.743 |
| ETTm2 | 96 | **0.115** | **0.206** | 0.137 | 0.219 | 0.178 | 0.268 | 0.489 | 0.540 | 0.489 | 0.333 | 1.052 | 0.684 | 0.686 | 0.555 |
| | 192 | **0.158** | **0.244** | 0.167 | 0.244 | 0.223 | 0.304 | 0.502 | 0.553 | 0.582 | 0.391 | 0.842 | 0.547 | 0.775 | 0.582 |
| | 336 | **0.201** | 0.278 | 0.212 | **0.276** | 0.285 | 0.346 | 0.565 | 0.623 | 0.763 | 0.516 | 0.894 | 0.581 | 0.730 | 0.567 |
| | 720 | 0.282 | 0.333 | **0.273** | **0.318** | 0.408 | 0.417 | 0.649 | 0.718 | 0.835 | 0.567 | 0.915 | 0.595 | 0.784 | 0.586 |
| Traffic | 96 | 0.766 | 0.392 | **0.412** | **0.268** | 0.943 | 0.492 | 0.990 | 0.522 | 0.990 | 0.522 | 1.420 | 1.010 | 1.966 | 0.977 |
| | 192 | 0.719 | 0.390 | **0.412** | **0.265** | 0.933 | 0.547 | 0.911 | 0.497 | 0.911 | 0.497 | 1.570 | 1.128 | 2.157 | 1.035 |
| | 336 | 0.670 | 0.375 | **0.407** | **0.268** | 0.957 | 0.561 | 1.071 | 0.751 | 1.823 | 1.206 | 1.317 | 0.892 | 2.236 | 1.045 |
| | 720 | 0.765 | 0.387 | **0.428** | **0.282** | 1.145 | 0.656 | 1.092 | 0.758 | 1.939 | 1.352 | 1.270 | 0.837 | 2.256 | 1.050 |
| Weather | 96 | **0.120** | **0.133** | 0.145 | 0.182 | 0.167 | 0.205 | 0.413 | 0.282 | 0.600 | 0.371 | 0.326 | 0.194 | 0.836 | 0.534 |
| | 192 | **0.136** | **0.171** | 0.188 | 0.230 | 0.231 | 0.276 | 0.439 | 0.301 | 0.539 | 0.330 | 0.417 | 0.263 | 0.948 | 0.578 |
| | 336 | **0.183** | **0.212** | 0.216 | 0.257 | 0.291 | 0.331 | 0.463 | 0.314 | 0.799 | 0.427 | 0.468 | 0.305 | 0.992 | 0.588 |
| | 720 | **0.254** | **0.270** | 0.278 | 0.308 | 0.413 | 0.419 | 0.540 | 0.373 | 0.885 | 0.529 | 0.553 | 0.375 | 0.974 | 0.582 |
| Electricity | 96 | 0.181 | 0.266 | **0.146** | **0.234** | 0.223 | 0.299 | 0.798 | 0.654 | 0.565 | 0.376 | 1.034 | 0.863 | 1.177 | 0.848 |
| | 192 | 0.188 | 0.274 | **0.155** | **0.242** | 0.263 | 0.334 | 0.796 | 0.651 | 0.592 | 0.416 | 1.167 | 0.956 | 1.270 | 0.880 |
| | 336 | 0.198 | 0.282 | **0.161** | **0.249** | 0.285 | 0.364 | 0.798 | 0.654 | 1.275 | 0.950 | 0.977 | 0.764 | 1.318 | 0.896 |
| | 720 | 0.216 | 0.296 | **0.193** | **0.275** | 0.399 | 0.447 | 0.805 | 0.666 | 1.362 | 1.037 | 0.890 | 0.715 | 1.315 | 0.895 |

Table 15: Zero-shot performance on the full GIFT-EVAL benchmark (MASE; lower is better). TimesFM results from the official repository.

| Forecast Horizon | TimesFM (MASE) | ViTime (MASE) | Relative Improvement |
|---|---|---|---|
| Long | 2.3706 | **1.4636** | **38.3%** |
| Medium | 1.9621 | **1.3939** | **28.9%** |
| Short | **2.3548** | 2.4557 | −4.3% |

Table 16: Zero-shot performance on GIFT-EVAL with M4 datasets excluded (MASE; lower is better).

| Forecast Horizon | TimesFM (MASE) | ViTime (MASE) | Relative Improvement |
|---|---|---|---|
| Long | 2.3706 | **1.4636** | **38.3%** |
| Medium | 1.9621 | **1.3939** | **28.9%** |
| Short | 2.6148 | **2.5687** | **1.8%** |

We hypothesize that this performance degradation is attributed to a combination of data-specific challenges and the behavior of full-parameter fine-tuning.

- **Data Characteristics**: The ETT datasets, particularly ETTh2, present significant challenges with their low signal-to-noise ratio. In long-horizon forecasting tasks (e.g., 720 steps), while the data shows clear seasonal cycles, it lacks other predictable patterns that models can reliably use. Most variations outside of seasonality appear random or too noisy for effective learning.

- **Overfitting and Forgetting**: When fine-tuning the entire model on a small (e.g., 10%) and potentially non-representative slice of the data, the model is prone to overfitting to short-term noise and atypical seasonal variations presenting in that specific subset. This process can lead to a form of catastrophic forgetting, where the robust and generalized seasonal priors learned during large-scale pre-training are partially overwritten.

To validate this hypothesis, we conducted an extended analysis on the ETTh2 dataset, comparing three different fine-tuning strategies across varying data proportions, full fine-tuning (Full FT), as well as parameter-efficient fine-tuning (PEFT) using LoRA with 10% and 1% of trainable parameters. The results are visualized in Figure 17.

As depicted in Figure 17, the performance of full fine-tuning exhibits a "U-shaped" trajectory: the MAE first increases, surpassing the ZS baseline, and then gradually decreases as more data becomes available, eventually outperforming the ZS model only when fine-tuned on the full dataset. This confirms our hypothesis that full fine-tuning on limited data is detrimental.

In stark contrast, the LoRA-based methods significantly alleviate this issue. The 10% LoRA strategy shows a much smaller initial performance dip, while the 1% LoRA strategy consistently improves upon or matches the ZS baseline across all data ratios. This demonstrates that by updating only a small fraction of the parameters, PEFT methods are less susceptible to overfitting on noisy - small-scale data and are more effective at preserving the valuable priors acquired during pre-training.

This analysis not only explains the initially counterintuitive results but also yields a crucial insight: for adapting large pre-trained models to downstream tasks with limited or noisy data, parameter-efficient fine-tuning methods like LoRA represent a more robust and reliable strategy than conventional full-model fine-tuning.

### D.5 Robust Inference Study

Complete results of the robust inference study are reported in Table 20-Table 22.

Table 17: CRPS Performance Comparison (Part 1)

| Dataset | H | ScaleValue=0.5 | | | ScaleValue=0.66 | | | ScaleValue=1.0 | | |
|---|---|---|---|---|---|---|---|---|---|---|
| | | ViTime | Moirai | Lag-Llama | ViTime | Moirai | Lag-Llama | ViTime | Moirai | Lag-Llama |
| ETTh1 | 96 | 0.321 | **0.295** | 0.452 | **0.321** | 0.399 | 0.456 | 0.332 | **0.288** | 0.336 |
| | 192 | **0.338** | 0.348 | 0.474 | **0.342** | 0.451 | 0.486 | 0.346 | **0.325** | 0.392 |
| | 336 | **0.381** | 0.455 | 0.500 | **0.366** | 0.697 | 0.499 | **0.353** | 0.754 | 0.501 |
| | 720 | **0.462** | 0.640 | 0.613 | **0.431** | 0.631 | 0.592 | **0.392** | 0.659 | 0.537 |
| | Avg | **0.376** | 0.434 | 0.510 | **0.365** | 0.544 | 0.508 | **0.356** | 0.506 | 0.441 |
| ETTh2 | 96 | **0.278** | 0.301 | 0.400 | **0.259** | 0.275 | 0.374 | 0.268 | **0.164** | 0.314 |
| | 192 | **0.302** | 0.322 | 0.444 | 0.302 | **0.280** | 0.444 | 0.300 | **0.189** | 0.333 |
| | 336 | **0.362** | 0.423 | 0.514 | **0.341** | 0.418 | 0.513 | **0.322** | 0.386 | 0.450 |
| | 720 | **0.485** | 0.566 | 0.568 | **0.426** | 0.444 | 0.545 | 0.386 | **0.357** | 0.507 |
| | Avg | **0.357** | 0.403 | 0.481 | **0.332** | 0.354 | 0.469 | 0.319 | **0.274** | 0.401 |
| ETTm1 | 96 | 0.319 | **0.303** | 0.457 | **0.317** | 0.509 | 0.435 | 0.313 | **0.295** | 0.366 |
| | 192 | 0.338 | **0.299** | 0.473 | **0.339** | 0.522 | 0.475 | **0.330** | 0.356 | 0.412 |
| | 336 | **0.357** | 0.855 | 0.513 | **0.363** | 0.847 | 0.502 | **0.349** | 0.855 | 0.459 |
| | 720 | **0.384** | 0.687 | 0.538 | **0.396** | 0.674 | 0.511 | **0.384** | 0.648 | 0.495 |
| | Avg | **0.350** | 0.536 | 0.495 | **0.354** | 0.638 | 0.481 | **0.344** | 0.538 | 0.435 |
| ETTm2 | 96 | 0.248 | **0.135** | 0.401 | 0.240 | **0.167** | 0.361 | 0.224 | **0.159** | 0.345 |
| | 192 | 0.291 | **0.153** | 0.448 | 0.286 | **0.193** | 0.431 | 0.266 | **0.211** | 0.403 |
| | 336 | **0.329** | 0.380 | 0.500 | **0.323** | 0.415 | 0.477 | **0.301** | 0.455 | 0.484 |
| | 720 | 0.374 | **0.330** | 0.517 | 0.379 | **0.378** | 0.516 | **0.352** | 0.410 | 0.495 |
| | Avg | 0.310 | **0.249** | 0.467 | 0.307 | **0.288** | 0.446 | **0.286** | 0.309 | 0.432 |
| Electricity | 96 | **0.262** | 0.285 | 0.497 | **0.252** | 0.557 | 0.495 | **0.243** | 0.284 | 0.318 |
| | 192 | **0.275** | 0.534 | 0.544 | **0.262** | 0.590 | 0.507 | **0.252** | 0.271 | 0.378 |
| | 336 | **0.297** | 0.424 | 0.595 | **0.279** | 0.896 | 0.556 | **0.266** | 0.800 | 0.543 |
| | 720 | **0.350** | 0.931 | 0.708 | **0.335** | 0.994 | 0.681 | **0.307** | 0.731 | 0.631 |
| | Avg | **0.296** | 0.543 | 0.586 | **0.282** | 0.759 | 0.560 | **0.286** | 0.521 | 0.467 |
| Traffic | 96 | **0.300** | 0.319 | 0.611 | **0.307** | 0.578 | 0.618 | 0.314 | **0.390** | 0.364 |
| | 192 | **0.293** | 0.340 | 0.588 | **0.310** | 0.609 | 0.595 | 0.318 | **0.262** | 0.446 |
| | 336 | **0.304** | 0.385 | 0.614 | **0.328** | 0.798 | 0.670 | **0.322** | 1.045 | 0.658 |
| | 720 | **0.338** | 0.928 | 0.698 | **0.368** | 0.713 | 0.679 | **0.354** | 0.808 | 0.724 |
| | Avg | **0.309** | 0.493 | 0.628 | **0.328** | 0.674 | 0.641 | **0.327** | 0.626 | 0.548 |
| Weather | 96 | **0.199** | 0.278 | 0.331 | **0.196** | 0.276 | 0.312 | **0.177** | 0.316 | 0.289 |
| | 192 | **0.246** | 0.359 | 0.410 | **0.248** | 0.352 | 0.399 | **0.222** | 0.417 | 0.363 |
| | 336 | **0.281** | 0.630 | 0.457 | **0.286** | 0.600 | 0.431 | **0.264** | 0.596 | 0.418 |
| | 720 | **0.327** | 0.721 | 0.510 | **0.329** | 0.631 | 0.501 | **0.312** | 0.696 | 0.507 |
| | Avg | **0.263** | 0.497 | 0.427 | **0.265** | 0.465 | 0.411 | **0.244** | 0.506 | 0.394 |

Table 18: CRPS Performance Comparison (Part 2)

| Dataset | H | ScaleValue=1.5 | | | ScaleValue=2.0 | | |
|---|---|---|---|---|---|---|---|
| | | ViTime | Moirai | Lag-Llama | ViTime | Moirai | Lag-Llama |
| ETTh1 | 96 | **0.316** | 0.446 | 0.438 | 0.314 | **0.310** | 0.351 |
| | 192 | **0.335** | 0.508 | 0.453 | **0.336** | 0.398 | 0.456 |
| | 336 | **0.352** | 0.836 | 0.500 | **0.357** | 0.907 | 0.488 |
| | 720 | **0.375** | 0.624 | 0.508 | **0.390** | 0.763 | 0.544 |
| | Avg | **0.345** | 0.604 | 0.475 | **0.349** | 0.595 | 0.460 |
| ETTh2 | 96 | 0.245 | **0.236** | 0.364 | **0.242** | 0.265 | 0.371 |
| | 192 | 0.272 | **0.258** | 0.407 | **0.282** | 0.300 | 0.417 |
| | 336 | **0.294** | 0.643 | 0.443 | **0.312** | 0.620 | 0.441 |
| | 720 | **0.330** | 0.542 | 0.498 | **0.344** | 0.657 | 0.492 |
| | Avg | **0.285** | 0.420 | 0.428 | **0.295** | 0.461 | 0.430 |
| ETTm1 | 96 | **0.305** | 0.552 | 0.419 | **0.303** | 0.564 | 0.418 |
| | 192 | **0.327** | 0.599 | 0.442 | **0.323** | 0.553 | 0.446 |
| | 336 | **0.348** | 0.970 | 0.457 | **0.349** | 0.939 | 0.473 |
| | 720 | **0.390** | 0.774 | 0.478 | **0.391** | 0.824 | 0.490 |
| | Avg | **0.343** | 0.724 | 0.449 | **0.342** | 0.720 | 0.457 |
| ETTm2 | 96 | 0.198 | **0.197** | 0.312 | **0.196** | 0.266 | 0.248 |
| | 192 | **0.237** | 0.264 | 0.357 | **0.232** | 0.291 | 0.359 |
| | 336 | **0.277** | 0.549 | 0.422 | **0.270** | 0.523 | 0.400 |
| | 720 | **0.326** | 0.544 | 0.471 | **0.318** | 0.567 | 0.458 |
| | Avg | **0.260** | 0.389 | 0.391 | **0.254** | 0.412 | 0.366 |
| Electricity | 96 | **0.216** | 0.482 | 0.410 | **0.227** | 0.272 | 0.313 |
| | 192 | **0.224** | 0.587 | 0.454 | **0.237** | 0.328 | 0.392 |
| | 336 | **0.240** | 0.867 | 0.478 | **0.248** | 0.910 | 0.519 |
| | 720 | **0.283** | 0.675 | 0.559 | **0.294** | 0.772 | 0.693 |
| | Avg | **0.241** | 0.653 | 0.475 | **0.252** | 0.571 | 0.479 |
| Traffic | 96 | **0.316** | 0.587 | 0.658 | **0.319** | 0.373 | 0.673 |
| | 192 | **0.307** | 0.647 | 0.653 | **0.322** | 0.391 | 0.695 |
| | 336 | **0.325** | 1.153 | 0.693 | **0.314** | 1.215 | 0.625 |
| | 720 | **0.366** | 0.899 | 0.612 | **0.353** | 1.248 | 0.708 |
| | Avg | **0.329** | 0.822 | 0.654 | **0.327** | 0.807 | 0.675 |
| Weather | 96 | **0.150** | 0.380 | 0.254 | **0.142** | 0.345 | 0.244 |
| | 192 | **0.200** | 0.467 | 0.306 | **0.186** | 3.237 | 0.273 |
| | 336 | **0.246** | 0.501 | 0.389 | **0.223** | 0.453 | 0.328 |
| | 720 | **0.301** | 0.610 | 0.467 | **0.276** | 0.535 | 0.355 |
| | Avg | **0.224** | 0.489 | 0.354 | **0.207** | 1.143 | 0.300 |

Table 19: MAE of different methods in fine-tuning study.

| Dataset | H | VITIME (FT) 10% | VITIME (FT) 100% | TimesFM (FT) 10% | GPT4TS (FT) 10% | TIME-LLM (FT) 10% | PatchTST 10% | SiMBA 100% | TIMESNET 100% | PatchTST 100% | iTransformer 100% | TimeMixer 100% |
|---|---|---|---|---|---|---|---|---|---|---|---|---|
| ETTh1 | 96 | 0.3938 ± 0.053 | **0.3814 ± 0.057** | 0.398 | 0.485 | 0.460 | 0.485 | 0.395 | 0.402 | 0.400 | 0.405 | 0.390 |
|  | 192 | 0.4098 ± 0.039 | **0.3978 ± 0.050** | 0.424 | 0.524 | 0.483 | 0.524 | 0.424 | 0.429 | 0.429 | 0.436 | 0.414 |
|  | 336 | 0.4256 ± 0.031 | **0.4076 ± 0.040** | 0.436 | 0.550 | 0.540 | 0.550 | 0.443 | 0.469 | 0.440 | 0.458 | 0.429 |
|  | 720 | 0.4578 ± 0.012 | **0.4358 ± 0.007** | 0.445 | 0.610 | 0.604 | 0.610 | 0.469 | 0.500 | 0.468 | 0.491 | 0.460 |
|  | Avg | 0.4218 | **0.4057** | 0.426 | 0.542 | 0.522 | 0.542 | 0.433 | 0.450 | 0.434 | 0.448 | 0.423 |
| ETTh2 | 96 | 0.3230 ± 0.010 | **0.2970 ± 0.002** | 0.356 | 0.389 | 0.326 | 0.389 | 0.339 | 0.374 | 0.337 | 0.349 | 0.330 |
|  | 192 | 0.3520 ± 0.003 | **0.3270 ± 0.003** | 0.400 | 0.414 | 0.373 | 0.414 | 0.390 | 0.414 | 0.382 | 0.400 | 0.402 |
|  | 336 | 0.3750 ± 0.005 | **0.3450 ± 0.004** | 0.428 | 0.441 | 0.429 | 0.441 | 0.406 | 0.452 | 0.384 | 0.432 | 0.396 |
|  | 720 | 0.4290 ± 0.008 | **0.4060 ± 0.004** | 0.457 | 0.480 | 0.449 | 0.480 | 0.431 | 0.468 | 0.422 | 0.445 | 0.408 |
|  | Avg | 0.3698 | **0.3438** | 0.410 | 0.431 | 0.394 | 0.431 | 0.392 | 0.427 | 0.381 | 0.407 | 0.384 |
| ETTm2 | 96 | 0.2579 ± 0.004 | **0.2328 ± 0.012** | 0.263 | 0.274 | 0.261 | 0.274 | 0.263 | 0.267 | 0.256 | 0.264 | 0.254 |
|  | 192 | 0.2896 ± 0.006 | **0.2750 ± 0.017** | 0.309 | 0.317 | 0.314 | 0.317 | 0.306 | 0.309 | 0.296 | 0.309 | 0.295 |
|  | 336 | 0.3224 ± 0.016 | **0.3108 ± 0.024** | 0.349 | 0.353 | 0.327 | 0.353 | 0.343 | 0.351 | 0.329 | 0.348 | 0.330 |
|  | 720 | 0.3779 ± 0.009 | **0.3696 ± 0.017** | 0.415 | 0.427 | 0.390 | 0.427 | 0.399 | 0.403 | 0.385 | 0.407 | 0.383 |
|  | Avg | 0.3120 | **0.2970** | 0.334 | 0.343 | 0.323 | 0.343 | 0.328 | 0.333 | 0.317 | 0.332 | 0.316 |
| ETTm1 | 96 | 0.3393 ± 0.001 | **0.3319 ± 0.004** | 0.345 | 0.419 | 0.388 | 0.419 | 0.360 | 0.375 | 0.346 | 0.368 | 0.340 |
|  | 192 | 0.3624 ± 0.002 | **0.3518 ± 0.003** | 0.374 | 0.434 | 0.416 | 0.434 | 0.382 | 0.387 | 0.370 | 0.391 | 0.365 |
|  | 336 | 0.3839 ± 0.002 | **0.3702 ± 0.002** | 0.397 | 0.454 | 0.426 | 0.454 | 0.405 | 0.411 | 0.392 | 0.420 | 0.381 |
|  | 720 | 0.4189 ± 0.007 | **0.4083 ± 0.002** | 0.436 | 0.556 | 0.476 | 0.556 | 0.437 | 0.450 | 0.420 | 0.459 | 0.417 |
|  | Avg | 0.3761 | **0.3656** | 0.388 | 0.466 | 0.426 | 0.466 | 0.396 | 0.406 | 0.382 | 0.410 | 0.376 |
| Traffic | 96 | 0.2446 ± 0.013 | **0.2336 ± 0.008** |  | Not Reported |  | 0.268 | 0.268 | 0.321 | 0.249 | 0.268 | 0.249 |
|  | 192 | 0.2444 ± 0.005 | **0.2376 ± 0.004** |  | Not Reported |  | 0.274 | 0.317 | 0.336 | 0.256 | 0.276 | 0.250 |
|  | 336 | 0.2472 ± 0.002 | **0.2437 ± 0.002** |  | Not Reported |  | 0.282 | 0.284 | 0.336 | 0.264 | 0.283 | 0.270 |
|  | 720 | **0.2670 ± 0.005** | 0.2775 ± 0.005 |  | Not Reported |  | 0.319 | 0.297 | 0.350 | 0.286 | 0.302 | 0.281 |
|  | Avg | 0.2508 | **0.2481** |  | Not Reported |  | 0.286 | 0.291 | 0.336 | 0.264 | 0.282 | 0.263 |
| Weather | 96 | 0.1865 ± 0.004 | **0.1827 ± 0.002** |  | Not Reported |  | 0.221 | 0.219 | 0.220 | 0.198 | 0.214 | 0.197 |
|  | 192 | 0.2281 ± 0.007 | **0.2234 ± 0.001** |  | Not Reported |  | 0.261 | 0.260 | 0.261 | 0.241 | 0.254 | 0.239 |
|  | 336 | 0.2696 ± 0.006 | **0.2686 ± 0.003** |  | Not Reported |  | 0.300 | 0.297 | 0.306 | 0.282 | 0.296 | 0.280 |
|  | 720 | **0.3126 ± 0.010** | 0.3198 ± 0.001 |  | Not Reported |  | 0.351 | 0.349 | 0.359 | 0.334 | 0.347 | 0.330 |
|  | Avg | 0.2492 | **0.2486** |  | Not Reported |  | 0.283 | 0.281 | 0.286 | 0.264 | 0.278 | 0.262 |
| Electricity | 96 | 0.2275 ± 0.012 | 0.2222 ± 0.005 |  | Not Reported |  | 0.235 | 0.253 | 0.272 | **0.222** | 0.240 | 0.224 |
|  | 192 | 0.2350 ± 0.009 | 0.2339 ± 0.004 |  | Not Reported |  | 0.250 | 0.262 | 0.289 | 0.240 | 0.253 | **0.220** |
|  | 336 | 0.2522 ± 0.004 | **0.2476 ± 0.005** |  | Not Reported |  | 0.270 | 0.277 | 0.300 | 0.259 | 0.269 | 0.255 |
|  | 720 | 0.2867 ± 0.002 | **0.2777 ± 0.003** |  | Not Reported |  | 0.315 | 0.305 | 0.320 | 0.290 | 0.317 | 0.287 |
|  | Avg | 0.2504 | **0.2454** |  | Not Reported |  | 0.268 | 0.274 | 0.295 | 0.253 | 0.270 | 0.246 |

Table 20: Performance comparison of TimesFM and ViTime: Gaussian noise levels 0.1 and 0.3

| Dataset | H | TimesFM GN(0.1) | | ViTime GN(0.1) | | TimesFM GN(0.3) | | ViTime GN(0.3) | |
|---|---|---|---|---|---|---|---|---|---|
| | | MSE | MAE | MSE | MAE | MSE | MAE | MSE | MAE |
| ETTh1 | 96 | **0.411±0.003** | 0.415±0.002 | 0.423±0.005 | 0.411±0.002 | **0.419±0.007** | **0.419±0.004** | 0.427±0.009 | 0.419±0.006 |
| | 192 | 0.461±0.003 | 0.445±0.002 | **0.452±0.006** | **0.429±0.003** | 0.464±0.015 | 0.447±0.005 | **0.453±0.011** | **0.435±0.006** |
| | 336 | 0.513±0.004 | 0.478±0.003 | **0.484±0.003** | **0.452±0.002** | 0.515±0.028 | 0.481±0.007 | **0.485±0.008** | **0.458±0.005** |
| | 720 | 0.617±0.008 | 0.544±0.002 | **0.563±0.005** | **0.503±0.001** | 0.618±0.066 | 0.546±0.009 | **0.568±0.007** | **0.509±0.002** |
| | Mean | 0.500 | 0.471 | **0.480** | **0.449** | 0.504 | 0.473 | **0.483** | **0.455** |
| ETTh2 | 96 | 0.268±0.000 | 0.326±0.001 | **0.238±0.001** | **0.304±0.001** | 0.261±0.001 | 0.325±0.002 | **0.238±0.002** | **0.307±0.002** |
| | 192 | 0.326±0.000 | 0.367±0.001 | **0.277±0.001** | **0.336±0.000** | 0.312±0.001 | 0.364±0.002 | **0.279±0.001** | **0.340±0.001** |
| | 336 | 0.377±0.000 | 0.405±0.001 | **0.318±0.001** | **0.368±0.001** | 0.363±0.001 | 0.402±0.001 | **0.321±0.002** | **0.372±0.001** |
| | 720 | 0.480±0.000 | 0.475±0.001 | **0.414±0.001** | **0.434±0.001** | 0.464±0.001 | 0.470±0.001 | **0.411±0.001** | **0.433±0.001** |
| | Mean | 0.363 | 0.393 | **0.312** | **0.360** | 0.350 | 0.390 | **0.312** | **0.363** |
| ETTm1 | 96 | 0.519±0.025 | 0.444±0.009 | **0.407±0.003** | **0.394±0.001** | 0.519±0.027 | 0.439±0.007 | **0.404±0.004** | **0.400±0.003** |
| | 192 | 0.569±0.051 | 0.475±0.011 | **0.442±0.002** | **0.412±0.001** | 0.557±0.021 | 0.466±0.008 | **0.436±0.003** | **0.416±0.002** |
| | 336 | 0.641±0.046 | 0.509±0.008 | **0.489±0.002** | **0.437±0.001** | 0.619±0.071 | 0.497±0.015 | **0.482±0.003** | **0.440±0.002** |
| | 720 | 0.720±0.123 | 0.551±0.017 | **0.557±0.003** | **0.474±0.001** | 0.680±0.044 | 0.536±0.008 | **0.557±0.004** | **0.479±0.002** |
| | Mean | 0.612 | 0.495 | **0.474** | **0.429** | 0.594 | 0.485 | **0.470** | **0.434** |
| ETTm2 | 96 | 0.211±0.005 | 0.281±0.004 | **0.194±0.003** | **0.270±0.003** | 0.199±0.004 | 0.278±0.003 | **0.197±0.005** | **0.276±0.004** |
| | 192 | 0.289±0.013 | 0.328±0.005 | **0.251±0.002** | **0.309±0.002** | 0.264±0.006 | 0.320±0.004 | **0.255±0.004** | **0.315±0.004** |
| | 336 | 0.357±0.015 | 0.369±0.005 | **0.306±0.001** | **0.345±0.001** | 0.324±0.008 | 0.359±0.005 | **0.307±0.005** | **0.348±0.003** |
| | 720 | 0.459±0.017 | 0.430±0.006 | **0.380±0.001** | **0.393±0.001** | 0.425±0.007 | 0.419±0.004 | **0.375±0.004** | **0.392±0.002** |
| | Mean | 0.329 | 0.352 | **0.283** | **0.329** | 0.303 | 0.344 | **0.283** | **0.333** |
| Electricity | 96 | 0.267±0.010 | 0.338±0.006 | **0.217±0.014** | **0.309±0.007** | 0.273±0.012 | 0.348±0.007 | **0.225±0.012** | **0.326±0.008** |
| | 192 | 0.322±0.008 | 0.376±0.004 | **0.229±0.013** | **0.319±0.007** | 0.325±0.010 | 0.384±0.006 | **0.238±0.010** | **0.336±0.007** |
| | 336 | 0.378±0.006 | 0.414±0.003 | **0.254±0.007** | **0.337±0.004** | 0.378±0.008 | 0.423±0.006 | **0.260±0.011** | **0.352±0.006** |
| | 720 | 0.486±0.009 | 0.486±0.004 | **0.337±0.006** | **0.393±0.004** | 0.495±0.005 | 0.502±0.004 | **0.341±0.012** | **0.406±0.007** |
| | Mean | 0.363 | 0.403 | **0.259** | **0.340** | 0.368 | 0.414 | **0.266** | **0.355** |
| Traffic | 96 | 0.757±0.430 | 0.457±0.021 | **0.723±0.286** | **0.389±0.017** | 0.766±0.262 | 0.467±0.013 | **0.694±0.395** | **0.412±0.017** |
| | 192 | 0.815±0.305 | 0.489±0.017 | **0.719±0.313** | **0.385±0.012** | 0.812±0.132 | 0.495±0.009 | **0.700±0.342** | **0.410±0.018** |
| | 336 | 0.861±0.189 | 0.517±0.013 | **0.741±0.314** | **0.394±0.012** | 0.854±0.185 | 0.522±0.010 | **0.716±0.319** | **0.419±0.015** |
| | 720 | 0.980±0.159 | 0.583±0.010 | **0.835±0.351** | **0.438±0.013** | 0.981±0.195 | 0.587±0.012 | **0.801±0.315** | **0.462±0.014** |
| | Mean | 0.853 | 0.512 | **0.754** | **0.402** | 0.853 | 0.518 | **0.728** | **0.426** |
| Weather | 96 | **0.156±0.002** | **0.202±0.004** | 0.176±0.008 | 0.209±0.006 | **0.164±0.005** | **0.213±0.006** | 0.184±0.012 | 0.223±0.007 |
| | 192 | **0.213±0.007** | **0.253±0.005** | 0.227±0.023 | 0.256±0.009 | **0.220±0.009** | **0.263±0.005** | 0.238±0.025 | 0.270±0.011 |
| | 336 | **0.270±0.007** | **0.298±0.007** | 0.288±0.024 | 0.300±0.009 | **0.272±0.007** | **0.305±0.005** | 0.289±0.023 | 0.306±0.011 |
| | 720 | 0.375±0.006 | 0.367±0.003 | **0.371±0.020** | **0.353±0.001** | 0.372±0.006 | 0.370±0.005 | **0.365±0.018** | **0.355±0.008** |
| | Mean | **0.254** | **0.280** | 0.266 | 0.279 | **0.257** | 0.288 | 0.269 | 0.288 |

Table 21: Performance comparison of TimesFM and ViTime: Gaussian noise levels 0.5 and 0.7

| Dataset | H | TimesFM GN(0.5) | | ViTime GN(0.5) | | TimesFM GN(0.7) | | ViTime GN(0.7) | |
|---|---|---|---|---|---|---|---|---|---|
| | | MSE | MAE | MSE | MAE | MSE | MAE | MSE | MAE |
| ETTh1 | 96 | **0.428**±0.010 | **0.424**±0.005 | 0.441±0.006 | 0.432±0.004 | **0.434**±0.044 | **0.428**±0.009 | 0.471±0.006 | 0.456±0.005 |
| | 192 | 0.473±0.020 | 0.451±0.007 | **0.465**±0.006 | **0.447**±0.003 | 0.482±0.041 | 0.456±0.009 | **0.477**±0.005 | **0.462**±0.004 |
| | 336 | 0.525±0.014 | 0.486±0.006 | **0.494**±0.005 | **0.468**±0.002 | 0.531±0.053 | 0.490±0.009 | **0.506**±0.005 | **0.480**±0.004 |
| | 720 | 0.634±0.014 | 0.553±0.005 | **0.576**±0.009 | **0.518**±0.003 | 0.638±0.045 | 0.556±0.009 | **0.606**±0.006 | **0.539**±0.003 |
| | Mean | 0.515 | 0.479 | **0.494** | **0.466** | 0.521 | 0.483 | 0.515 | 0.484 |
| ETTh2 | 96 | 0.262±0.002 | 0.327±0.003 | **0.244**±0.002 | **0.315**±0.003 | 0.268±0.002 | 0.332±0.003 | 0.275±0.004 | 0.342±0.003 |
| | 192 | 0.314±0.002 | 0.366±0.003 | **0.283**±0.002 | **0.346**±0.002 | 0.319±0.003 | 0.369±0.003 | 0.310±0.003 | 0.369±0.003 |
| | 336 | 0.365±0.002 | 0.403±0.002 | **0.326**±0.002 | **0.378**±0.002 | 0.368±0.004 | 0.405±0.003 | 0.355±0.002 | 0.402±0.002 |
| | 720 | 0.464±0.002 | 0.470±0.002 | **0.416**±0.001 | **0.440**±0.001 | 0.467±0.002 | 0.471±0.002 | 0.441±0.002 | 0.461±0.002 |
| | Mean | 0.351 | 0.392 | **0.317** | **0.370** | 0.355 | 0.394 | 0.345 | 0.394 |
| ETTm1 | 96 | 0.534±0.035 | 0.442±0.012 | **0.419**±0.005 | **0.415**±0.003 | 0.546±0.031 | 0.448±0.007 | 0.433±0.006 | 0.418±0.003 |
| | 192 | 0.576±0.023 | 0.469±0.008 | **0.448**±0.003 | **0.429**±0.002 | 0.580±0.018 | 0.473±0.005 | 0.445±0.004 | 0.425±0.003 |
| | 336 | 0.628±0.022 | 0.500±0.007 | **0.486**±0.005 | **0.449**±0.002 | 0.645±0.010 | 0.505±0.003 | 0.480±0.004 | 0.444±0.003 |
| | 720 | 0.695±0.028 | 0.539±0.007 | **0.559**±0.007 | **0.486**±0.003 | 0.693±0.022 | 0.542±0.007 | 0.550±0.006 | 0.484±0.003 |
| | Mean | 0.608 | 0.488 | **0.478** | **0.445** | 0.616 | 0.492 | **0.477** | **0.443** |
| ETTm2 | 96 | **0.199**±0.005 | **0.279**±0.004 | 0.202±0.006 | 0.283±0.005 | **0.205**±0.006 | **0.285**±0.006 | 0.211±0.008 | 0.292±0.007 |
| | 192 | 0.263±0.007 | 0.321±0.004 | **0.257**±0.005 | **0.319**±0.003 | 0.270±0.006 | 0.327±0.004 | **0.262**±0.007 | **0.326**±0.006 |
| | 336 | 0.324±0.010 | 0.360±0.007 | **0.308**±0.004 | **0.351**±0.003 | 0.331±0.008 | 0.364±0.006 | **0.311**±0.005 | **0.358**±0.004 |
| | 720 | 0.419±0.007 | 0.418±0.004 | **0.377**±0.004 | **0.395**±0.002 | 0.426±0.006 | 0.421±0.004 | **0.389**±0.004 | **0.407**±0.002 |
| | Mean | 0.301 | 0.345 | **0.286** | **0.337** | 0.308 | 0.349 | **0.293** | **0.346** |
| Electricity | 96 | 0.283±0.008 | 0.362±0.004 | **0.242**±0.015 | **0.344**±0.007 | 0.297±0.010 | 0.377±0.004 | **0.257**±0.012 | **0.357**±0.006 |
| | 192 | 0.334±0.011 | 0.398±0.006 | **0.252**±0.015 | **0.352**±0.007 | 0.347±0.008 | 0.413±0.005 | **0.260**±0.013 | **0.359**±0.006 |
| | 336 | 0.390±0.010 | 0.440±0.005 | **0.273**±0.013 | **0.368**±0.006 | 0.405±0.009 | 0.457±0.004 | **0.277**±0.013 | **0.371**±0.006 |
| | 720 | 0.528±0.013 | 0.532±0.006 | **0.357**±0.014 | **0.422**±0.008 | 0.556±0.010 | 0.554±0.005 | **0.355**±0.012 | **0.419**±0.006 |
| | Mean | 0.384 | 0.433 | **0.281** | **0.371** | 0.401 | 0.450 | **0.287** | **0.377** |
| Traffic | 96 | 0.761±0.268 | 0.478±0.016 | **0.747**±0.348 | **0.447**±0.013 | **0.800**±0.217 | **0.495**±0.011 | 0.833±0.654 | 0.503±0.026 |
| | 192 | 0.825±0.139 | 0.507±0.009 | **0.755**±0.256 | **0.446**±0.012 | 0.831±0.279 | 0.519±0.020 | **0.815**±0.486 | **0.498**±0.021 |
| | 336 | 0.863±0.095 | 0.533±0.009 | **0.776**±0.222 | **0.455**±0.012 | 0.871±0.168 | 0.547±0.012 | **0.829**±0.393 | **0.502**±0.019 |
| | 720 | 0.993±0.148 | 0.600±0.012 | **0.862**±0.212 | **0.496**±0.013 | 1.008±0.182 | 0.613±0.014 | **0.918**±0.361 | **0.538**±0.018 |
| | Mean | 0.860 | 0.529 | **0.785** | **0.461** | 0.877 | 0.543 | **0.849** | **0.510** |
| Weather | 96 | **0.171**±0.005 | **0.222**±0.004 | 0.189±0.014 | 0.233±0.010 | **0.177**±0.005 | **0.230**±0.005 | 0.188±0.016 | 0.240±0.010 |
| | 192 | **0.225**±0.006 | **0.271**±0.005 | 0.238±0.029 | 0.276±0.016 | **0.234**±0.008 | **0.278**±0.006 | 0.237±0.028 | 0.280±0.017 |
| | 336 | **0.282**±0.010 | 0.313±0.005 | 0.286±0.024 | **0.310**±0.014 | **0.290**±0.005 | 0.320±0.005 | 0.291±0.024 | **0.317**±0.016 |
| | 720 | 0.379±0.009 | 0.376±0.005 | **0.358**±0.021 | **0.356**±0.012 | 0.386±0.009 | 0.381±0.005 | **0.372**±0.019 | **0.366**±0.011 |
| | Mean | **0.264** | 0.295 | 0.268 | **0.294** | 0.272 | 0.302 | 0.272 | 0.301 |

Table 22: Performance comparison of TimesFM and ViTime: Gaussian noise level 1.0 and data missing (DM 0.3)

| Dataset | H | TimesFM GN(1.0) | | ViTime GN(1.0) | | ViTime DM(0.3) | |
|---|---|---|---|---|---|---|---|
| | | MSE | MAE | MSE | MAE | MSE | MAE |
| ETTh1 | 96 | **0.436±0.056** | **0.436±0.008** | 0.481±0.012 | 0.459±0.008 | 0.433±0.005 | 0.414±0.002 |
| | 192 | **0.483±0.042** | **0.462±0.010** | 0.502±0.013 | 0.472±0.007 | 0.463±0.006 | 0.433±0.003 |
| | 336 | 0.537±0.096 | 0.493±0.011 | **0.522±0.004** | **0.487±0.003** | 0.496±0.005 | 0.456±0.002 |
| | 720 | 0.636±0.227 | 0.555±0.019 | **0.599±0.007** | **0.530±0.003** | 0.574±0.009 | 0.507±0.002 |
| | Mean | **0.523** | **0.487** | 0.526 | 0.487 | 0.491 | 0.453 |
| ETTh2 | 96 | **0.280±0.002** | **0.340±0.002** | 0.292±0.008 | 0.356±0.007 | 0.255±0.001 | 0.319±0.002 |
| | 192 | **0.328±0.003** | **0.375±0.002** | 0.330±0.007 | 0.385±0.006 | 0.297±0.001 | 0.352±0.001 |
| | 336 | 0.379±0.004 | 0.410±0.003 | **0.370±0.005** | **0.414±0.004** | 0.343±0.001 | 0.387±0.001 |
| | 720 | 0.474±0.006 | 0.473±0.004 | **0.466±0.003** | **0.478±0.003** | 0.441±0.001 | 0.455±0.001 |
| | Mean | 0.365 | **0.399** | **0.364** | 0.408 | 0.334 | 0.378 |
| ETTm1 | 96 | 0.550±0.053 | 0.459±0.016 | **0.470±0.016** | **0.449±0.005** | 0.408±0.004 | 0.392±0.001 |
| | 192 | 0.589±0.027 | 0.484±0.006 | **0.491±0.007** | **0.459±0.003** | 0.451±0.002 | 0.413±0.001 |
| | 336 | 0.635±0.035 | 0.510±0.006 | **0.524±0.007** | **0.474±0.003** | 0.507±0.003 | 0.440±0.001 |
| | 720 | 0.695±0.020 | 0.545±0.004 | **0.579±0.008** | **0.500±0.003** | 0.595±0.003 | 0.484±0.001 |
| | Mean | 0.617 | 0.500 | **0.516** | **0.471** | 0.490 | 0.432 |
| ETTm2 | 96 | **0.217±0.006** | **0.295±0.005** | 0.228±0.009 | 0.307±0.007 | 0.199±0.005 | 0.274±0.004 |
| | 192 | **0.283±0.009** | **0.336±0.007** | 0.279±0.007 | 0.340±0.006 | 0.262±0.004 | 0.317±0.004 |
| | 336 | 0.343±0.010 | 0.372±0.006 | **0.327±0.004** | **0.369±0.003** | 0.320±0.005 | 0.354±0.003 |
| | 720 | 0.442±0.009 | 0.431±0.006 | **0.404±0.003** | **0.415±0.002** | 0.398±0.004 | 0.403±0.002 |
| | Mean | 0.321 | 0.359 | **0.309** | **0.358** | 0.295 | 0.337 |
| Electricity | 96 | 0.319±0.009 | 0.398±0.006 | **0.296±0.021** | **0.389±0.010** | 0.222±0.014 | 0.311±0.007 |
| | 192 | 0.385±0.009 | 0.446±0.006 | **0.306±0.021** | **0.397±0.010** | 0.235±0.013 | 0.323±0.007 |
| | 336 | 0.431±0.012 | 0.480±0.006 | **0.326±0.021** | **0.410±0.010** | 0.260±0.007 | 0.341±0.004 |
| | 720 | 0.583±0.012 | 0.575±0.005 | **0.417±0.020** | **0.465±0.010** | 0.343±0.006 | 0.397±0.004 |
| | Mean | 0.429 | 0.475 | **0.336** | **0.415** | 0.265 | 0.343 |
| Traffic | 96 | **0.814±0.123** | **0.519±0.006** | 0.846±0.416 | 0.536±0.025 | 0.782±0.348 | 0.406±0.017 |
| | 192 | 0.862±0.095 | 0.544±0.006 | **0.858±0.354** | **0.536±0.022** | 0.772±0.256 | 0.400±0.012 |
| | 336 | 0.907±0.093 | 0.571±0.006 | **0.869±0.309** | **0.538±0.020** | 0.792±0.222 | 0.409±0.012 |
| | 720 | 1.037±0.149 | 0.633±0.009 | **0.953±0.302** | **0.574±0.019** | 0.886±0.212 | 0.454±0.013 |
| | Mean | 0.905 | 0.567 | **0.881** | **0.546** | 0.808 | 0.417 |
| Weather | 96 | **0.188±0.006** | **0.241±0.005** | 0.197±0.011 | 0.250±0.008 | 0.170±0.014 | 0.209±0.010 |
| | 192 | **0.244±0.009** | **0.288±0.006** | 0.244±0.019 | 0.288±0.011 | 0.231±0.029 | 0.261±0.016 |
| | 336 | 0.302±0.011 | 0.330±0.008 | **0.291±0.018** | **0.320±0.012** | 0.288±0.024 | 0.302±0.014 |
| | 720 | 0.399±0.010 | 0.390±0.007 | **0.361±0.014** | **0.364±0.009** | 0.366±0.021 | 0.353±0.012 |
| | Mean | 0.283 | 0.312 | **0.273** | **0.305** | 0.264 | 0.281 |

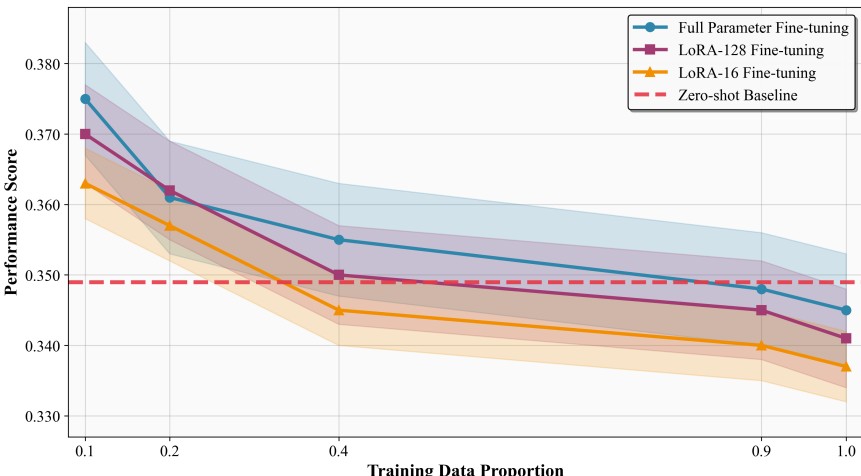

Figure 17: Performance comparison of different fine-tuning strategies on the ETTh2 dataset with varying data ratios. The zero-shot (ZS) performance is shown as a dashed line for reference. Full fine-tuning (Full FT) initially degrades performance on small data subsets before improving. In contrast, parameter-efficient LoRA methods, especially with fewer parameters (1% LoRA), effectively mitigate this degradation by preserving pre-trained knowledge.

### D.5.1 Analysis on Pre-Denoising Methods

To examine the effectiveness of pre-denoising algorithms in improving numerical model accuracy under moderate noise interference, we conducted a series of experiments evaluating standard denoising techniques applied to time series data before model input. The experiments were performed on the Electricity dataset, to which we added Gaussian noise with a variance of $\sigma^2 = 0.5$ to simulate moderate noise conditions. We then applied three common pre-processing denoising methods before feeding the data into the baseline TimesFM model. The methods evaluated are:

- **3-Sigma Rule**: A simple outlier detection and removal method.

- **Moving Average**: A smoothing filter with a window size of 5.

- **Savitzky-Golay Filter**: A polynomial-based smoothing filter (polyorder=3).

The results as shown in Figure 18, illustrate the performance improvements and trade-offs associated with these pre-processing denoising steps under moderate noise conditions.

As depicted in the figure, under moderate noise interference, the effectiveness of pre-denoising algorithms in improving model accuracy varies significantly depending on both the method and the prediction horizon. The **3-sigma** method demonstrates modest accuracy improvements for short prediction lengths (96 and 192), achieving positive ReMAE improvement rates. However, its performance degrades and even becomes detrimental for longer horizons. This limitation arises because the method only removes extreme outliers but fails to address the underlying noise distribution that significantly impacts long-term trend forecasting accuracy.

Smoothing-based denoising methods like **moving average** and **Savitzky-Golay (savgol)** show contrasting results for numerical model accuracy under moderate noise. These methods exhibit poor performance for short-term predictions, showing negative improvement rates and actually reducing model accuracy. This degradation occurs because smoothing techniques inherently average out recent fluctuations and introduce temporal lag in the data. While this characteristic helps reveal long-term trends and improves accuracy for

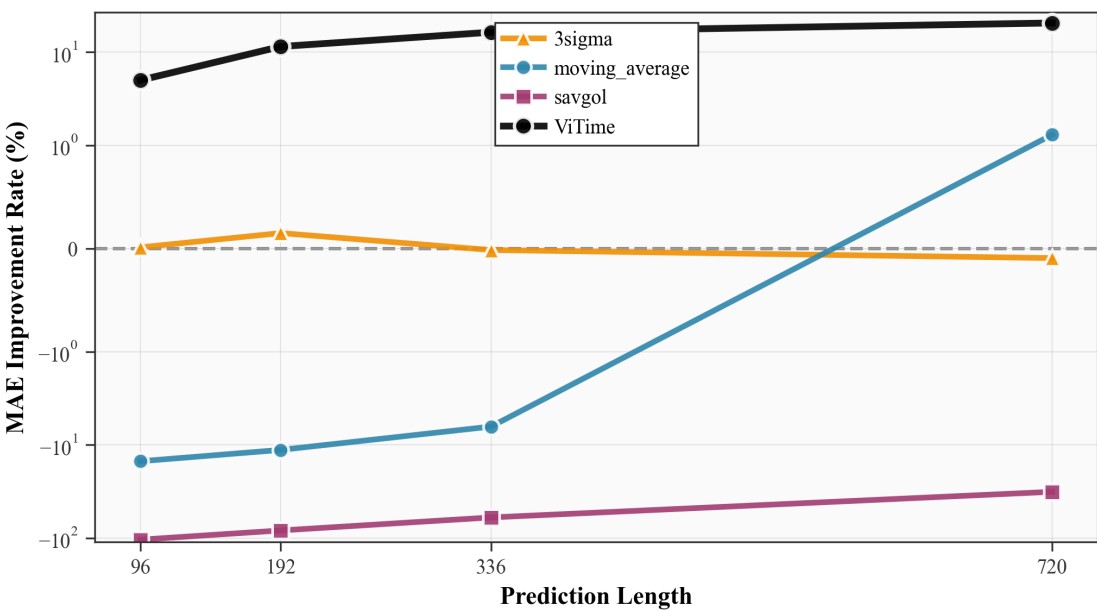

**Performance Comparison of TimesFM with Various Denoising Methods**

Figure 18: Performance comparison of TimesFM combined with various pre-denoising methods on the Electricity dataset under moderate Gaussian noise ($\sigma^2 = 0.5$). The y-axis represents the ReMAE improvement rate (%) compared to the baseline TimesFM without any denoising. Our proposed method, ViTime, is included as a reference to showcase its superior performance.

extended horizons (as evidenced by the moving average's positive performance at the 720-step horizon), it corrupts the fine-grained, high-frequency patterns essential for accurate short-term forecasting.

In stark contrast, our proposed **ViTime** method, serving as a reference line in the chart, consistently achieves significant ReMAE improvements across all prediction lengths under moderate noise conditions. This superior performance demonstrates that ViTime's intrinsic architecture for handling time series data provides inherent robustness against moderate noise interference. Unlike traditional pre-processing denoising pipelines that often involve accuracy trade-offs between different prediction horizons, ViTime successfully denoises and extracts meaningful features while maintaining high numerical precision, thereby establishing its **superior robustness** in moderately noisy environments.

### D.6 Computational Complexity Analysis

We conduct extensive experiments to analyze the computational complexity and prediction accuracy of our proposed models. All experiments are performed with batch size 4, input sequence length 512, and prediction horizon 720 on a single Nvidia 3090 GPU.

Table 23: Model Performance and Computational Resource Requirements

| Model | GPU Mem. | Inference Time (s/batch) | | Params | Avg. ReMAE |
|---|---|---|---|---|---|
| | (MB) | Total | Map. & Inv. Map. | (M) | |
| TimesFM | 18,154 | 0.130 | - | 200 | 0.423 |
| ViTime w/ Refining | 3,120 | 2.890 | 0.0068 | 95 | **0.376** |
| ViTime w/o Refining | **667** | **0.082** | 0.0068 | **74** | 0.381 |

The results are reported in Table 23. The baseline TimesFM model requires substantial computational resources with 18.1GB GPU memory and 200M parameters, while achieving an average ReMAE of 0.423. In contrast, our proposed ViTime architecture demonstrates remarkable improvements in both efficiency and accuracy. The basic version without the refining module strikes an optimal balance between computational efficiency and performance - it requires only 667MB GPU memory ($27\times$ reduction), achieves faster inference at 0.082s per batch, uses 63% fewer parameters (74M), while maintaining competitive accuracy with an average ReMAE of 0.381.

For applications prioritizing prediction accuracy, the ViTime variant with refining module achieves the best performance with an average ReMAE of 0.376, representing an 11.1% improvement over TimesFM. This comes at the cost of increased computational overhead - 3.1GB GPU memory and 2.89s inference time per batch, though still maintaining a $5.8\times$ reduction in memory compared to TimesFM. Notably, the mapping & inverse mapping between image space and numerical space in ViTime variants consume only 0.0068s, representing 8.3% and 0.24% of the total inference time for the basic and refined versions, respectively.

These results demonstrate that our proposed architecture offers flexible deployment options: the basic version for resource-constrained scenarios requiring good accuracy and computational efficiency, and the refined version for applications where prediction accuracy is paramount. Both variants significantly outperform the baseline in terms of the computation-accuracy trade-off.

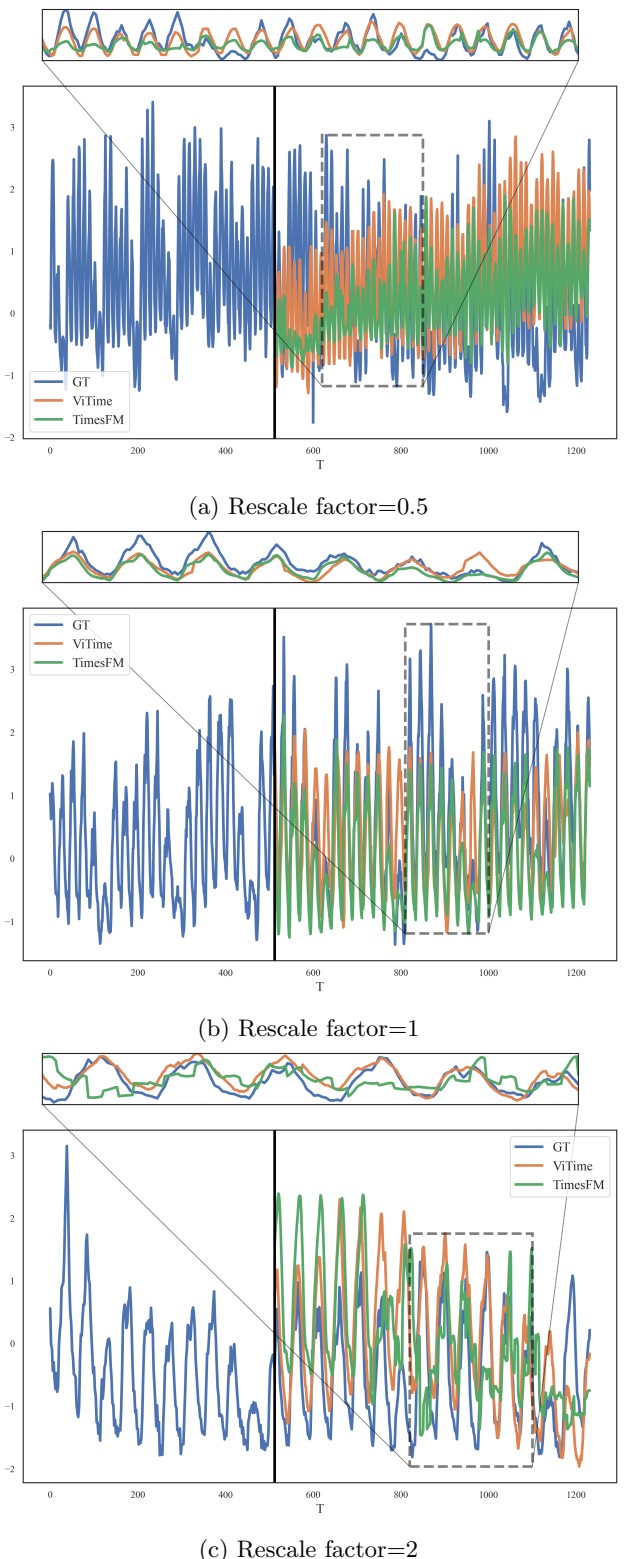

(a) Rescale factor=0.5

(b) Rescale factor=1

(c) Rescale factor=2

Figure 19: Illustrative example of Electricity dataset.

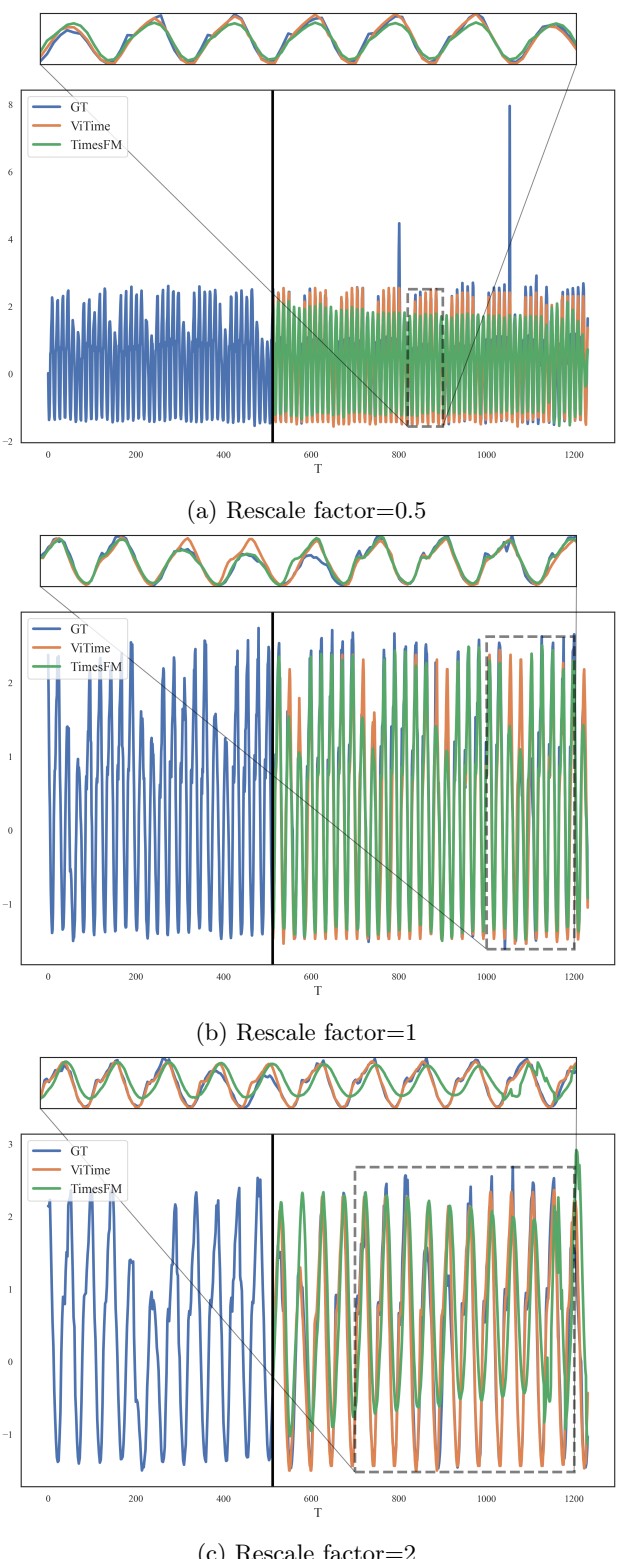

(a) Rescale factor=0.5

(b) Rescale factor=1

(c) Rescale factor=2

Figure 20: Illustrative example of Traffic dataset.

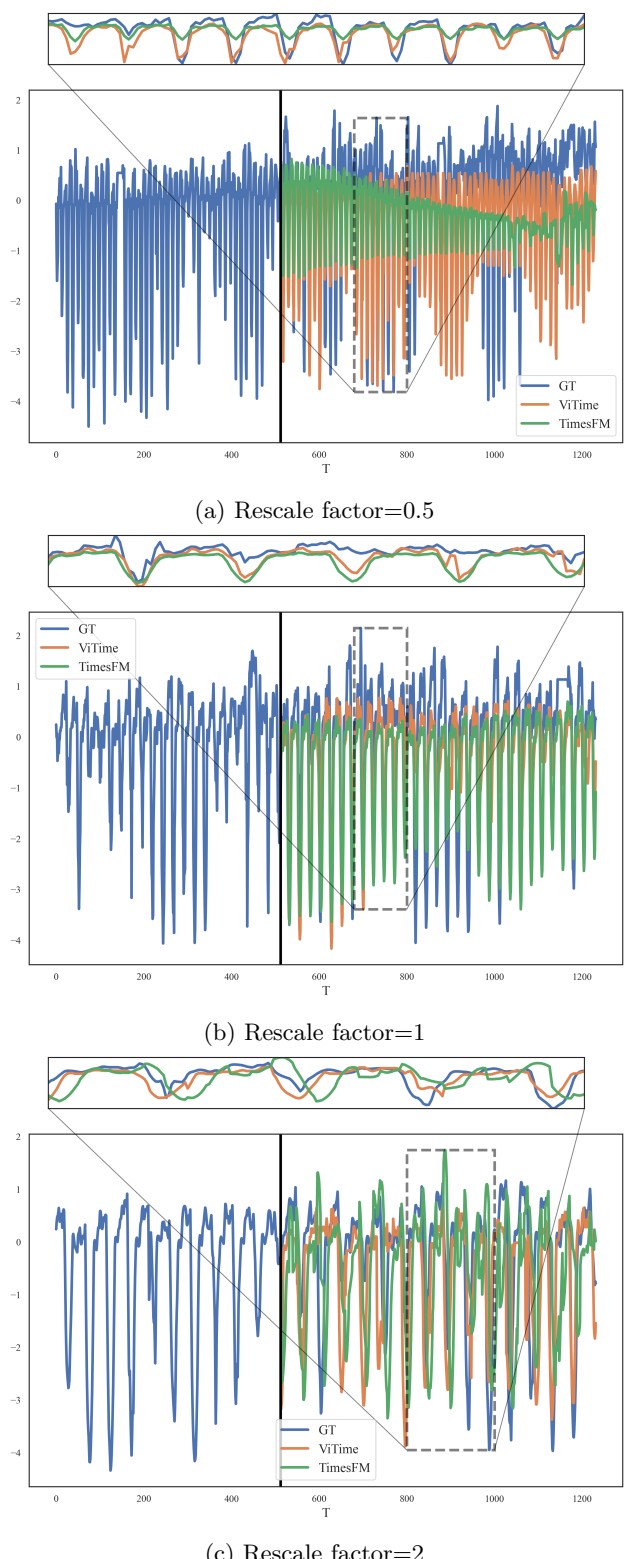

(a) Rescale factor=0.5

(b) Rescale factor=1

(c) Rescale factor=2

Figure 21: Illustrative example of ETTh1 dataset.

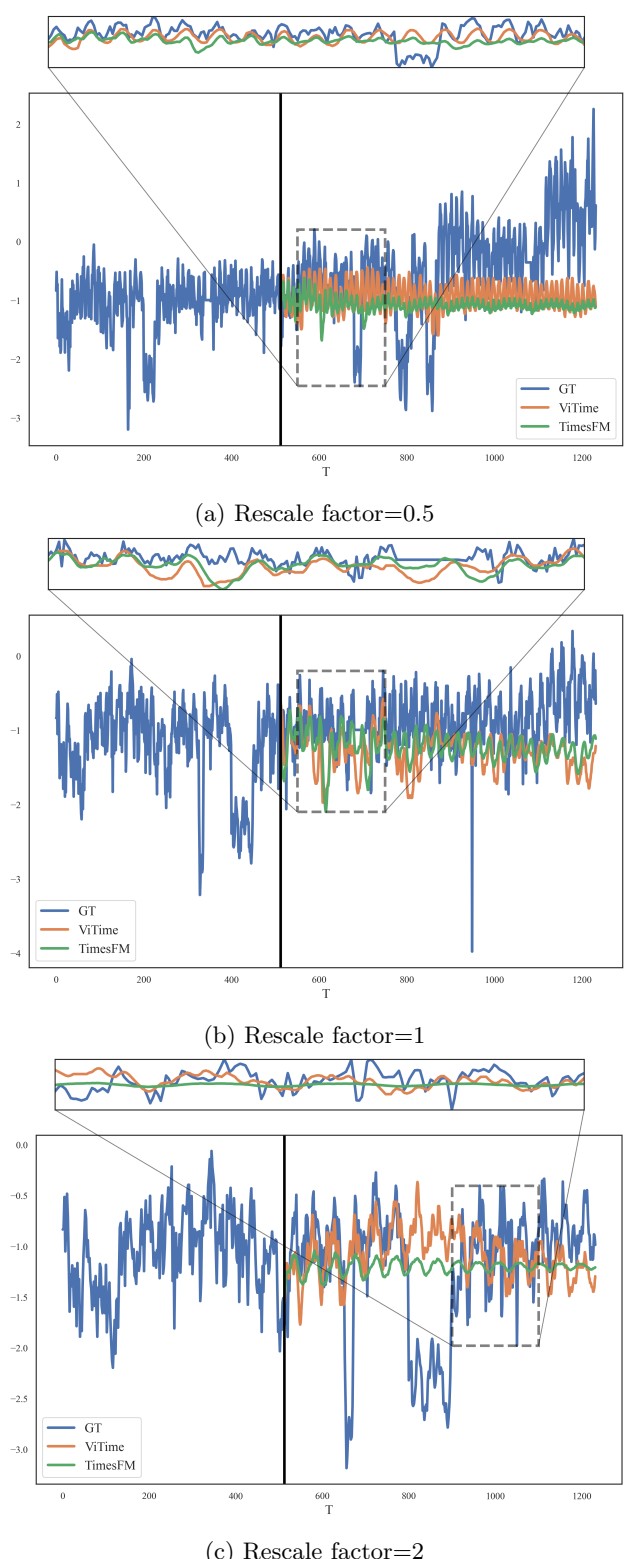

(a) Rescale factor=0.5

(b) Rescale factor=1

(c) Rescale factor=2

Figure 22: Illustrative example of ETTh2 dataset.

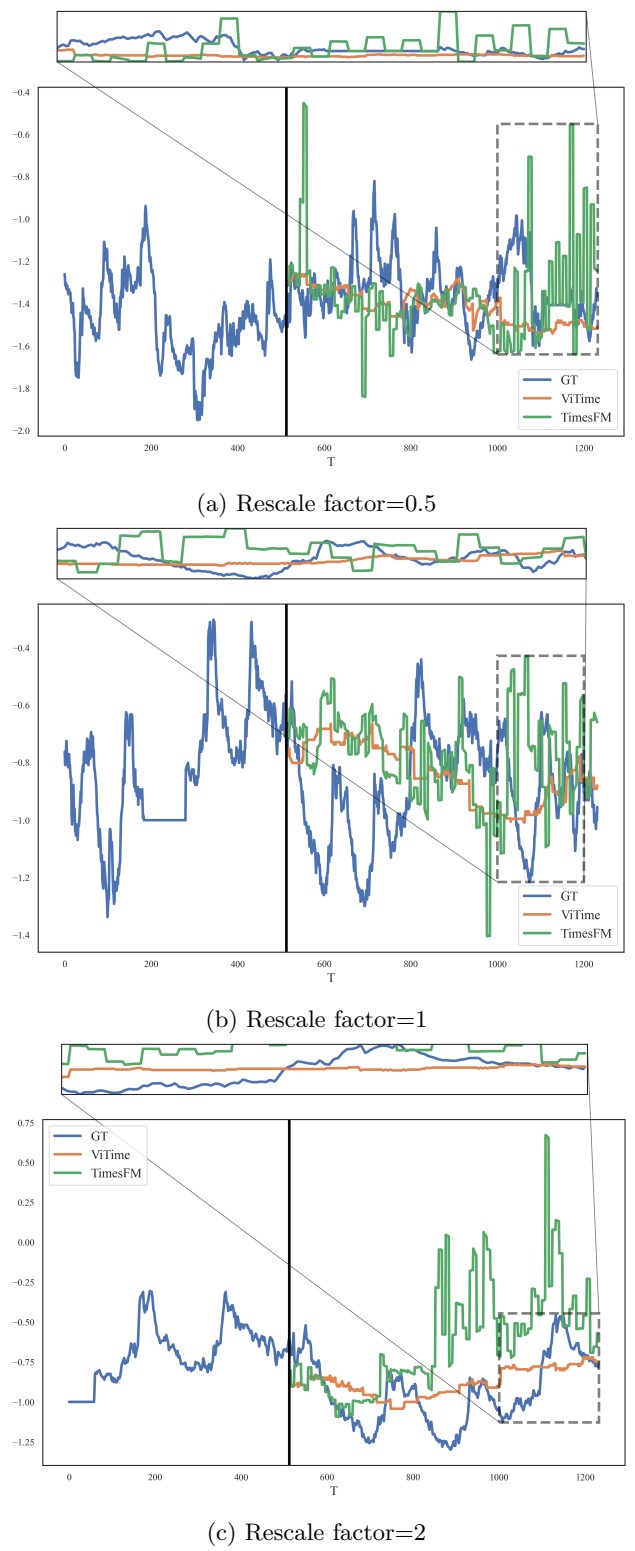

(a) Rescale factor=0.5

(b) Rescale factor=1

(c) Rescale factor=2

Figure 23: Illustrative example of ETTm1 dataset.

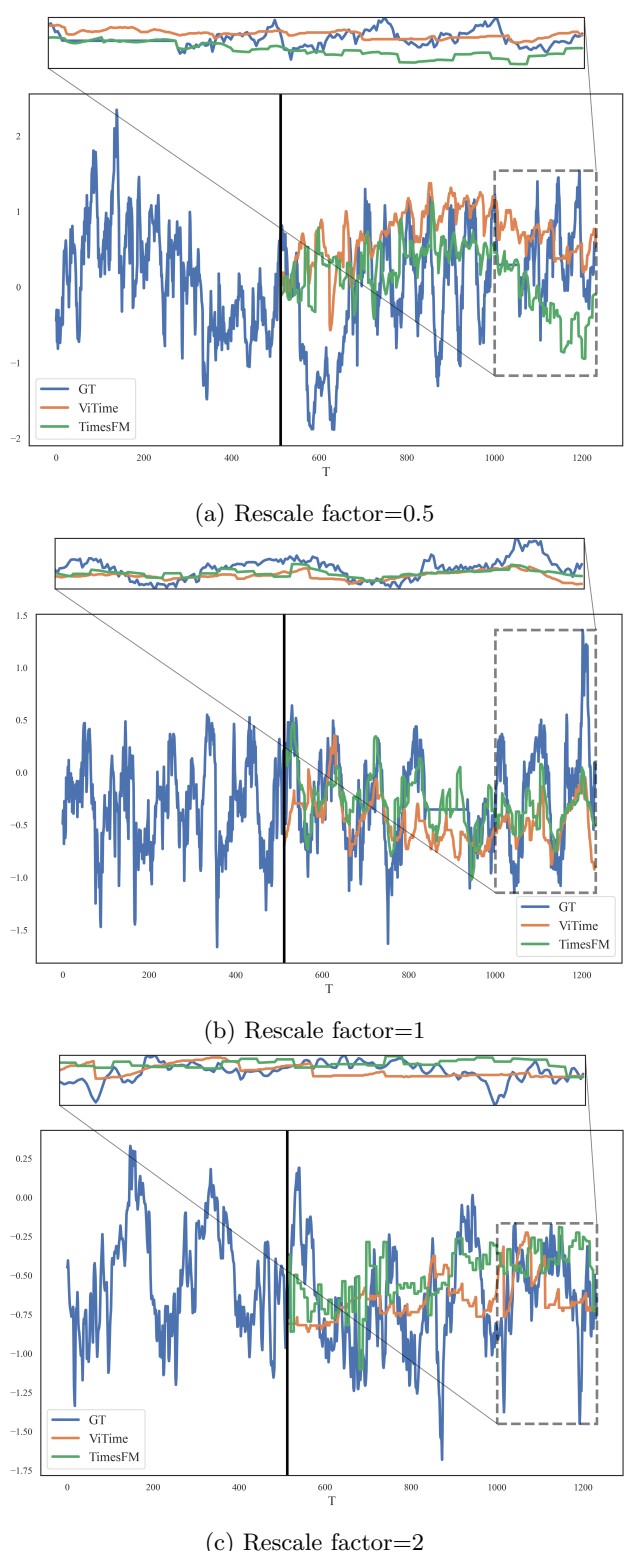

(a) Rescale factor=0.5

(b) Rescale factor=1

(c) Rescale factor=2

Figure 24: Illustrative example of ETTm2 dataset.

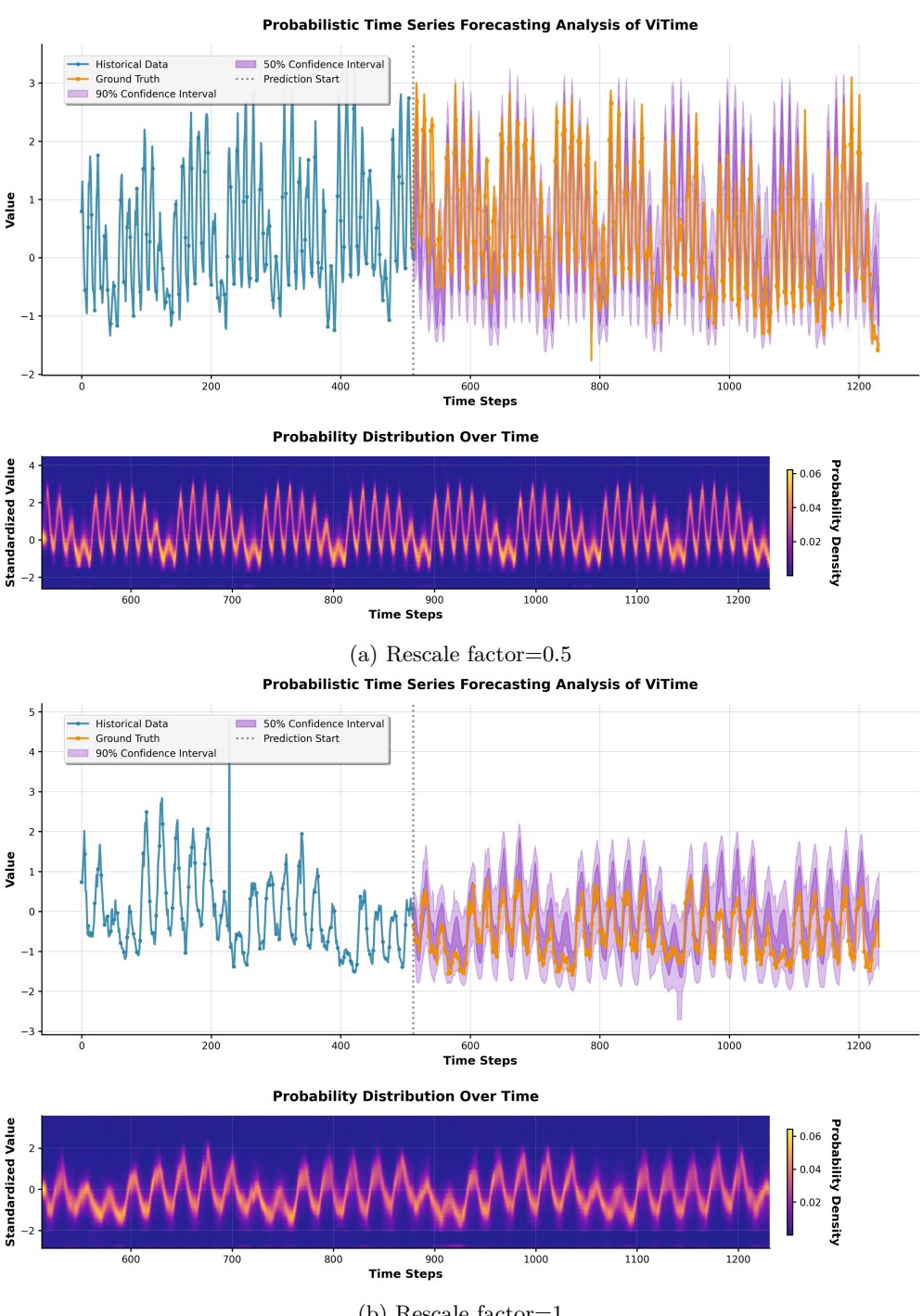

(a) Rescale factor=0.5

(b) Rescale factor=1

Figure 25: Illustrative example of dataset electricity (Part 1).

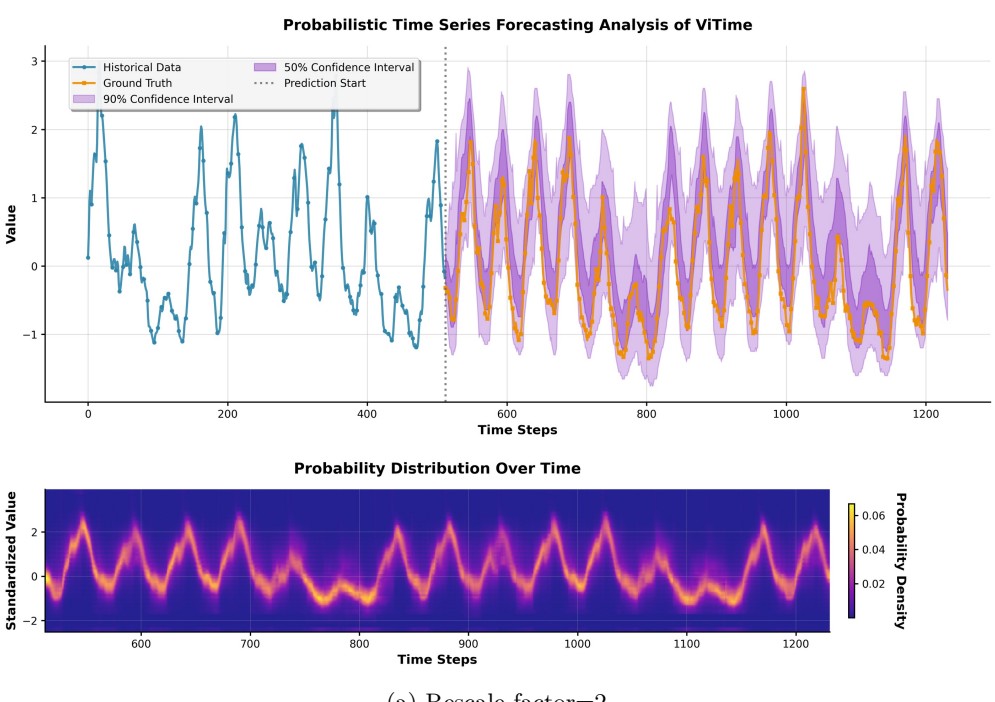

(a) Rescale factor=2

Figure 26: Illustrative example of dataset electricity (Part 2).

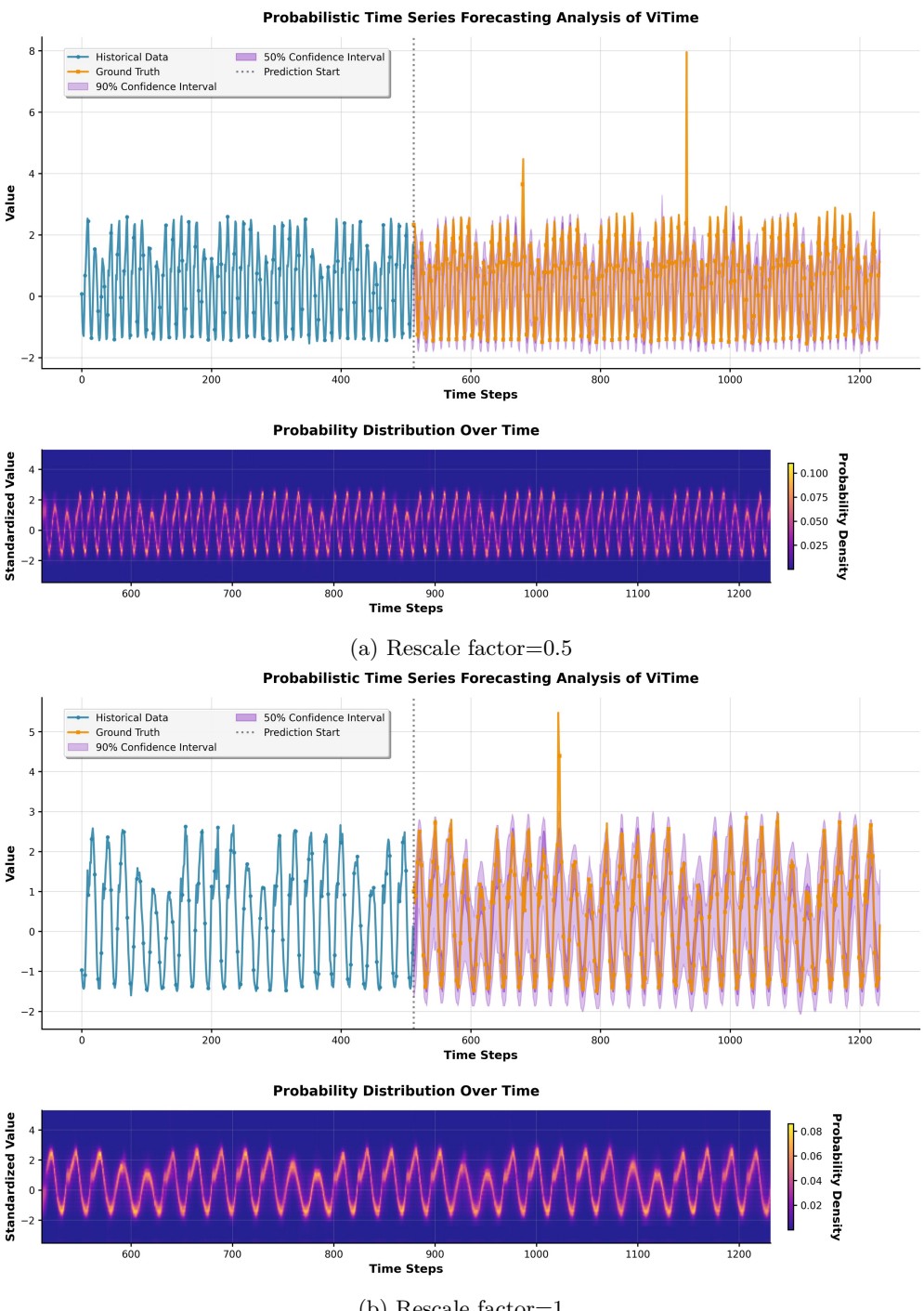

(a) Rescale factor=0.5

(b) Rescale factor=1

Figure 27: Illustrative example of dataset traffic (Part 1).

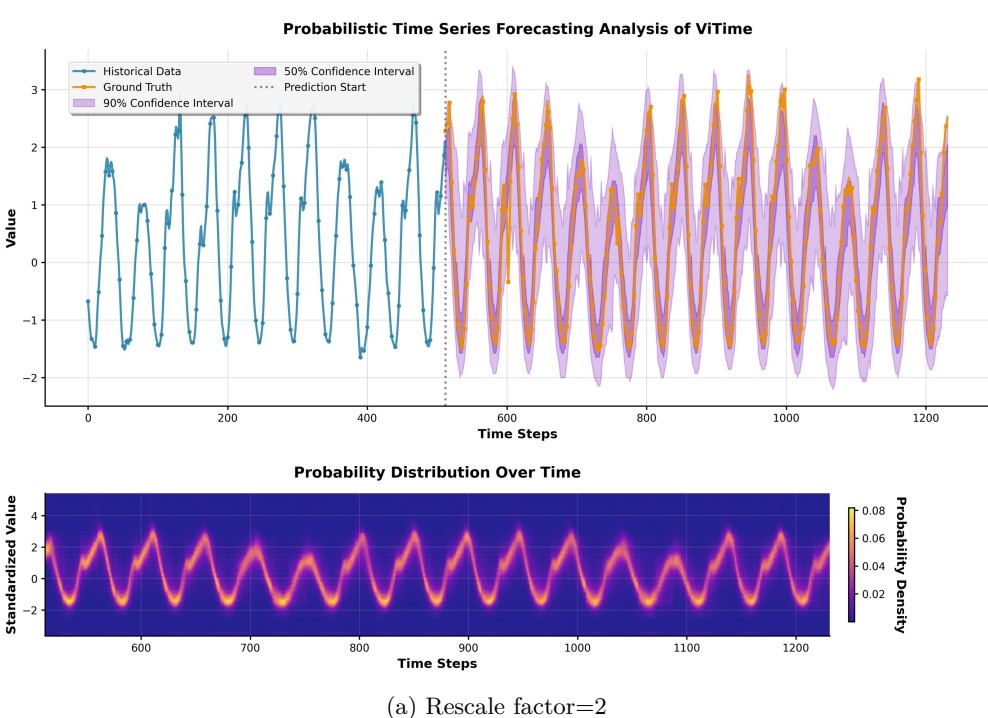

(a) Rescale factor=2

Figure 28: Illustrative example of dataset traffic (Part 2).

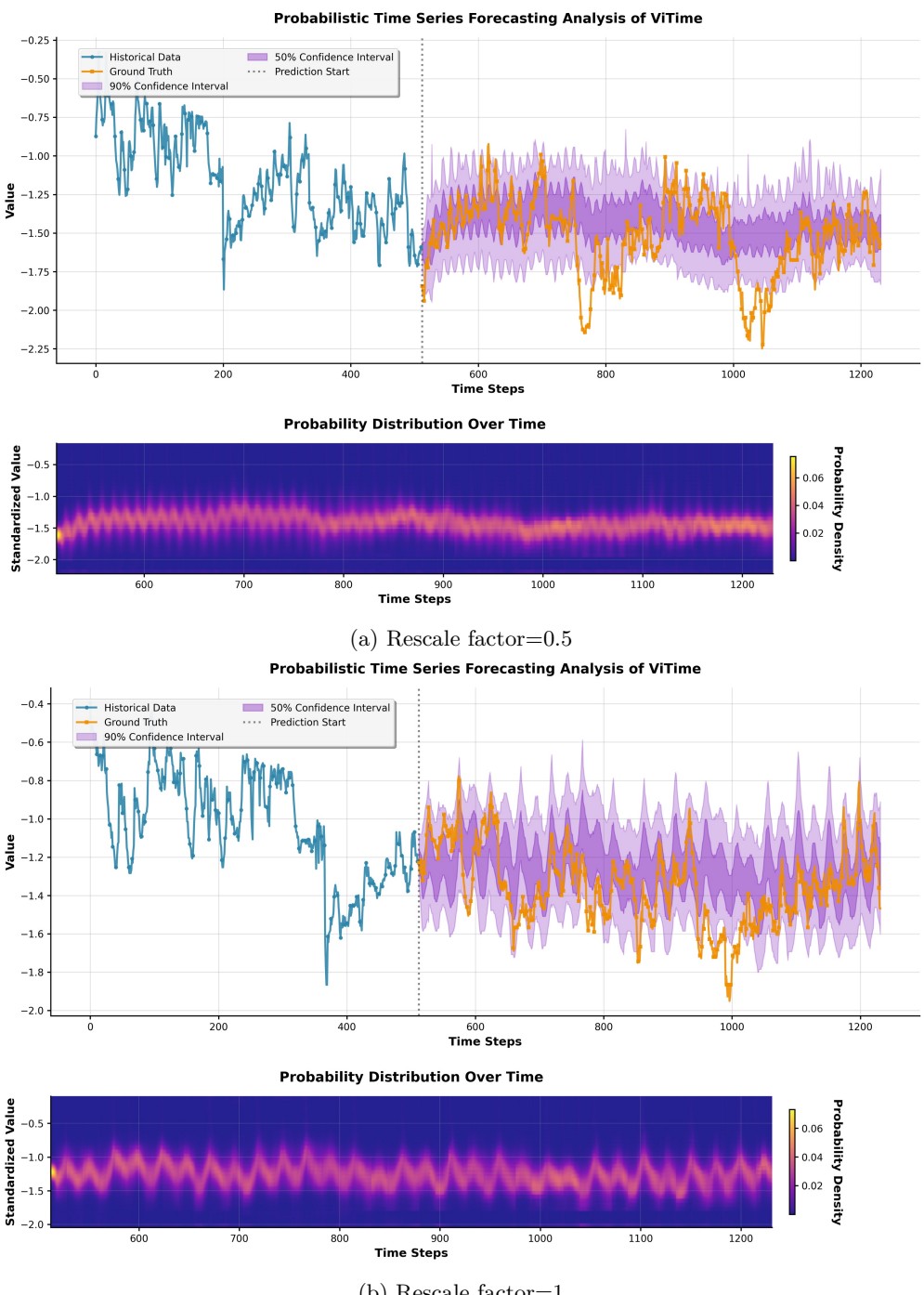

(a) Rescale factor=0.5

(b) Rescale factor=1

Figure 29: Illustrative example of dataset ETTh1 (Part 1).

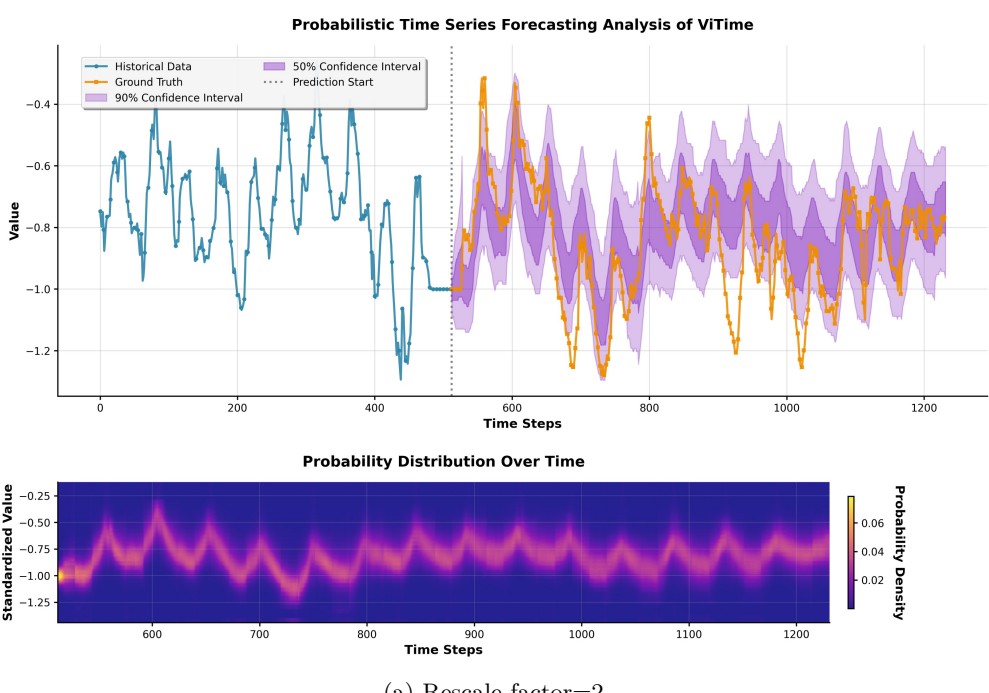

(a) Rescale factor=2

Figure 30: Illustrative example of dataset ETTh1 (Part 2).

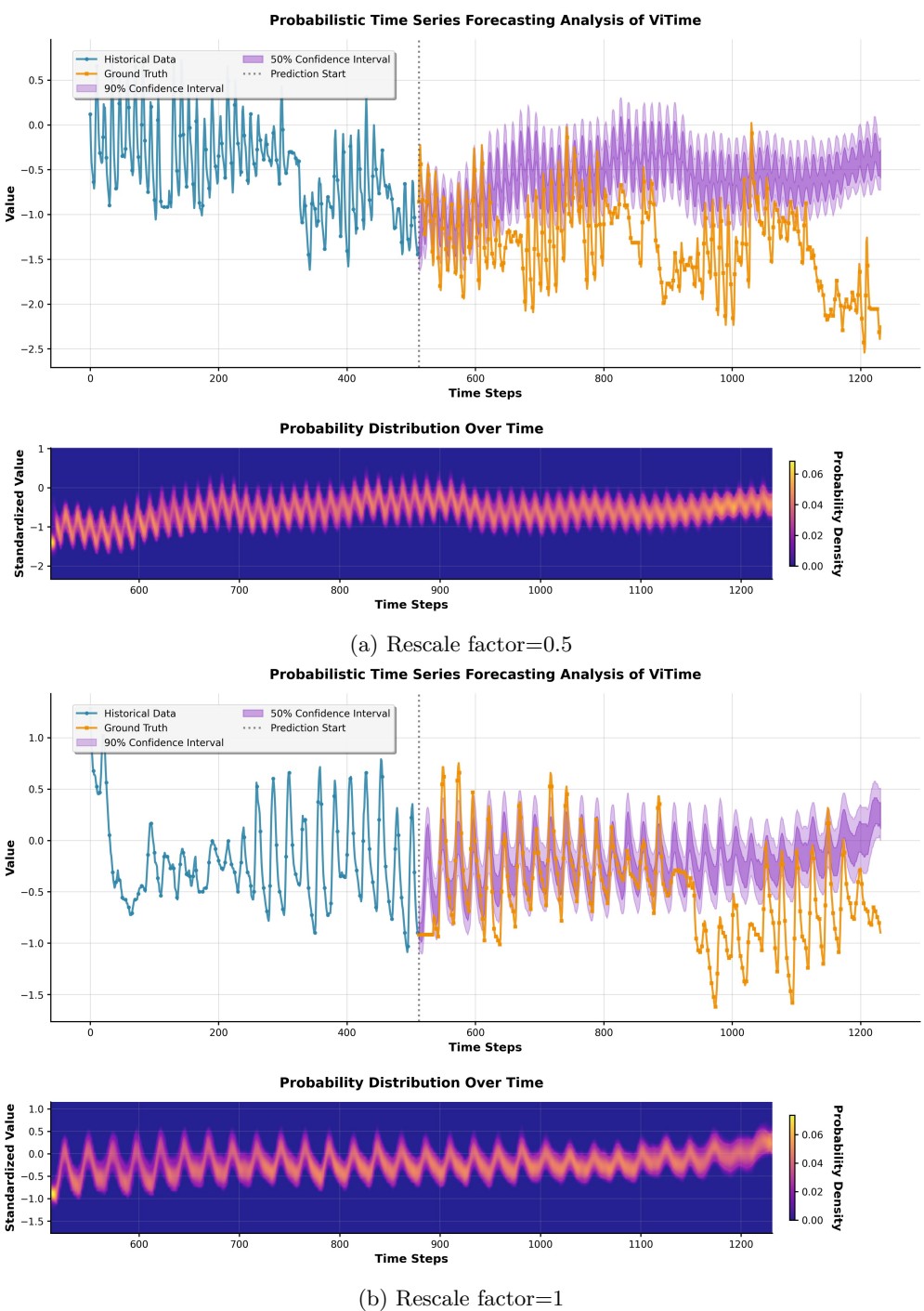

(a) Rescale factor=0.5

(b) Rescale factor=1

Figure 31: Illustrative example of dataset ETTh2 (Part 1).

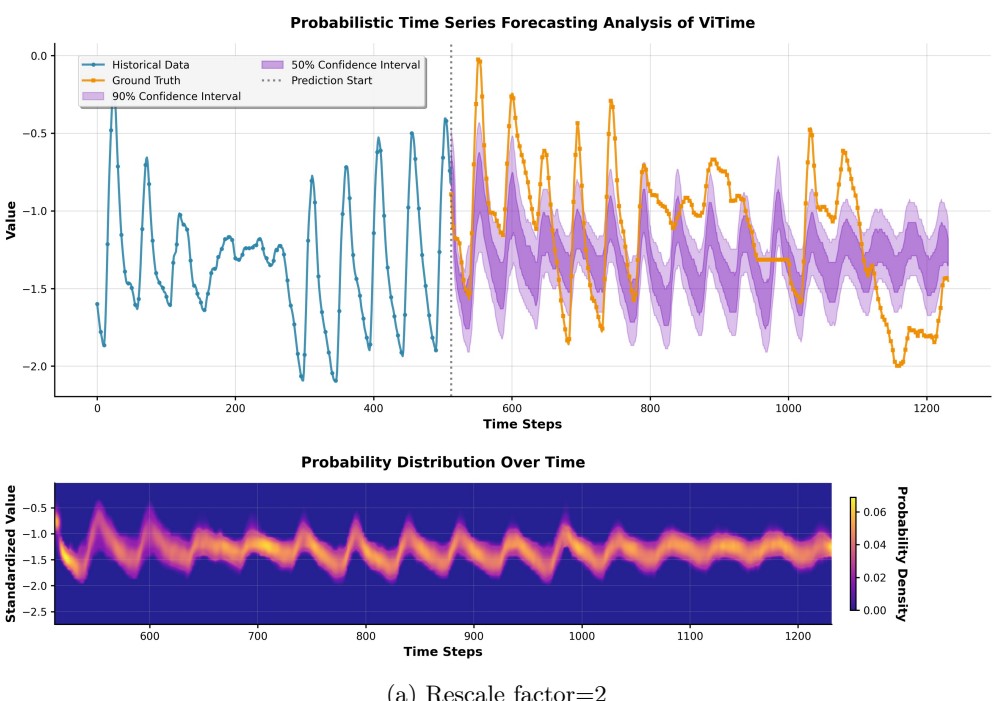

(a) Rescale factor=2

Figure 32: Illustrative example of dataset ETTh2 (Part 2).

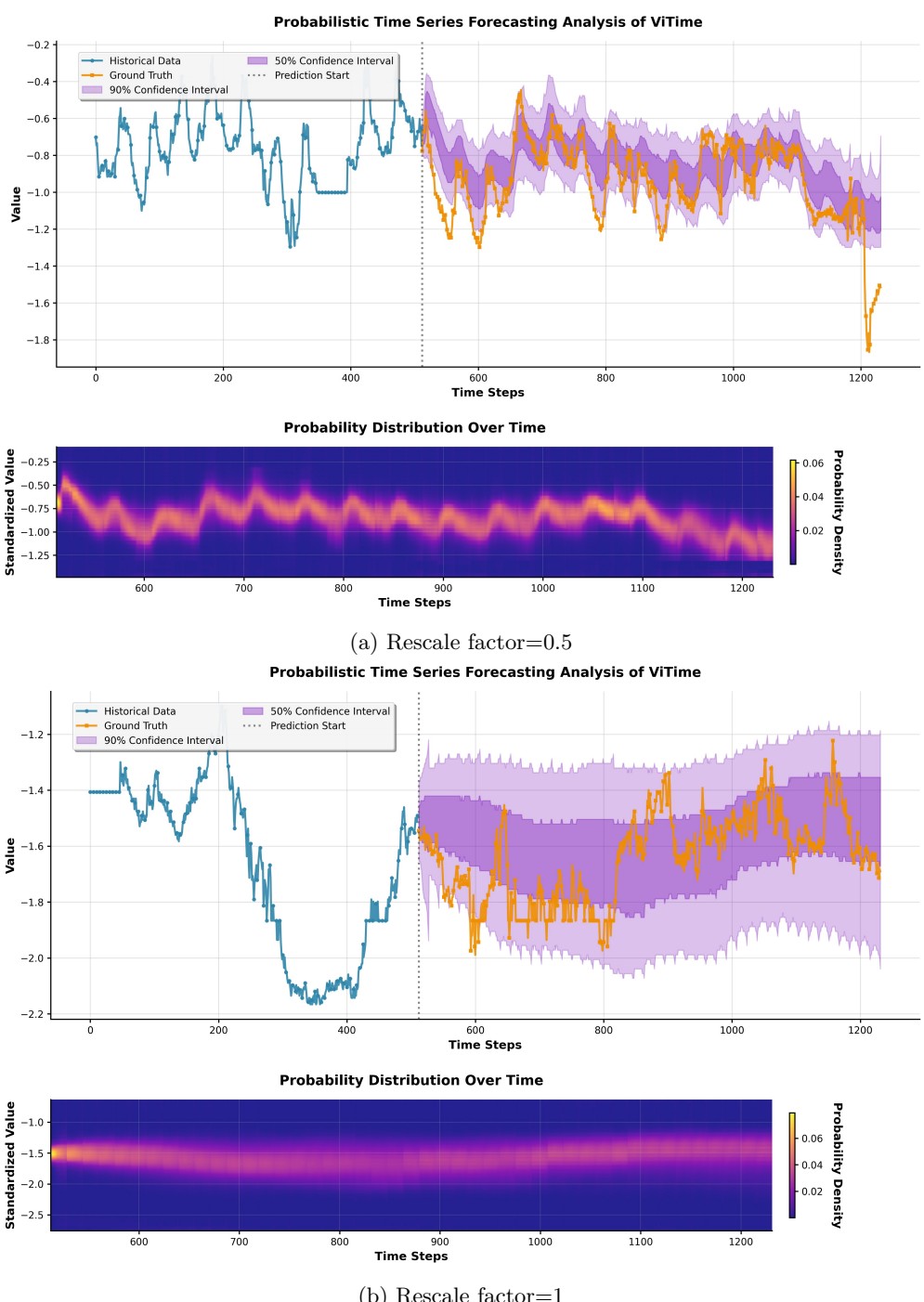

(a) Rescale factor=0.5

(b) Rescale factor=1

Figure 33: Illustrative example of dataset ETTm1 (Part 1).

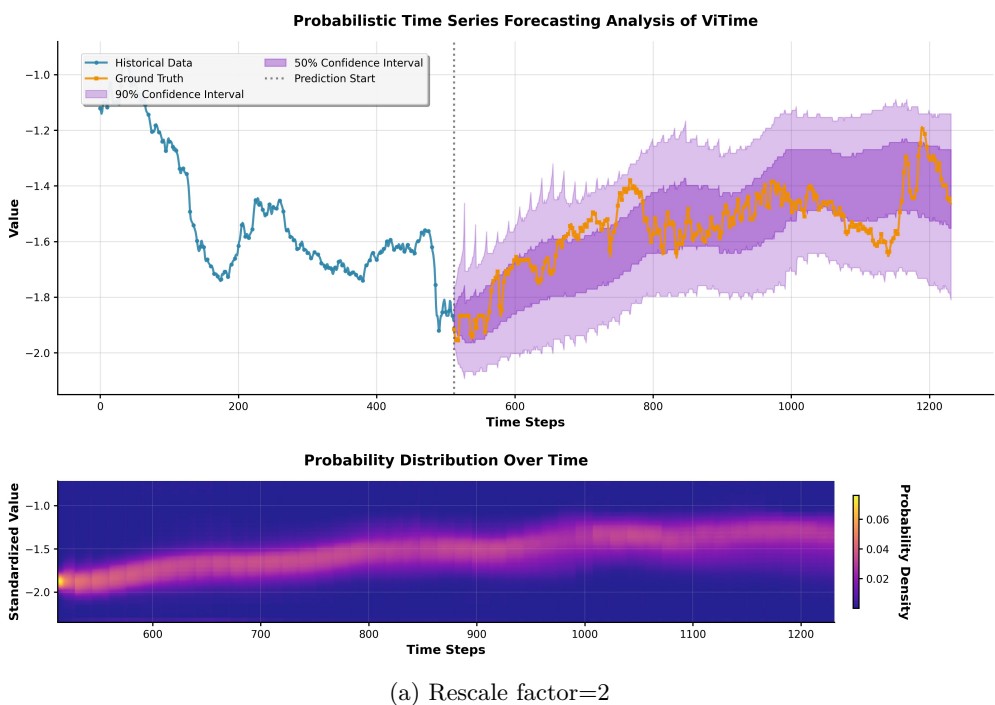

(a) Rescale factor=2

Figure 34: Illustrative example of dataset ETTm1 (Part 2).

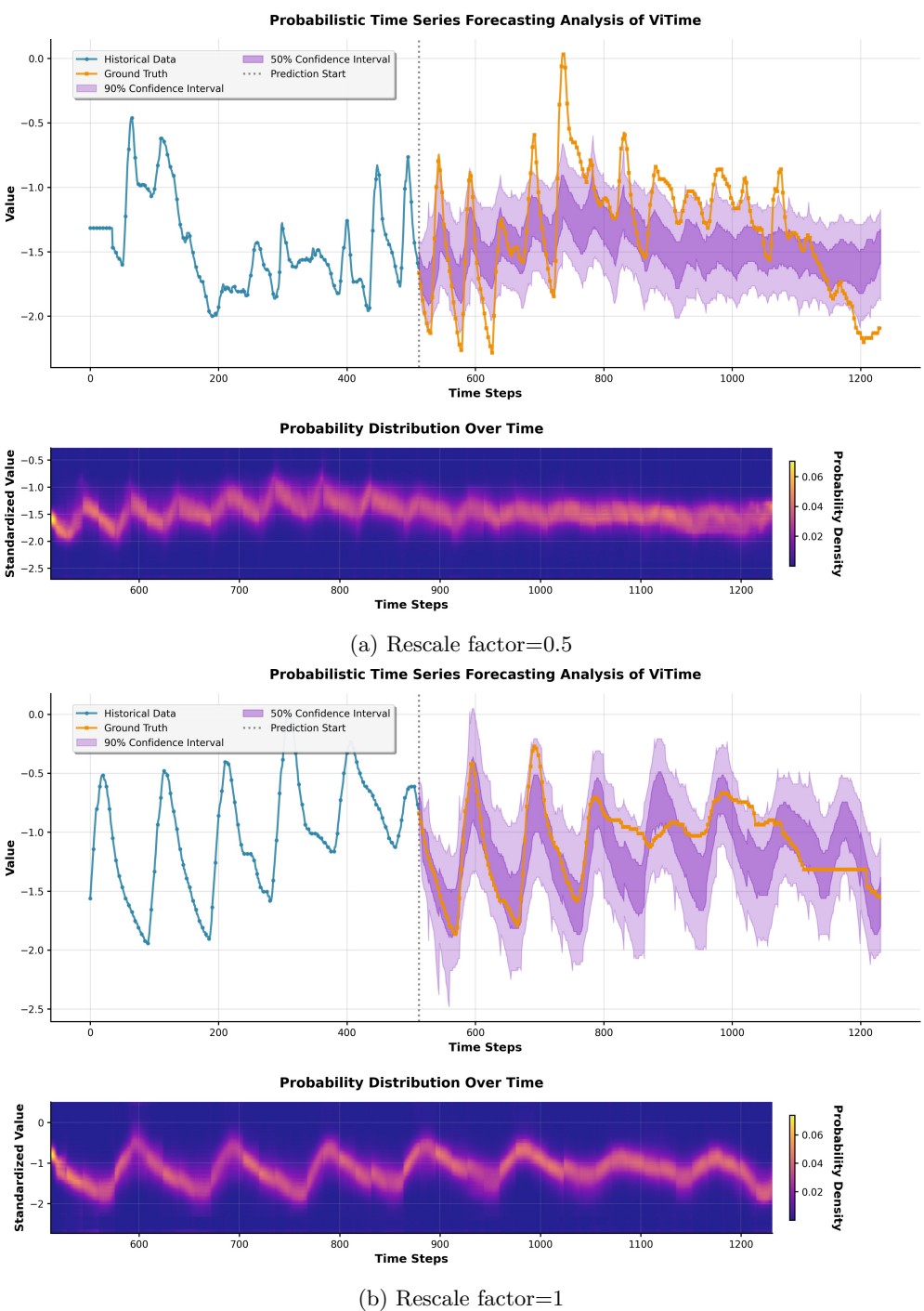

(a) Rescale factor=0.5

(b) Rescale factor=1

Figure 35: Illustrative example of dataset ETTm2 (Part 1).

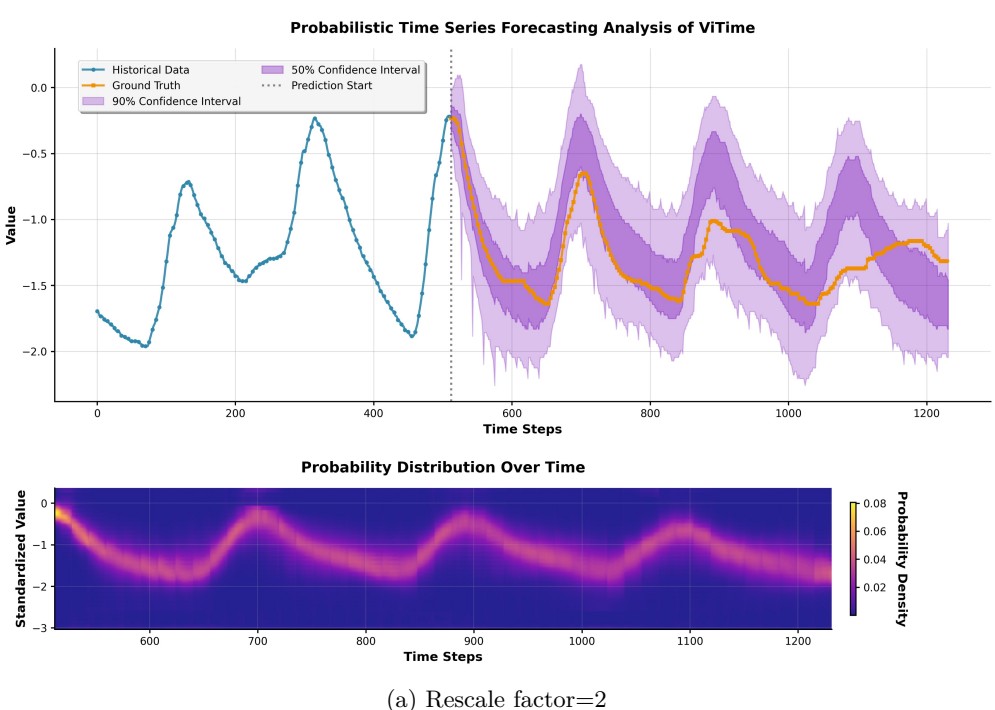

(a) Rescale factor=2

Figure 36: Illustrative example of dataset ETTm2 (Part 2).

