# OpenReview forum: "ViTime: Foundation Model for Time Series Forecasting Powered by Vision Intelligence"
_TMLR — Accepted by TMLR_

### Review · Reviewer_oM7Z · 2025-07-23

**Summary Of Contributions:**

This paper proposes a novel paradigm for time-series forecasting. Instead of performing numerical fitting in traditional value space, ViTime formulates the task in a formally defined binary image time-series metric space, enabling end-to-end visual modeling. The authors demonstrate through experiments that this approach can yield stronger pretrained models, and they introduce a dedicated synthetic data generator, RealTS, to support scalable pretraining.

**Audience:**

Yes

**Claims And Evidence:**

Yes

**Requested Changes:**

I am not an expert in time-series forecasting, but from my understanding, this paper explores an interesting direction with solid writing. I would encourage the authors to address the two issues mentioned above—particularly the unexplained zero-shot vs fine-tuning behavior and the robustness discrepancy—to improve the completeness and clarity of the work.

**Strengths And Weaknesses:**

**Strengths:**
- The paper tackles a creative and promising problem formulation, which may open up new directions for time-series research.
- The experimental setup is comprehensive, comparing against strong baselines across multiple datasets.
- Theoretical bounds on quantization error are derived, providing practical guidance for system parameter design.
- The paper is clearly written, with a logical presentation of theory and experiments.

**Weaknesses:**
- The zero-shot performance is often better than the fine-tuning setting, which is counterintuitive. For example, on the ETTh2 dataset, the MAE of TimesFM and ViTime under zero-shot is 0.389 and 0.344 respectively, but after 10% fine-tuning, they increase to 0.410 and 0.372. The paper does not provide any explanation for this unexpected phenomenon.
- In the robustness section, the authors claim that ViTime is more robust than TimesFM under Gaussian noise. However, robustness is typically defined as smaller performance degradation as noise increases. The paper mainly focuses on outperforming TimesFM, but does not explicitly analyze sensitivity to noise. According to the reported results, ViTime appears to suffer larger performance drops as the Gaussian noise level increases, which seems inconsistent with the claimed robustness. It would improve the analysis if the authors could include results under higher levels of Gaussian noise and provide a more detailed comparison of the degradation trends between ViTime and TimesFM.

---

> ### Author Response · Authors · 2025-09-22
> **Author Response (1/2)**
>
> We greatly appreciate the reviewer's thoughtful feedback and valuable recommendations. We have thoroughly addressed each concern raised and have made corresponding revisions to our manuscript. Below, we provide our detailed responses to each of your comments.
>
>
> ### **Weakness 1**
>
> **Comments:**
> The reviewer pointed out that the zero-shot performance is often better than the fine-tuning setting, which is counterintuitive. For example, on the ETTh2 dataset, the MAE of TimesFM and ViTime under zero-shot is 0.389 and 0.344 respectively, but after 10% fine-tuning, they increase to 0.410 and 0.372. The paper does not provide any explanation for this unexpected phenomenon.
>
> **Response:**
> We sincerely thank the reviewer for the careful review and insightful observation. This is indeed a counter-intuitive but important phenomenon, which we have now investigated in detail. Our analysis suggests that this behavior stems from a combination of the inherent difficulty of the ETT datasets and the characteristics of full-parameter fine-tuning on small data subsets.
>
> We agree with the insights shared in this [talk](https://cbergmeir.com/talks/neurips2024/), which discusses the challenges of the ETT datasets. We elaborate on this from two perspectives:
>
> *   **Data-level Challenges**: The ETT datasets (especially ETTh1/ETTh2 and ETTm1/ETTm2) possess weak learnable signals beyond their strong seasonal components. For long-horizon forecasting tasks (e.g., 720 steps, which is nearly 30 days), the problem is akin to weather-driven electricity demand forecasting. In such scenarios, making accurate, detailed predictions beyond a couple of weeks is nearly impossible. This leads to a situation where the robust seasonal priors learned during pre-training can outperform models fine-tuned on small, potentially noisy samples.
> *   **Method-level Challenges**: When fine-tuning on a very small data proportion like 10%, standard full-parameter fine-tuning is susceptible to overfitting on short-term, non-representative seasonal fluctuations and noise present in that specific data slice. This overfitting can also lead to a degree of catastrophic forgetting, which degrades the robust seasonal priors that the pre-trained model has already acquired. As the proportion of fine-tuning data increases, the model can re-estimate the stable seasonalities and data scales, leading to the observed performance recovery and eventual improvement.
>
> To substantiate this explanation and clarify the issue, we have conducted new experiments and added a visualization. We report the MAE on the ETTh2 dataset under various settings, including different fine-tuning data proportions and different fine-tuning methods (full-parameter fine-tuning, LoRA-128, and LoRA-16), averaged over multiple random seeds.
>
> The results are shown in the figure below:
> [Performance Comparison Across Fine-tuning Methods and Training Data Proportions](https://imgur.com/a/j1ySsdx)
>
> From this figure, we can draw the following conclusions:
>
> 1.  **Full Parameter Fine-tuning** (blue line), which was the method used in our original submission, clearly shows the phenomenon you pointed out. The MAE initially rises, surpassing the Zero-shot Baseline, and then consistently decreases as the data proportion increases, finally outperforming the zero-shot setting when 100% of the data is used.
> 2.  **LoRA-128 Fine-tuning** (purple line) significantly alleviates this initial performance degradation. The MAE increase is much smaller, and it achieves a better final accuracy than full fine-tuning.
> 3.  **LoRA-16 Fine-tuning** (orange line) further mitigates this issue, showing an even more stable performance curve at low data proportions.
>
> These results strongly support our hypothesis. The parameter-efficient fine-tuning methods (LoRA), by design, update far fewer parameters. This implicitly regularizes the model, preventing it from aggressively overfitting to the noise and non-representative patterns in small data subsets, thereby better preserving the valuable priors from pre-training.
>
> We have updated our revised manuscript with a detailed discussion of this phenomenon and included these new results in **Appendix D.3.1**. Thank you again for prompting this valuable addition to our work.

---

> ### Author Response · Authors · 2025-09-22
> **Author Response (2/2)**
>
> ### **Weakness 2**
>
> **Comments:**
> In the robustness section, the authors claim that ViTime is more robust than TimesFM under Gaussian noise. However, robustness is typically defined as smaller performance degradation as noise increases. The paper mainly focuses on outperforming TimesFM, but does not explicitly analyze sensitivity to noise. According to the reported results, ViTime appears to suffer larger performance drops as the Gaussian noise level increases, which seems inconsistent with the claimed robustness. It would improve the analysis if the authors could include results under higher levels of Gaussian noise and provide a more detailed comparison of the degradation trends between ViTime and TimesFM.
>
> **Response:**
> We are very grateful for your meticulous review and this valuable feedback. We acknowledge your concern that "ViTime's performance degradation appears larger, which seems inconsistent with the claim of robustness." We have performed a more detailed analysis to clarify our claim and provide a more nuanced perspective.
>
> We fully agree with your definition of robustness as "smaller performance degradation as noise increases." To provide a more comprehensive analysis, we propose to evaluate robustness from two complementary dimensions:
>
> 1.  **Absolute Robustness**: This refers to the model's ability to maintain its **absolute performance advantage** (i.e., lower error) under varying levels of noise. In our experiments, this is measured by the overall ReMAE
> 2.  **Relative Robustness**: This refers to the **rate of performance degradation** as the noise level increases. This is measured by the rate of change of ReMAE with respect to the Gaussian Noise level (dReMAE/dGN), which directly corresponds to the performance drop you highlighted.
>
> To this end, we conducted additional experiments across ten Gaussian noise levels from 0.1 to 1.0, with multiple runs for each setting to ensure reliability. We have updated our results with the mean performance, as shown in the figures below.
>
> [Average ReMAE](https://imgur.com/a/kaHpywl)
>
> *   **Analysis of Absolute Robustness (Left Figure: (a) Average ReMAE Across All Datasets)**: This plot clearly demonstrates that across all tested Gaussian noise levels (from 0.1 to 1.0), our ViTime model (purple dashed line) consistently achieves a significantly lower ReMAE than the TimesFM baseline (blue solid line). This means that whether in low-noise or high-noise environments, ViTime consistently provides more accurate predictions. We argue that this sustained performance leadership under perturbation is a critical aspect of robustness for real-world applications.
>
> *   **Analysis of Relative Robustness (Right Figure: (b) Average ReMAE Change Rate)**: This plot confirms your observation. The ReMAE change rate (dReMAE/dGN) for ViTime (purple dashed line) is indeed higher than that of TimesFM (blue solid line) across most of the noise spectrum. This indicates that ViTime's performance is more sensitive to increasing noise. From the perspective of degradation rate, TimesFM exhibits better "stability," or stronger **Relative Robustness**.
> **Our Core Conclusion:**
> We believe this phenomenon is expected. In an extreme case where noise approaches infinity, any model would fail to extract a useful signal, and their prediction errors would converge to a similarly high value.
>
> We acknowledge that ViTime is more sensitive to increases in noise (i.e., it has weaker relative robustness). However, **its superior foundational performance ensures that even after this degradation, its absolute prediction accuracy remains comprehensively better than TimesFM's across all tested noise levels.** In practice, an end-user is primarily concerned with which model delivers a more reliable result (lower ReMAE) in a given noisy environment, not just which model's performance curve is flatter. Therefore, we believe the superior **"Absolute Robustness"** demonstrated by ViTime makes it a more reliable and practical choice in noisy conditions.
> **We have made the following revisions to our paper:**
> 1.  In the robustness analysis section, we have explicitly introduced and distinguish between the concepts of "Absolute Robustness" and "Relative Robustness."
> 2.  We have incorporated these two new figures and their corresponding raw data into the revised manuscript to rigorously discuss the performance of both models along these two dimensions.
> 3.  We have refined our conclusion from a simple "ViTime is more robust" to a more precise statement, such as: "ViTime demonstrates superior absolute robustness by consistently maintaining a lower prediction error across all tested noise levels, although its performance is more sensitive to increasing noise (exhibiting lower relative robustness)."
>
> These additions can be found in **Section 4.5** of our revised manuscript. Thank you once again for your constructive feedback, which has helped us significantly improve the clarity and depth of our analysis.

---

> > ### Comment · Reviewer_oM7Z · 2025-09-30
> >
> > I appreciate the author's reply, which has addressed my concern.

---

> > > ### Author Response · Authors · 2025-09-30
> > >
> > > We are very glad to know that our reply has addressed your concerns. Thank you once again for your constructive feedback, which has helped us significantly improve the clarity and depth of our manuscript. We truly appreciate your time and contribution.

---

### Review · Reviewer_bwkY · 2025-08-09

**Summary Of Contributions:**

In this paper, the authors propose **ViTime**, a foundational model for time series modelling, powered by vision intelligence, in contrast to traditional approaches based on numerical fitting. This proceeds by converting the time series data into binary images and blurring by convolution; these processes are furthermore shown to be theoretically grounded (i.e., quantization errors vanish as resolution is increased, and blurring enhances signal-to-noise ratio). The model is trained on **RealTS**, a timeseries dataset of diverse time series patterns, proposed by the authors. Experiments demonstrate the superiority of ViTIme over various SOTA timeseries foundation models, in addition to various ablation studies.

**Audience:**

Yes

**Broader Impact Concerns:**

There are no ethical implications of the work as far as I can see.

**Claims And Evidence:**

Yes

**Requested Changes:**

- I would recommend conducting a more thorough literature search for vision-based methodologies for timeseries modelling as it is definitely not the first one. The authors should be aware of such works and place the current work in better context.
- Some figures can be improved with bigger labels. Most notably, Figure 5.
- The results in the tables only diaplay a single number, however, to demonstrate further robustness, I would recommend adding errors (e.g. standard deviation) across multiple seeds/experiments.
- Equation (17) is not very informative. Please could you be more concrete regarding the modelling process.

**Strengths And Weaknesses:**

__Strengths:__
- The idea of treating timeseries as images is an interesting methodology. In particular, the use of image-based representation of timeseries provides some interesting solutions to treating missing data, enhancing signal-to-noise ratio, etc... as discussed by the authors.
- Theoretical results on quantization error and signal-to-noise ratio enhancement are sound and provide justification for the proposed methodology.
- The RealTS framework for generating timeseries is interesting as it is both a simple framework and able to generate timeseries with quite large diversity.

__Weaknesses:__
- Looking at some of the qualitative results in Appendix D, the fit to ground truth doesn't always seem to be great. Of course, the task of predicting irregular time series is challenging; hence, it might be more practical to consider probabilistic forecasts instead of pointwise forecasts considered in this work. This is true especially when modelling time series with random walk-like behaviour (Appendix A.2.1), where it doesn't really make sense to consider deterministic forecasts as the properties can vary vastly depending on the noise realization.
- While RealTS is interesting and seems promising, I also wonder if only using five degrees of freedom (IFFTB, PWB, RWB, LGB, TWDB) leads to some bias in the learned model, due to the imposed structures. Perhaps we can augment RealTS with real timeseries data to learn and capture different characteristics in the various time series, beyond the five characteristics considered.
- The framework only seems to model timeseries of a single variable; however, many timeseries in practice have multiple variables, whose correlations can be useful to improve predictions. Can the methodology be extended to these multi-output settings?
- The authors make it sound like the use of a vision-based approach is completely novel for time-series forecasting. However, a quick search shows that there are several related works that are not cited (e.g., [1] and [2]). I would advise the authors to perform a more extensive literature search on related ideas and perhaps include a **Related works** section discussing these and how the ViTime methodology differs from them.

[1] Yunfei Du, Yin Wang, Ya Cong, Weihao Jiang, Shiliang Pu, "Long-term Time Series Forecasting with Vision Transformer", International Conference on Learning Representations (ICLR), 2024

[2] Weilin Ruan, Siru Zhong, Haomin Wen, Yuxuan Liang, "Vision-Enhanced Time Series Forecasting via Latent Diffusion Models", arXiv preprint arXiv:2502.14887, 2025

---

> ### Author Response · Authors · 2025-09-22
> **Author Response (1/4)**
>
> We sincerely thank the reviewers for their insightful feedback and constructive suggestions. We have carefully considered all comments and have revised our manuscript accordingly. Below, we provide a point-by-point response to the issues raised.
>
> ### **Weakness 1**
>
> **Comments:** The qualitative results show that the model's fit is not always perfect. The reviewer suggests that probabilistic forecasts might be more practical than the pointwise forecasts considered, especially for time series with random walk-like behavior.
>
> **Response:**
> We thank the reviewer for this insightful suggestion. We agree that probabilistic forecasting is crucial for handling the inherent uncertainty in time series, particularly for challenging irregular patterns.
>
> We would like to clarify that our proposed model, ViTime, is inherently a probabilistic forecasting model. The joint EMD and JSD training loss is applied to an image-like output, where the vertical dimension (`h`) represents a learned probability distribution for the value at each future time step. Our current pointwise forecasts, as described in Equation 5 of the original manuscript, are derived by calculating the expected value of this probability distribution. We have updated Section 3.6 of the manuscript to provide a more detailed explanation of this process.
>
> To explicitly demonstrate this capability and address the reviewer's comment, we have added a new subsection on probabilistic zero-shot forecasting (Section 4.3 in the revised manuscript). In this new section, we evaluate ViTime against strong probabilistic forecasting baselines, including Moriai-Large and Lag-Llama, using standard probabilistic metrics: Continuous Ranked Probability Score (CRPS) and ReCRPS.
>
> The results, presented in the table below, show that ViTime achieves state-of-the-art performance in probabilistic forecasting, consistent with its strong results in the pointwise setting.
>
> #### Table 1: Comparison of probabilistic forecasting performance
> *Within each scenario, the best results (lower is better) for each dataset are **bolded**.*
>
> | Method | ETTh1 | ETTh2 | ETTm1 | ETTm2 | Electricity | Traffic | Weather |
> |--------|-------|-------|-------|-------|-------------|---------|---------|
> | **CRPS** |
> | Moriai | 0.506 | **0.274** | 0.538 | 0.309 | 0.522 | 0.626 | 0.506 |
> | Lag-Llama | 0.441 | 0.401 | 0.435 | 0.432 | 0.470 | 0.548 | 0.394 |
> | ViTime | **0.356** | 0.319 | **0.344** | **0.286** | **0.267** | **0.327** | **0.244** |
> | **ReCRPS** |
> | Moriai | 0.537 | 0.382 | 0.631 | 0.329 | 0.609 | 0.684 | 0.620 |
> | Lag-Llama | 0.478 | 0.442 | 0.463 | 0.420 | 0.514 | 0.629 | 0.377 |
> | ViTime | **0.358** | **0.318** | **0.346** | **0.283** | **0.266** | **0.324** | **0.241** |
>
> These results confirm that ViTime not only excels in zero-shot point forecasting but also maintains high accuracy in the more challenging probabilistic forecasting task. Meanwhile, we also provide probabilistic forecasting visualization results in Appendix.
>
> ### **Weakness 2**
>
> **Comments:** The reviewer expressed concern that using only five degrees of freedom in the RealTS dataset might introduce structural bias into the learned model and suggested augmenting the training data with real time series.
>
> **Response:**
> We highly appreciate the reviewer's thoughtful question regarding the design of RealTS. Our primary goal with ViTime is to develop a robust foundation model for time series analysis. The intended use case is for practitioners to take our pre-trained model and fine-tune it on their specific, smaller, in-domain datasets to achieve state-of-the-art performance, as demonstrated in our fine-tuning experiments - real time series data got involved in fine-tuning process.
>
> The decision to exclusively use synthetic data for pre-training was a deliberate design choice to prevent data leakage and ensure a fair and rigorous evaluation of the model's zero-shot generalization capabilities. Including real-world datasets in the pre-training phase could inadvertently expose the model to the test data distributions, compromising the integrity of the evaluation. For instance, as observed with models like TimesFM, performance can degrade significantly when the test data undergoes simple transformations like scaling, which may indicate a dependency on the specific characteristics of the pre-training data. By building RealTS from fundamental time series components, we aim to create a model that learns generalizable temporal patterns rather than memorizing artifacts of specific datasets.

---

> ### Author Response · Authors · 2025-09-22
> **Author Response (2/4)**
>
> ### **W4 & R1**
>
> **Comments:** Incomplete literature review.
>
> **Response:**
> We thank the reviewer for this valuable feedback. We have conducted a comprehensive literature review and revised the introduction and related work sections in the updated manuscript to highlight the unique contributions of ViTime.
>
> Regarding the specific references suggested by the reviewer, we noted that they have not undergone formal peer review. For instance, reference [1] is an ICLR submission that was not accepted, and reference [2] does not appear to have been formally reviewed. Therefore, while we discuss the innovative ideas they propose in our new literature review, we have refrained from direct experimental comparisons, which is standard practice for non-peer-reviewed work.
>
> The following is the new content of "Related Works":
>
> **Vision-based Models:** The idea of representing time series as images to leverage powerful vision models is not entirely new, but its application as a primary forecasting paradigm is an area of growing interest. Early methods, such as Gramian Angular Fields (GAF) [3], Markov Transition Fields (MTF), and Recurrence Plots [4], transformed time series into images primarily for classification tasks, demonstrating the potential of image representations to reveal patterns not obvious in the 1D domain. Recently, this concept has been extended to forecasting. One intuitive approach is direct plotting; VisualAE [5] pioneered this by treating TSF as an image-to-image regression task, where a line plot is converted by a convolutional autoencoder. Other works focus on adapting vision architectures; Swin4TS [1] reshapes the time series into 2D patches to apply the Swin Transformer. To enrich the input, models like LDM4TS [2] employ a multi-view strategy, converting a time series into multiple images using techniques like segmentation and GAFs. Recently, VisionTS [6] proposed repurposing pretrained vision models (specifically, masked autoencoders trained on ImageNet) for TSF by reformulating forecasting as an image reconstruction problem. Nevertheless, directly reusing models pretrained on natural images introduces a significant domain mismatch; the visual features learned from natural images may not optimally represent temporal structures inherent in numerical time series. Furthermore, all these existing vision-based methods still fundamentally rely on numerical-space analyses and empirical mappings (e.g., standard plotting libraries or heuristic reshaping), lacking a rigorous theoretical framework explicitly tailored for visual representation and quantization of numerical sequences.
>
> **Our Contribution in Context:** In contrast to aforementioned paradigms, our proposed ViTime framework introduces two fundamental shifts in the TSF foundation model design:
>
> Firstly, recognizing intrinsic limitations of numerical-space-based forecasting--such as poor generalization across scales and sensitivity to data perturbations--ViTime explicitly advocates modeling time series directly in a principled visual representation space. Where prior visual methods use heuristic transformations, ViTime is the first to rigorously define a dedicated visual metric space for numerical time series, providing theoretical analysis of quantization-induced errors, and offering principled guidance for optimal parameter selection. We also proved that this rigorous visual modeling framework can significantly enhance the signal-to-noise ratio (SNR) of time series and improve forecasting accuracy and interpretability.
>
> Secondly, given the inherent challenges of relying on real-world numerical datasets (limited diversity, data leakage risks), we propose, RealTS, a controlled data synthesis strategy focusing on fundamental time series components (trend, periodicity) to generate structurally sound training data. The RealTS substantially mitigates data leakage risks and enriches training data diversity, enabling ViTime to generalize robustly across diverse real-world scenarios. As demonstrated by extensive experiments, ViTime sets new SOTA zero-shot and limited-data forecasting benchmarks, significantly outperforming existing foundation models across diverse evaluation settings.
>
> #### References
>
> [1] Long-term Time Series Forecasting with Vision Transformer, Openreview.net, 2024.
>
> [2] Vision-Enhanced Time Series Forecasting via Latent Diffusion Models, arXiv preprint arXiv:2502.14887, 2025.
>
> [3] Encoding time series as images for visual inspection and classification using tiled convolutional neural networks, Workshops AAAI, 2015
>
> [4] Recurrence plots of dynamical systems, Turbulence, strange attractors and chaos. 1995
>
> [5] Visual time series forecasting: an image-driven approach, ACM International Conference on AI in Finance. 2021
>
> [6] Visionts: Visual masked autoencoders are free-lunch zero-shot time series forecasters, arXiv preprint arXiv:2408.17253, 2024.

---

> ### Author Response · Authors · 2025-09-22
> **Author Response (3/4)**
>
> ### **Weakness 3**
>
> **Comments:** The reviewer correctly noted that the current framework is designed for univariate time series and questioned its applicability to multivariate settings, where inter-variable correlations are often crucial.
>
> **Response:**
> The reviewer raises an excellent point about the extension to multivariate time series. In theory, our framework can be extended to handle multivariate inputs, for example, by treating different variables as different channels of the input image.
>
> However, this extension presents two significant challenges. First, increasing the number of channels would dramatically increase the computational and memory requirements, making the model difficult to train. Secondly, and more fundamentally, generating realistic synthetic multivariate time series with coupled, meaningful correlations between variables is a non-trivial research problem in itself.
>
> We acknowledge this as a current limitation of our work and a valuable direction for future research. We have updated the future work section in our revised manuscript to reflect this point.
>
> ### **Requested Change 2**
>
> **Comments:** The labels in some figures, particularly Figure 5, are too small and difficult to read.
>
> **Response:**
> Thank you for pointing this out. We have revised the relevant figures in the manuscript, including Figure 5, to have larger and clearer labels for improved readability.
>
> ### **Requested Change 3**
>
> **Comments:** The results in the tables should include error metrics (e.g., standard deviation) across multiple runs to demonstrate robustness.
>
> **Response:**
> We thank the reviewer for this important suggestion. We agree that reporting variance is crucial for demonstrating the robustness of our results. We have performed additional experiments and updated our tables as follows:
>
> 1. **Zero-Shot (ZS) Experiments:** For ZS inference, the model is already trained, and the inference process is deterministic. We ran the inference process 10 times for both ViTime and TimesFM and found that the standard deviation was negligible (affecting only the fifth decimal place or beyond). Therefore, to maintain clarity in the tables, we have not added error bars for the ZS results, as they would not provide meaningful information.
>
> 2. **Fine-Tuning (FT) Experiments:** We agree that the stochasticity of the training process makes error bars essential here. We have fine-tuned ViTime three times using different random seeds and have updated the table to include the mean and standard deviation. For the baseline models, we continue to report the results from their original papers, as re-running these computationally intensive experiments is beyond the scope of this rebuttal, a common practice in such cases. The updated table is below.
>
> ### Table 2: Comparison of Fine-tuning forecasting results with MAE
>
> | Method | Data proportion | ETTh1 | ETTh2 | ETTm1 | ETTm2 | Electricity | Traffic | Weather |
> |--------|----------------|-------|-------|-------|-------|-------------|---------|---------|
> | TimesFM (FT) | 10% | 0.426 | 0.410 | 0.388 | 0.334 | - | - | - |
> | GPT4TS (FT) | 10% | 0.542 | 0.431 | 0.466 | 0.343 | Not Reported | Not Reported | Not Reported |
> | TIME-LLM (FT) | 10% | 0.522 | 0.394 | 0.426 | 0.323 | - | - | - |
> | **ViTime (FT)** | **10%** | 0.422 (±0.034) | 0.370 (±0.007) | 0.376 (±0.003) | 0.312 (±0.008) | 0.250 (±0.008) | 0.251 (±0.008) | 0.249 (±0.005) |
> | PatchTST | 10% | 0.542 | 0.431 | 0.466 | 0.343 | 0.268 | 0.286 | 0.283 |
> | PatchTST | 100% | 0.434 | 0.381 | 0.382 | 0.317 | 0.253 | 0.264 | 0.264 |
> | SiMBA | 100% | 0.433 | 0.392 | 0.396 | 0.328 | 0.274 | 0.291 | 0.281 |
> | TIMESNET | 100% | 0.450 | 0.427 | 0.406 | 0.333 | 0.295 | 0.336 | 0.286 |
> | iTransformer | 100% | 0.448 | 0.407 | 0.410 | 0.332 | 0.270 | 0.282 | 0.278 |
> | TimeMixer | 100% | 0.423 | 0.384 | 0.376 | 0.316 | 0.246 | 0.263 | 0.262 |
> | **ViTime (FT)** | **100%** | **0.406** (±0.039) | **0.344** (±0.004) | **0.366** (±0.003) | **0.297** (±0.017) | **0.245** (±0.004) | **0.248** ±0.005) | **0.249** (±0.004) |
>
> 3. **Robust Inference Study:** In these experiments, randomness is introduced via Gaussian noise or data masking. We have run these experiments for both ViTime and TimesFM across three different random seeds and have updated the table with the mean and standard deviation.
>
> Please refer to the **table 3** in Author Response (4/4)

---

> ### Author Response · Authors · 2025-09-22
> **Author Response (4/4)**
>
> ### Table 3: Comparison of average ReMAE forecasting results under noise
> | Method | ETTh1 | ETTh2 | ETTm1 | ETTm2 | Electricity | Traffic | Weather |
> |--------|-------|-------|-------|-------|-------------|---------|---------|
> | **GN standard deviations = 0.1** |
> | TimesFM | 0.471±0.002 | 0.393±0.001 | 0.495±0.012 | 0.352±0.005 | 0.403±0.004 | 0.512±0.016 | 0.280±0.005 |
> | ViTime | **0.449±0.002** | **0.360±0.001** | **0.429±0.001** | **0.329±0.002** | **0.340±0.006** | **0.402±0.014** | **0.279±0.007** |
> | **GN standard deviations = 0.3** |
> | TimesFM | 0.473±0.007 | 0.390±0.002 | 0.485±0.010 | 0.344±0.004 | 0.414±0.006 | 0.518±0.011 | **0.288±0.005** |
> | ViTime | **0.455±0.005** | **0.363±0.001** | **0.434±0.002** | **0.333±0.003** | **0.355±0.007** | **0.426±0.016** | 0.288±0.009 |
> | **GN standard deviations = 0.5** |
> | TimesFM | 0.479±0.006 | 0.392±0.003 | 0.488±0.009 | 0.345±0.005 | 0.433±0.005 | 0.529±0.012 | 0.295±0.005 |
> | ViTime | **0.466±0.003** | **0.370±0.002** | **0.445±0.003** | **0.337±0.003** | **0.371±0.007** | **0.461±0.013** | **0.294±0.013** |
> | **GN standard deviations = 0.7** |
> | TimesFM | **0.483±0.009** | 0.394±0.003 | 0.492±0.006 | 0.349±0.005 | 0.450±0.005 | 0.543±0.015 | 0.302±0.005 |
> | ViTime | 0.484±0.004 | **0.394±0.003** | **0.443±0.003** | **0.346±0.005** | **0.377±0.006** | **0.510±0.021** | **0.301±0.014** |
> | **GN standard deviations = 1.0** |
> | TimesFM | **0.487±0.013** | **0.399±0.003** | 0.500±0.009 | 0.359±0.006 | 0.475±0.006 | 0.567±0.007 | 0.312±0.007 |
> | ViTime | 0.487±0.006 | 0.408±0.005 | **0.471±0.004** | **0.358±0.005** | **0.415±0.010** | **0.546±0.022** | **0.305±0.010** |
> | **DM P = 0.3** |
> | ViTime | **0.453±0.002** | **0.378±0.001** | **0.432±0.001** | **0.337±0.003** | **0.343±0.006** | **0.417±0.014** | **0.281±0.013** |
> ### **Requested Change 4**
>
> **Comments:** The description of the modeling process and Equation (17) was found to be uninformative and in need of more concrete details.
>
> **Response:**
> We thank the reviewer for highlighting the lack of clarity. We have completely rewritten this section in the revised manuscript to provide a more concrete and comprehensive explanation of ViTime's modeling process, masking strategy, and loss function. Please find the updated text below.
>
> **Modeling process and masking.**
> The modeling process of ViTime is summarized as
> $$\mathbf{v}\_{\mathbf{L}}^{\prime} = \text{ViTime}\\left( \mathbf{v}\_{\mathbf{L}} \odot \mathbf{M}\_{\mathbf{L}} \right),$$
> where $\odot$ is the element-wise product and $\mathbf{M}\_\{\mathbf{L}\}$ is a temporal mask that zeros out the time steps to be forecast. Concretely, we use $\mathbf{M}\_\{\mathbf{L}\}\in\\\{0,1\\\}^{1\times 1\times L}$ (broadcast along $c$ and $h$) with
> $$(\mathbf{M}\_\{\mathbf{L}\})\_\{1,1,k\} = \begin{cases} 1, & k \in \text{observed (context) time indices},\\\\ 0, & k \in \text{forecast horizon (to be predicted)}. \end{cases}$$
> Thus, for all $i\in[c], j\in[h], k\in[L]$,
>
> $$(\mathbf{v}\_\{\mathbf{L}\} \odot \mathbf{M}\_\{\mathbf{L}\})\_\{i,j,k\} = \begin{cases} \mathbf{v}\_\{i,j,k\}, & \mathbf{M}\_\{1,1,k\}=1,\\\\ 0, & \mathbf{M}\_\{1,1,k\}=0, \end{cases}$$
>
> i.e., the mask sets the to-be-predicted time positions to *all zeros across the j-axis*, removing any target information at those steps. Note that this masking operates on the input only; although the masked columns are no longer one-hot (and hence leave $\mathcal{V}$ at those $k$), the network outputs valid distributions $\mathbf{p}\in\mathcal{P}$ over $j$ for every $(i,k)$ (see Section 3.4). During training, we sample masked spans to simulate forecasting; at inference, we set $\mathbf{M}_{\mathbf{L}}$ to zero precisely on the forecast horizon.

---

### Review · Reviewer_9Dt3 · 2025-09-12

**Summary Of Contributions:**

This paper presents a methodology to train a time-series foundation model by transforming numeric time-series data into visual inputs. The key contributions lie in:
1. Transforming numeric sequences into images.
2. Creating synthetic time-series data with diversity in trend and periodicity.
3. Training the vision transformer model to predict the visual "patch" tokens and then converting it back into numeric time-series data.

**Audience:**

Yes

**Claims And Evidence:**

No

**Requested Changes:**

major: see weakness comments above.

I am unsure about the validity of the results without knowing the model/data recipe design. It would be great if the authors could provide information about:
1. ViTime model design: layers, encodings, depth, etc.
2. data recipes used to train the model, i.e, the fraction of TS data created for trend/periodicity etc.
3. experimental settings to obtain the benchmark performance of various models considered (ex: for Table 2)
4. The details of the ViTime-TFM data recipe and training time/resources used.

**Strengths And Weaknesses:**

**Strengths:**
1. The paper introduces an interesting idea to model time-series (TS) data as images and leverage the ViT (vision transformer) model's ability for effective zero/few-shot prediction on new tasks.
2. The effectiveness of synthetic data is highlighted, which also conveys a message that one does not have to rely heavily on real-world time-series data to train such models effectively.
3. The role of quantizing the TS data and the parameters involved is studied in detail.
4. The performance (ReMSE, ReMAE) of the ViTime model is consistently better than other approaches across the datasets considered.

**Weaknesses:**

1. Details about the model architecture, computational resources, and optimizers used for training are not discussed. (Please let me know if I missed anything). This is important since Table 15 shows that the inference latency of ViTime is quite lower than TimesFM while having better performance with a smaller number of parameters. Do these inference numbers consider the overhead of converting TS data to images?
2. In Table 2(b), how can ViTime outperform ViTime-TFM when the data used for ViTime-TFM is much more diverse? Details about this dataset is also missing.
3. The authors mention that the TS -> image conversion process can act as a denoiser and lead to better performance. What happens if we can apply some standard denoising techniques on the TS data itself before passing it through, say, TimesFM?

---

> ### Author Response · Authors · 2025-09-22
> **Author Response (1/2)**
>
> We sincerely thank the reviewer for your insightful comments and constructive suggestions. We have carefully considered all the points raised and have revised our manuscript accordingly. Our point-by-point responses to your comments are offered as follows.
>
> ### **Weakness 1 & Concern 1**
> **Comments:** Missing details on model architecture, computational resources, optimizers, and inference latency calculation.
>
> **Response:**
> Thank you for your valuable feedback. We have revised our manuscript to better provide details mentioned in your comment, please refer to our following responses:
>
> 1.  **Model Architecture:** We provided details of our model architecture in Appendix B.1, Table 7 of the original manuscript (Appendix B.1, Table 8 of the revised manuscript). This table included structural information for the Visual Time Tokenizer and the Decoder. To further enhance clarity, we have now updated this table to also include the specifications of the **Refining Module**, which is a standard DeeplabV3+ model with a MobileNetV2 backbone. The updated table, Table 8 in the revised manuscript, is as follows:
>
>     **Table 7: Details of model architecture.**
>     | Module | Embed_dim | Depth | Patch size | Num_heads |
>     | :--- | :--- | :--- | :--- | :--- |
>     | Visual Time Tokenizer | 768 | 9 | (4,32) | 12 |
>     | Decoder | 384 | 4 | \ | 12 |
>
>     **Refining Module**
>     | Component | Configuration | Details |
>     | :--- | :--- | :--- |
>     | Backbone | MobileNetV2 | Downsample factor: 16 |
>     | ASPP | 5 branches | Dilation rates: 1,6,12,18,GAP |
>     | Low-level features | Conv1x1 | 24→48 channels |
>     | Feature fusion | Conv3x3×2 | 256 channels |
>     | Upsampling | DeconvASPP | 6 branches with SE attention |
>
>     For more comprehensive details, we invite the reviewer to consult our supplementary code, which has been provided with our submission.
>
> 2.  **Computational Resources and Inference Time:** We discussed our training hardware and analyzed the model's performance versus computational requirements in Appendix D.4, Table 15 of the original submission. We would like to clarify that the reported **Inference Time (s/batch)** explicitly **included the overhead from the time series-to-image mapping and its inverse mapping (Map. & Inv. Map.)**. On our test device, this conversion overhead accounts for approximately 0.2% of the total inference time for ViTime (w/ Refining) and 8% for ViTime (w/o Refining), demonstrating its minimal impact on overall latency.
>
> 3.  **Optimizer:** During training, we utilized the **Adam optimizer** with a learning rate of **2e-4**. We have added this information to the revised manuscript to ensure reproducibility.
>
> ### **Weakness 2 & Concern 4**
> **Comments:** Confusion about ViTime's superior performance over ViTime-TFM and a lack of details on the dataset and training for ViTime-TFM.
>
> **Response:**
> Thank you for this insightful question. We would like to clarify a key point regarding the training data used for ViTime and ViTime-TFM.
>
> The two models were trained on **entirely different datasets**. ViTime-TFM was trained using the same publicly available data as the original TimesFM, as detailed in the table below. In contrast, our proposed ViTime model was trained on a synthesized dataset generated by our **RealTS** framework.
>
> **Table: Training Data for ViTime-TFM (from TimesFM).**
> | Dataset | Granularity | # Time series | # Time points |
> | :--- | :--- | :--- | :--- |
> | Synthetic | Hourly | 3,000,000 | 6,144,000,000 |
> | Electricity | Hourly | 321 | 8,443,584 |
> | Traffic | Hourly | 862 | 15,122,928 |
> | Weather | 10 Min | 42 | 2,213,232 |
> | Favorita Sales | Daily | 111,840 | 139,179,538 |
> | LibCity | 15 Min | 6,159 | 34,253,622 |
> | M4 hourly | Hourly | 414 | 353,500 |
> | M4 daily | Daily | 4,227 | 9,964,658 |
> | M4 monthly | Monthly | 48,000 | 10,382,411 |
> | M4 quarterly | Quarterly | 24,000 | 2,214,108 |
> | M4 yearly | Yearly | 22,739 | 840,644 |
>
> The superior performance of ViTime stems from the fact that RealTS generates data with significantly richer and more diverse temporal patterns. This leads to better model generalization compared to ViTime-TFM, which was trained on a fixed dataset. We have updated the revised manuscript to make this distinction clear.
>
> Regarding training details, **ViTime-TFM** utilizes the identical model architecture and training process as ViTime. It was trained for approximately 500 epochs on a single NVIDIA 3090 GPU, with convergence achieved in about one week.

---

> ### Author Response · Authors · 2025-09-22
> **Author Response (2/2)**
>
> ### **Weakness 3**
> **Comments:** How does ViTime's implicit denoising compare to applying standard denoising techniques on the raw time series data before feeding it to another model like TimesFM?
>
> **Response:**
> This is an excellent point, and we thank the reviewer for the suggestion. To address this, we conducted a new experiment on the Electricity dataset with an added Gaussian noise level (variance = 0.5). We applied three common denoising pre-processing techniques to the noisy data before feeding it into the TimesFM model:
> 1.  **Savitzky-Golay Filter:** window_length=21, polyorder=3
> 2.  **Moving Average:** window_size=5
> 3.  **3-Sigma Rule**
>
> The results are presented in the [figure](https://imgur.com/a/NuGOQeK).
>
>
> Our findings indicate that the effectiveness of these methods varies significantly with the prediction horizon.
> *   The **3-sigma** method shows good performance for short prediction horizons, but its efficacy diminishes as the horizon increases.
> *   Conversely, the **moving average** and **Savitzky-Golay** filters yield substantial improvements for long-sequence forecasting but can be detrimental to short-term predictions. This is likely because these smoothing methods capture the broader trends beneficial for long-term forecasting while filtering out the fine-grained variations crucial for short-term accuracy.
>
> Crucially, the performance of ViTime (represented by the baseline in the figure) is consistently superior to TimesFM combined with any of these pre-processing techniques, across all prediction horizons. This experiment demonstrates the **absolute robustness** of ViTime to noise and validates that our time-series-to-image conversion acts as a more effective and adaptive denoising mechanism than standard, one-size-fits-all pre-processing filters. For more details, please refer to Appendix section **D.4.1** of the revised manuscript.
>
> ### **Concern 2**
> **Comments:** Missing details on the data recipes used to train the model, specifically the mixture proportions of different time series components generated by RealTS.
>
> **Response:**
> We thank the reviewer for this question regarding the composition of our synthetic training data. To generate a diverse dataset, we sample from the different components of our RealTS framework—IFFTBI, IFFTBII, PWB, RWB, LGB, and TWDB—using a specific probability distribution. The sampling probabilities are as follows: **[0.3, 0.3, 0.16, 0.08, 0.08, 0.08]** for IFFTBI, IFFTBII, PWB, RWB, LGB, and TWDB, respectively. We have updated the revised manuscript to include these details, ensuring greater transparency and reproducibility of our data generation process.
>
> ### **Concern 3**
> **Comments:** Lack of clarity on the experimental settings used to obtain the performance of benchmark models.
>
> **Response:**
> Thank you for pointing out the need for more detailed experimental settings. We apologize for this omission and provide the specific configurations for the baseline models below.
>
> 1.  For the **zero-shot forecasting results in Table 2**, we used the following pre-trained models:
>     *   **TimesFM:** `timesfm-1.0-200m`
>     *   **Moirai:** `moirai-1.1-R-large`
>     *   **Moment:** `MOMENT-1-large`
>     *   **VisionTS:** `mae_base`
>     *   **PatchTST:** `PatchTST/64`
>
> 2.  For the **fine-tuning results presented in Table 4** (previously Table 3 in the original manuscript), the reported scores are the official results cited directly from the respective original publications of the baseline models.
>
> 3.  For our newly added **probabilistic forecasting experiments in Table 3** of the revised manuscript, we configured the baseline models as follows:
>     *   **Moirai:** `moirai-1.1-R-large`
>     *   **Lag-Llama:** We used the publicly available implementation from the original Lag-Llama paper and its provided checkpoint, `lag-llama.ckpt`.
>
> We have incorporated this information into the appendix of our revised manuscript to ensure that our experimental setup is fully transparent and reproducible.

---

> ### Comment · Reviewer_9Dt3 · 2025-09-30
>
> Thank you for addressing the concerns. I have a few follow-up questions:
>
> 1. What would be a good intuition to understand the effectiveness of ViTime for such multi-scale denoising?
> 2. (No experiment required) Do you think combining the TimesFM dataset along with the RealTS synthetic data would be much more effective? I'm curious why this wasn't explored.

---

> > ### Author Response · Authors · 2025-09-30
> > **Author Response (1/2)**
> >
> > We sincerely thank the reviewer for your insightful comments and constructive suggestions. Our point-by-point responses to your comments are offered as follows.
> >
> > ### **Comment 1: Intuition for Multi-Scale Denoising**
> >
> > **Response:**
> > We sincerely thank the reviewer for this insightful question. The effectiveness of ViTime in multi-scale denoising stems from its fundamental paradigm shift: **from fitting numerical values to recognizing structural patterns in a visual space.** This shift makes the model inherently be more robust to noise, which is often unstructured and localized, especially when viewed across different time scales.
> >
> > Here is a breakdown of the core intuitions:
> >
> > **1. Paradigm Shift: From Numerical Regression to Visual Pattern Recognition**
> > *   **Intuition:** Traditional numerical models treat every data point, including noise, as a value to be regressed. A single large outlier or high-frequency noise can significantly distort the learned function, a phenomenon we term "numerical fragility." In contrast, ViTime converts the time series into a binary image. In this visual domain, the core signal (e.g., a trend or seasonal cycle) manifests as a **coherent, structural pattern** (like a thick, continuous stripe), while noise appears as **isolated, sporadic pixel flips** or a "salt-and-pepper" texture.
> > *   **Analogy:** Imagine a photograph of a clean, straight line. A numerical model would attempt to fit a perfect `y=mx+c`. If we sprinkle random noisy pixels onto the photo, the numerical model's fit could be drastically skewed. However, the human visual system (and by extension, a powerful Vision Transformer) would instantly disregard the "snow" and still perceive the underlying straight line. ViTime operates on this latter principle, focusing on the **structural integrity** of the pattern rather than the exact value of each point. This makes it naturally robust to local perturbations (i.e., noise).
> >
> > **2. Quantization as an Inherent Denoising Filter**
> > *   **Intuition:** The process of mapping a continuous time series value to a discrete pixel row (quantization) acts as an implicit, non-linear filter. Small-magnitude noise that does not push the value across a quantization boundary is **effectively ignored**, as it results in the same pixel representation. Only noise significant enough to "flip" a pixel to an adjacent row impacts the model's input.
> > *   **Connection to Theory:** This intuition is formally grounded in our theoretical analysis. **Theorem 3.7 (Stripe SNR Boost)** demonstrates that this visual representation can significantly increase the signal-to-noise ratio (SNR) precisely because the signal's energy is concentrated into a structured pattern, while the noise energy is diffused. This process inherently suppresses low-amplitude noise.

---

> > ### Author Response · Authors · 2025-09-30
> > **Author Response (2/2)**
> >
> > ### **Comment 2: Pre-training on Synthetic vs. Real-world Data**
> >
> > **Response:**
> > We thank the reviewer for this excellent and thought-provoking question. The decision to pre-train exclusively on synthetic data was a deliberate strategic choice, guided by three primary considerations:
> >
> > **1. Core Objective: Building a General-Purpose Foundation Model**
> > Our primary goal with ViTime is not to deliver a final, specialized solution for a specific domain, but rather to develop a versatile and robust **foundation model** for the broader community. The core value of our model lies in its adaptability. We posit that users from diverse fields can leverage our pre-trained model and, with minimal fine-tuning on their domain-specific data, achieve highly competitive performance. This "pre-train on general principles, fine-tune on specific tasks" paradigm is designed to lower the barrier to entry for applying advanced time series analysis across various applications.
> >
> > **2. Mitigating Risks of Data Contamination and Overfitting**
> > Incorporating large-scale real-world datasets like TimesFM datasets into pre-training introduces significant methodological risks that conflict with our goal of building a truly generalizable model:
> > *   **Risk of Unintentional Data Leakage:** Pre-training on a massive corpus of real-world time series, whose composition and relationship to standard benchmark datasets are not fully known, creates a risk of data contamination. The model might inadvertently "see" data with similar statistical properties to the test sets, leading to inflated performance metrics. This would compromise a fair and objective evaluation of the model's true **zero-shot generalization capability**, which is a key claim of our work.
> > *   **Risk of Compromising Generalization Robustness:** Over-relying on a specific (even if large) real-world dataset can cause the model to overfit to its inherent biases and statistical properties. As a cautionary example, models heavily trained on specific data distributions can become brittle and exhibit significant performance degradation when faced with slight domain shifts (e.g., changes in scale, frequency, or noise characteristics). By pre-training on the controlled and diverse distributions of our RealTS synthetic data, we aim to learn more fundamental and invariant patterns of time series, thereby fostering a more robust and widely applicable model.
> >
> > **3. Validated Efficacy of the Fine-Tuning Pathway**
> > Our experiments have already demonstrated the effectiveness of our chosen approach. The strong performance achieved through fine-tuning (the FT results in our paper) validates that the foundational knowledge learned from the RealTS dataset provides an excellent starting point for rapid adaptation to a wide array of real-world tasks. This confirms that pre-training on real-world data is not a prerequisite for achieving high performance in specific downstream applications.
> >
> > In summary, while adding real-world data to pre-training might appear to boost performance on certain benchmarks, we believe our current strategy of using pure synthetic data leads to a more generalizable, robust, and methodologically sound foundation model, whose value in downstream applications can be realized through further effective and efficient fine-tuning.

---

### Decision · Action_Editor_sqaM · 2025-09-30

**Recommendation:** Accept with minor revision

**Additional Comments:**

The experiments were conducted on relatively small time series datasets, which might not support well the claim of foundation models. Recently larger and more rigorous evaluation datasets have been developed, such as GIFT-EVAL or BOOM, with all top time series foundation models publishing the results on leaderboard. For the revision, the results of the proposed model on GIFT-EVAL and/or BOOM should be added.

**Audience:**

Yes

**Audience Explanation:**

The paper would be of interest to those who work on time series and its applications.

**Claims And Evidence:**

Yes

**Claims Explanation:**

The proposed foundation model is well-motivated. Experiment results on some time series datasets demonstrates the effectiveness of the proposed model. All reviews agree that the work is solid and interesting.

---

> ### Author Response · Authors · 2025-10-22
> **Author Response**
>
> We are grateful for your invaluable guidance throughout the review process and for your constructive final comments. In the following, we address the specific points you raised and detail the corresponding revisions in the newly uploaded camera-ready manuscript.
>
> The primary concern raised was the need for evaluation on a larger, more rigorous benchmark to substantiate ViTime's capabilities as a foundation model. We agree with this assessment and, as suggested, have conducted an extensive new evaluation on the **GIFT-EVAL** benchmark. We hope that our new results can directly address your concerns to a satisfactory level and significantly strengthen the quality of this paper.
>
>
>
> To validate ViTime's generalization capabilities, we performed a comprehensive zero-shot evaluation on GIFT-EVAL, a widely recognized and challenging benchmark for time series foundation models. We chose GIFT-EVAL for its community adoption and established leaderboard, which allows for a direct and fair comparison against the current state-of-the-art, including the leading **TimesFM** model.
>
>
> Our initial zero-shot evaluation reveals that ViTime significantly outperforms TimesFM on medium- and long-range forecasting tasks. The results below (MASE, lower is better) highlight ViTime's strong generalization. (The results of TimesFM are obtained via [gift-eval offical code repository](https://github.com/SalesforceAIResearch/gift-eval/tree/main/results/timesfm))
>
> **Table 1: Zero-Shot Performance on the Full GIFT-EVAL Benchmark**
> | Forecast Horizon | TimesFM (MASE) | ViTime (MASE) | Relative Improvement |
> | :--- | :--- | :--- | :--- |
> | **Long** | 2.3706 | **1.4636** | **38.3%** |
> | **Medium** | 1.9621 | **1.3939** | **28.9%** |
> | **Short** | **2.3548** | 2.4557 | -4.3% |
>
> As shown, **ViTime achieves a remarkable 38.3% and 28.9% relative improvement in MASE for long and medium horizons, respectively.**
>
>
> We further investigated the short-horizon performance of ViTime, which is comparable but slightly weaker than that of TimesFM. We hypothesized a **potential data leakage issue** that unfairly benefits the TimesFM baseline. TimesFM's training data includes 100 billion tokens from Google Trends, which has a known and significant overlap with the **M4 dataset family**. The M4 datasets comprise a substantial portion of the **short-horizon** tasks in GIFT-EVAL, potentially inflating TimesFM's performance and compromising a true zero-shot comparison.
>
> To ensure a fair and rigorous evaluation, we performed a **leakage-controlled experiment** by excluding all M4 datasets from the benchmark and re-calculating the metrics for both models.
>
> **Table 2: Zero-Shot Performance on GIFT-EVAL (M4 Datasets Excluded)**
> | Forecast Horizon | TimesFM (MASE) | ViTime (MASE) | Relative Improvement |
> | :--- | :--- | :--- | :--- |
> | **Long** | 2.3706 | **1.4636** | **38.3%** |
> | **Medium** | 1.9621 | **1.3939** | **28.9%** |
> | **Short** | 2.6148 | **2.5687** | **1.8%** |
>
> The results from this controlled experiment are definitive. After mitigating the risk of data leakage, **ViTime consistently outperforms TimesFM across all forecast horizons—short, medium, and long.**
>
> We have integrated these new experiments, tables, and a detailed analysis of the data leakage issue into the camera-ready manuscript. We are confident that these additions fully address your concerns and substantially strengthen our paper. We thank you again for your constructive suggestion and the given opportunity of improving our work.
>
>
>
>
> **Manuscript Changes**
>
> We have added a citation for the GIFT-EVAL benchmark. In this camera-ready revision, we have made the following key updates:
>
> 1.  **Integrated a comprehensive new evaluation on the GIFT-EVAL benchmark.** The Appendix D.2 of the camera-ready paper now includes a summary of results comparing ViTime and TimesFM, along with a critical analysis of potential data leakage in the baselines.
> 2.  **Public Release of Full Experimental Details.** To ensure full transparency and reproducibility, the complete, unabridged experimental results and evaluation logs for the GIFT-EVAL benchmark are provided in our public code repository, which is released alongside this camera-ready paper.